# Vehicle-based in-situ observations of the water vapor isotopic composition across China: spatial and seasonal distributions and controls

Di Wang[1,2,3], Lide Tian[1,3], Camille Risi[2], Xuejie Wang[1,3], Jiangpeng Cui[4], Gabriel J. Bowen[5], Kei Yoshimura[6], Zhongwang Wei[7], Laurent Z.X Li[2]

[1] *Institute of International Rivers and Eco-security, Yunnan University, Kunming 650500, Yunnan, China*

[2] *Laboratoire de Météorologie Dynamique, IPSL, CNRS, Sorbonne Université, Campus Pierre et Marie Curie, Paris 75005, France*

[3] *Yunnan Key Laboratory of International Rivers and Transboundary Eco–security, Kunming 650500, Yunnan, China*

[4] *Sino-French Institute for Earth System Science, College of Urban and Environmental Sciences, Peking University, Beijing 100871, China*

[5] *Department of Geology and Geophysics, and Global Change and Sustainability Center, University of Utah, Salt Lake City, Utah 84108, USA*

[6] *Institute of Industrial Science, The University of Tokyo, Tokyo113-8654, Japan*

[7] *School of Atmospheric Sciences, Sun Yat-sen University, Guangzhou 510275, Guangdong, China*

*Corresponding author: wangdishi.mp@hotmail.com; ldtian@ynu.edu.cn.*

**Abstract**

Stable water isotopes are natural tracers in the hydrological cycle and have been applied in hydrology, atmospheric science, ecology, and paleoclimatology. However, the factors controlling the isotopic distribution, both at spatial and temporal scales, are debated in low and middle latitudes regions, due to the significant influence of large-scale atmospheric circulation and complex sources of water vapor. For the first time, we made in-situ observations of near-surface vapor isotopes over a large region (over 10000 km) across China in both pre-monsoon and monsoon seasons, using a newly-designed vehicle-based vapor isotope monitoring system. Combined with daily and multi-year monthly mean outputs from the isotope-incorporated global spectral model (Iso-GSM) and IASI satellite to calculate the relative contribution, we found that the observed spatial variations in both periods represent mainly seasonal-mean spatial variations, but are influenced by more significant synoptic-scale variations during the monsoon period. The spatial variations of vapor $\delta^{18}O$ are mainly controlled by Rayleigh distillation along air mass trajectories during the pre-monsoon period, but are significantly influenced by different moisture sources, continental recycling processes and convection during moisture transport in the monsoon period. Thus, the North-South gradient observed during the pre-monsoon period is counteracted during the monsoon period. The seasonal variation of vapor $\delta^{18}O$ reflects the influence of the summer monsoon convective precipitation in southern China, and a dependence on temperature in the North. The spatial and seasonal variations in d-excess reflect the different moisture sources and the influence of continental recycling. Iso-GSM successfully captures the spatial distribution of vapor $\delta^{18}O$ during the pre-monsoon period, but the performance is weaker during the monsoon period, maybe due to the underestimation of local or short-term high-frequency synoptic variations. These results provide an overview of the spatial distribution and seasonal variability of water isotopic composition in East Asia and their controlling factors, and emphasize the need to interpret proxy records in the context of the regional system.

**Keywords:** Vapor isotopes, Spatial distribution, Seasonal difference, East Asia, Moisture sources, Moisture propagation

## 1. Introduction

Stable water isotopes have been applied to study a wide range of hydrological and climatic processes (Gat, 1996;Bowen et al., 2019;West et al., 2009). This is because water isotopes vary with the water phases (e.g., evaporation, condensation), and therefore produce a natural labeling effect within the global water cycle. Stable isotopic signals recorded in natural precipitation archives are used in the reconstructions of ancient continental climate and hydrological cycles due to their strong relationship with local meteorological conditions. Examples include ice cores (Thompson, 2000;Yao et al., 1991;Tian et al., 2006), tree-ring cellulose (Liu et al., 2017), stalagmites (Van Breukelen et al., 2008), and lake deposits (Hou et al., 2007). However, unlike in polar ice cores, isotopic records in ice cores from low and middle latitudes regions have encountered challenges as temperature proxies (Brown et al., 2006;Thompson et al., 1997).

East Asian country China is the main distribution areas of ice cores in the low and middle latitudes (Schneider and Noone, 2007). Where the interpretation of isotopic variations in natural precipitation archives are debated, because they can be interpreted as recording temperature (Thompson et al., 1993;Thompson et al., 1997;Thompson et al., 2000;Thompson, 2000), regional-scale rainfall or strength of the Indian monsoon (Pausata et al., 2011), origin of air masses (Aggarwal et al., 2004;Risi et al., 2010). This is because China has a typical monsoon climate and moisture from several sources mix in this region (Wang, 2002;Domrös and Peng, 2012). In general, large parts of the country are affected by the Indian monsoon and the East Asian monsoon in summer, which bring humid marine moisture from the Indian Ocean, South China Sea, and Northwestern Pacific Ocean (Fig.1). During the non-monsoon seasons, the Westerlies influence most of northern China (Fig.1). Westerlies brings extremely cold and dry air masses. Occasional moisture flow from the Indian Ocean and/or Pacific Ocean brings moisture to southern China. Continental recycling, i.e. the moistening of the near-surface air by the evapo-transpiration from the land surface (transpiration by plants, evaporation of bare soil or standing water bodies,(Brubaker et al., 1993)), is also an important source of water vapor in both seasons. Some of the spatial and seasonal patterns of water vapor transport are imprinted in the observed station-based precipitation isotopes (Araguás-Araguás et al., 1998;Tian et al., 2007;Wright, 1993;Mei'e et al., 1985;Tan, 2014). However, precipitation isotopes can only be obtained at a limited number of stations and only on rainy days. The lack of continuous information makes it limited to analyze the effects of water vapor propagation and alternating monsoon and westerlies. In addition, the seasonal pattern and the spatial variation of water isotopes can strongly influenced by synoptic-scale processes, through their influence on moisture source, transport, convection and mixing processes (Klein et al., 2015;Sánchez-Murillo et al., 2019;Wang et al., 2021), which requires higher frequency observations. For example, some studies founded the impact of tropical cyclones (Gedzelman, 2003;Bhattacharya et al., 2022) the Northern Summer Intra-Seasonal Oscillation (BSISO) (Kikuchi, 2021), local or large-scale convections (Shi et al., 2020), cold front passages (Aemisegger et al., 2015), depressions(Saranya et al., 2018), and anticyclones (Khaykin et al., 2022) on water isotopes in the Asian region. Additional data and analysis refining our understanding of controls on the spatial and temporal variation of water isotopes in low-latitude regions therefore are needed.

Unlike precipitation, water vapor enters all stages of the hydrological cycle, experiencing frequent and intensive exchange with other water phases, in particular, directly linked with

water isotope fractionation. Furthermore, vapor isotopes can be measured in regions and periods without precipitation, and therefore, have significant potential to trace how water is transported, mixed, and exchanged (Galewsky et al., 2016;Noone, 2008), and to diagnose large-scale water cycle dynamics. Water vapor isotope data have been applied to various applications ranging from the marine boundary layer to continental recycling, and to various geographical regions from tropical convection to polar climate reconstructions (Galewsky et al., 2016). The development of laser-based spectroscopic isotope analysis made the precise, high-resolution and real-time measurements of both vapor $\delta^{18}O$ and $\delta^2H$ available in recent decades. However, most of the in-situ observation of water vapor isotopes are also station-based (e.g., (Li et al., 2020;Tian et al., 2020;Steen-Larsen et al., 2017;Aemisegger et al., 2014)), or performed during ocean cruises (Thurnherr et al., 2020;Bonne et al., 2019;JingfengLiu et al., 2014;Kurita, 2011;Benetti et al., 2017). One study made vehicle-based in-situ observations to document spatial variations, but this was restricted to the Hawaii island (Bailey et al., 2013). These observations provided new insight on moisture sources, synoptic influences, and sea surface evaporation fractionation processes. However, in-situ observations documenting continuous spatial variations at the continental scale do not exist. This paper presents the first isotope dataset documenting the spatial variations of vapor isotopes over a large continental region (over 10000 km) both during the pre-monsoon and monsoon periods, based on vehicle-based in-situ observations..

After describing our observed time series along the route (section 3.1 and 3.2), we quantify the relative contributions of seasonal-mean spatial variations and synoptic-scale variations that locally disturb the seasonal-mean to our observed time series (section 3.3). We show that our observed variations in both seasons are dominated by spatial variations, but are influenced by significant synoptic-scale variations during the monsoon period. On the basis of this, we then focus on analysing the main mechanisms underlying these distributions (section 4). Collectively, these data and analyses provide refined understanding of how the interaction of the summer monsoon and westerly circulation control water isotope ratios in East Asia.

## 2. Data and methods

2.1 Geophysical description

We conducted two campaigns to monitor vapor isotopes across a large part of China during the pre-monsoon (3[rd] to 26[th] March, 2019) and the monsoon (28[th] July to 18[th] August, 2018) periods, using a newly designed vehicle-based vapor isotope monitoring system (Fig.S1). The two campaigns run along almost the same route, with slight deviation in the far northeast of China (Fig.1). Our vehicle started from Kunming city in southwestern China, traveled northeast to Harbin, then turned to northwestern China (Hami), and returned to Kunming. The expedition traversed most of eastern China, with a total distance of above 10000km for each campaign.

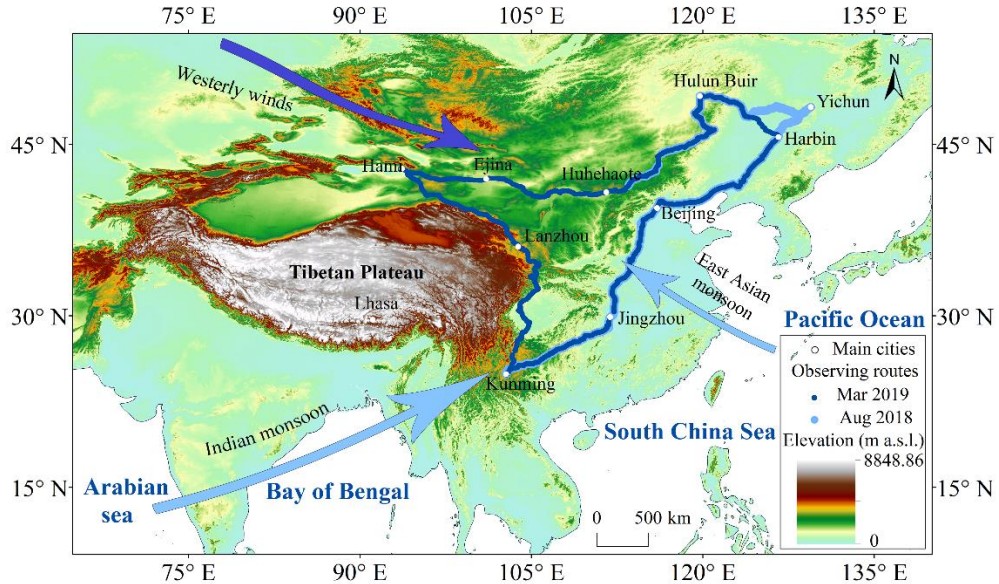


Fig.1. Topographical map of China, showing survey routes and the main atmospheric
circulation systems (arrows). Dark blue dots indicate the observation route for the 2019 pre-
monsoon period, and light blue dots show the observation route for the 2018 monsoon period,
with a slight deviation in the northeast.
2.2 Vapor isotope measurements
2.2.1 Isotopic definitions.
Isotopic compositions of samples were reported as the relative deviations from the
standard water (Vienna Standard Mean Ocean Water, VSMOW), using the δ-notion (McKinney
et al., 1950), where $R_{sample}$ and $R_{VSMOW}$ are the isotopic ratios ($H_2^{18}O/H_2^{16}O$ for $\delta^{18}O$, and
$^1H^2H^{16}O/H_2^{16}O$ for $\delta^2H$) of the sample and of the VSMOW, respectively:
$\delta = (R_{sample}/R_{VSMOW}-1)*1000$         (1)
The second-order d-excess parameter is computed based on the commonly used definition
(Dansgaard, 1964). The d-excess is usually interpreted as reflecting the moisture source and
evaporation conditions *(Jouzel et al., 1997)*, since the d-excess is more sensitive to non-
equilibrium fractionation occurs than $\delta^{18}O$:
d-excess $=\delta^2H-8*\delta^{18}O$         (2)

2.2.2 Instrument
We used a Picarro 2130i CRDS water vapor isotope analyzer fixed on a vehicle to obtain
large-scale in-suit measurements of near-surface vapor isotopes along the route. The analyzer
was powered by a lithium battery on the vehicle, enabling over 8 hours operation with a full
charge. Therefore, we only made measurements in daytime and recharged the battery at night.
The ambient air inlet of the instrument was connected to the outside of the vehicle, which was
1.5 m above ground, with a waterproof cover to keep large liquid droplets from entering. A
portable GPS unit was used to record position data along the route. The measured water vapor
mixing ratio and the $\delta^{18}O$ and $\delta^2H$ were obtained with a temporal resolution of ~1 second. The
dataset present in this study had been averaged to a 10-min temporal resolution after calibration,
with the horizontal footprint of about 15 km.
A standard delivery module (SDM) was used for the vapor isotope calibration during the surveys. The calibration protocols consists of humidity calibration (section 2.2.3), standard water calibration (section 2.2.4), and error estimation (section 2.2.5), following the methods of (Steen-Larsen et al., 2013).

### 2.2.3 Humidity-dependent isotope bias correction

The measured vapor isotopes are sensitive to air humidity (JingfengLiu et al., 2014;Galewsky et al., 2016), which vary substantially across our sampling route. The specific humidity measured by Picarro is very close to that measured by an independent sensor installed in the vehicle (Fig.4). The correlation between the humidity measured by the Picarro and the independent sensor are over 0.99, the slopes are approximately 1 and the average deviation are less than 1 g/kg both during pre-monsoon and monsoon periods. We develop a humidity-dependent isotope bias correction by measuring a water standard at different water concentration settings using the SDM. We define a reference level of 20,000 ppm of vapor humidity for our analysis (Eq. 3), since water vapor isotope measurement by Picarro is generally most accurate at this humidity, the calibrated vapor isotope with different air humidity would be (JingfengLiu et al., 2014;Schmidt et al., 2010):

$$\delta\_{measured} - \delta\_{humidity\ calibration} = f\ (humidity\_{measured} - 20000) \qquad (3)$$

where $\delta\_{measured}$ represents the measured vapor isotopes (the raw data), $\delta\_{humidity\ calibration}$ denotes the calibrated vapor isotopes, f is the equation of $\delta$ as a function of humidity , and humidity is in ppm. E.g., if we measured that f is $\delta = a*ln\ (humidity)+b$ by measuring standard water with different humidity, then the full equation for humidity-dependent isotope bias correction would be $\delta\_{measured} - \delta\_{humidity\ calibration} = a*ln\ (humidity\_{measured})+b - (a*ln\ (20000)+b)$.

We performed the humidity calibration before and after each campaign. In the calibration, the setting of humidity covered the actual range of humidity in the field. In the dry pre-monsoon period of 2019, the humidity was less than 5000 ppm along a large part of the route. In this case, we performed additional calibration tests with the humidity less than 5000 ppm after the field observations to guarantee the accuracy of the calibration results. The humidity-dependence calibration function is considered constant throughout each campaign (which each lasted less than 24 days).

### 2.2.4 Measurement normalization

All measured vapor isotope values were calibrated to the VSMOW-SLAP scale using two laboratory standard waters ($\delta^{18}O = -10.33‰$ and $\delta^2H = -76.95‰$, $\delta^{18}O = -29.86‰$ and $\delta^2H = -222.84‰$) covering the range of the expected ambient vapor values. We made the normalization test prior to the daily measurements (two humidity levels for each standard water). We adjusted the amount of the liquid standard injected everyday to keep the humidity of the standard waters consistent with the outside vapor measurements. Our calibration shows that no significant drift of the standard values were observed over time in the observation periods (For two standard waters, the standard deviation of standard measurements are 0.2‰ and 0.11‰ for $\delta^{18}O$, and 1.16‰ and 1.2‰ for $\delta^2H$ during the pre-monsoon period of 2019. During the monsoon period of 2018, the standard deviation of standard measurements are 0.09‰ and 0.06‰ for $\delta^{18}O$, and 0.6‰ and 0.33‰ for $\delta^2H$.).

2.2.5 Error estimation
We estimate the uncertainty based on the error between the measured (after calibration)
and true values of the two standards used during the campaigns. The estimated uncertainty is
in the range of -0.05~0.17 for $\delta^{18}O$, 0.11~1.19 for $\delta^2H$, and -0.81~1.23‰ for d-excess during
the pre-monsoon period of 2019, with the humidity ranges from 2000 ppm to 29000 ppm.
During the monsoon period of 2018, the range of uncertainty is -0.10~0.55‰ for $\delta^{18}O$, -
0.94~3.74‰ for $\delta^2H$, and -1.18~1.49‰ for d-excess, with the humidity ranges from 4000 to
34000 ppm.

2.2.6 Data processing
A few isotope measurements with missing GPS information were excluded from the
analysis. Since we want to focus on large-scale variations, we also removed the observations
during raining or snowing, to avoid situations where hydrometeor evaporation significantly
influenced the observations (Tian et al., 2020). Such data represents only 0.03% and 0.05% of
our observations, respectively (totally 48 data during pre-monsoon season and 59 data during
the monsoon season). We observed several d-excess pulses with extremely low values as low
as −18.0‰ during the pre-monsoon period and -4.9‰ during the monsoon period. These low
values are unusual in previous natural vapor isotope studies and occurred mostly when the
measurement vehicle was entering or leaving cities and/or stuck in traffic jams, and have a
much lower intercept in the linear $\delta^{18}O$ - $\delta^2H$ relationship (Fig.S6). Previous studies on urban
vapor isotopes (Gorski et al., 2015;Fiorella et al., 2018;Fiorella et al., 2019) showed that the
vapor d-excess closely tracked changes in $CO_2$ through inversion events and during the daily
cycle dominated by patterns of human activity, and combustion-derived water vapor is
characterized by a low d-excess value due to its unique source. We also find that the d-excess
values are especially low when the vehicle was in cities in the afternoon. The values increased
to normal during the night. This diurnal cycle is likely related to the emission intensity and
atmospheric processes (Fiorella et al., 2018). Some of these d-excess anomalies are not
excluded from being affected by the baseline effects emerging from rapid changes in
concentrations of different trace gases (Johnson and Rella, 2017;Gralher et al., 2016). We
therefore excluded these data (133 data points during the pre-monsoon period and 62 data points
during the monsoon period, represents 0.10% and 0.06% of our observations, respectively) in
the discussion on the general spatial feature (except Fig.4). Outside towns, country sources,
such as irrigation, farms, and power plants, cannot be completely ruled out. However, we expect
their influence to be much smaller than large-scale spatial variations.
2.3 Meteorological observations
We fixed a portable weather station on the roof of the vehicle to obtain air temperature (T),
dew-point temperature ($T_d$), air pressure (Pres) and relative humidity (RH). All sensors were
located near the ambient air intake. The specific humidity (q) of the near-surface air was
calculated from $T_d$ and Pres. Meteorological data, GPS location data and vapor isotope data
were synchronized according to their measurement times. And all of them also had been
averaged to a 10-min temporal resolution.
National Centers for Environmental Prediction/ National Center for Atmospheric Research
(NCEP/NCAR) 2.5-deg global reanalysis data are used to determine the large-scale factors
influencing the spatial pattern of the vapor isotopes, including the surface T, q, U-wind and V-
wind, and RH, which are available at
https://psl.noaa.gov/data/gridded/data.ncep.reanalysis.surface.html. Some missing
meteorological data (during the pre-monsoon period: q on 8th March and 18th March 2019;
during the monsoon period: T and q from 28th July to 31st July, q on 5th August) along the survey
routes due to instrument failure are acquired from the NCEP/NCAR reanalysis data. To match
the vapor isotope data along the route, we linearly interpolate the NCEP/NCAR data to the
location and time of each measurement. The interpolated T and q from NCEP/NCAR are highly
correlated with our measurement as shown in Figure 4h and j. The 1-deg precipitation amount
(P) from the Global Precipitation Climatology Project (GPCP) are used
(https://www.ncei.noaa.gov/data/global-precipitation-climatology-project-gpcp-daily/access/).
When comparing the time series of GPCP data with our observed isotopes, we linearly
interpolate the daily GPCP data to the location of each observation location (P-daily).We also
used the average of the GPCP precipitation over the entire observation period of about one
month for each observation location (P-mean). The 2.5-deg outgoing longwave radiation (OLR)
data can be obtained from NOAA
(http://www.esrl.noaa.gov/psd/data/gridded/data.interp_OLR.html).
2.4 Back-trajectory calculation and categorizing regions based on air mass origin
The vapor isotope composition is a combined result of moisture source (Tian et al.,
2007;Araguás-Araguás et al., 1998), condensation and mixing processes along the moisture
transport route (Galewsky et al., 2016). To interpret the observed spatial-temporal distribution
of vapor isotopes, we start with a diagnosis of the geographical origin of the air masses and
then analyze the processes along the back-trajectories.
To trace the geographical origin of the air masses, the HYSPLIT-compatible
meteorological dataset of the Global Data Assimilation System (GDAS) is used (available at
ftp://arlftp.arlhq.noaa.gov/pub/archives/gdas1/). We select the driving locations every 2 hours
as starting points for the backward trajectories, and make 10-day back-trajectories from 1000
m above ground using the Hybrid Single Particle Lagrangian Integrated Trajectory Model 4
(HYSPLIT4) (Draxler and Hess, 1998). This is representative of the water vapor near the
ground (Guo et al., 2017;Bershaw et al., 2012), since most water vapor in the atmosphere is
within 0–2 km above ground level (Wallace and Hobbs, 2006). The T, q, P and RH along the
back-trajectories are also interpolated by HYSPLIT4 model (Fig.2).

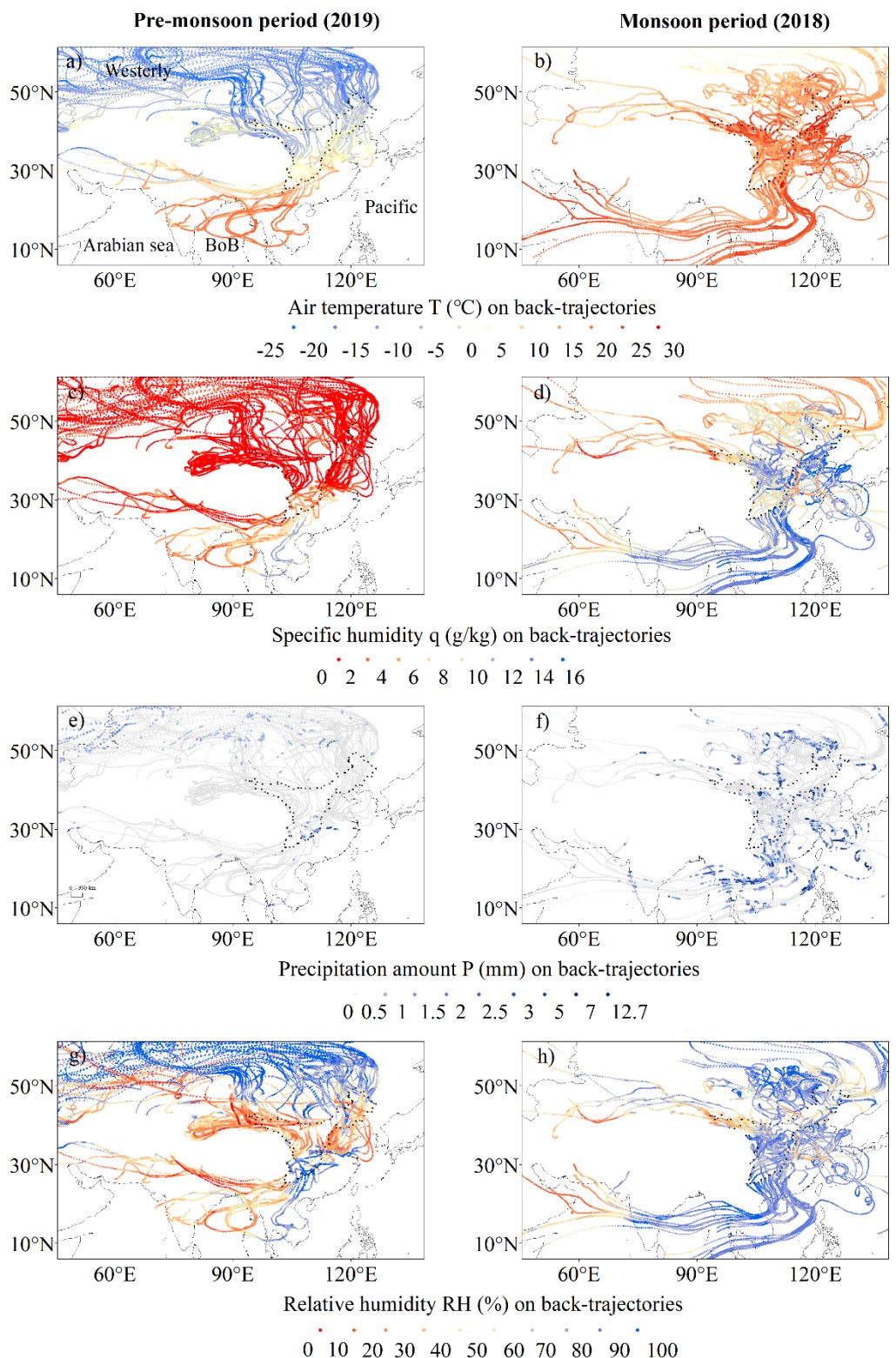

**Pre-monsoon period (2019)**      **Monsoon period (2018)**

Air temperature T (℃) on back-trajectories

-25  -20  -15  -10  -5  0  5  10  15  20  25  30

Specific humidity q (g/kg) on back-trajectories

0  2  4  6  8  10  12  14  16

Precipitation amount P (mm) on back-trajectories

0  0.5  1  1.5  2  2.5  3  5  7  12.7

Relative humidity RH (%) on back-trajectories

0  10  20  30  40  50  60  70  80  90  100


Fig.2 Meteorological conditions simulated by HYSPLIT4 model along the 10-day air back-
trajectories for the on-route sampling positions during the two surveys: (a, b) air temperature T
(℃), (c, d) specific humidity q (g/kg), (e, f) precipitation amount P (mm) and (g, h) relative
humidity RH (%). The left panel is for the pre-monsoon period and the right is for the monsoon
period. The driving locations and time every 2 hours are used as starting points. Note: BoB is
the abbreviation for the Bay of Bengal.

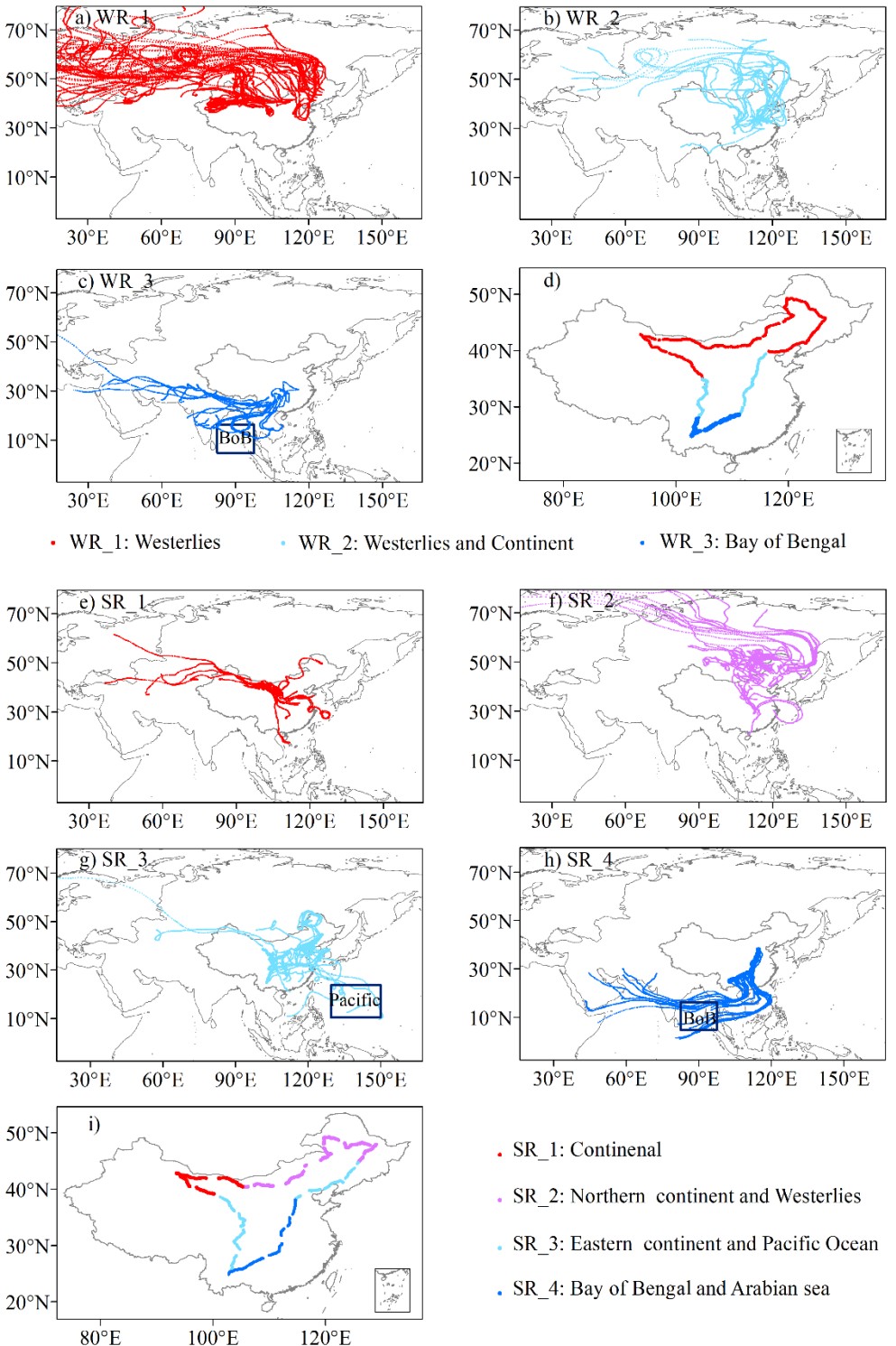


Fig.3 The backward trajectory results (a, b and c for the pre-monsoon period, and e, f, g and h for the monsoon period) and the dividing of the study zones based on geographical origin of the air masses (d for the pre-monsoon period and i for the monsoon period). Note: BoB is the abbreviation for the Bay of Bengal.

Based on the tracing results from HYSPLIT4 model, we speculate on the potential water vapor sources (Fig.3 and Table 1):

During the pre-monsoon period, we categorize our domain into 3 regions (Table 1).

(1) In northern China (WR_1), the air is mainly advected by the Westerlies.
(2) In central China (WR_2), the air also comes from the Westerlies but with a slower wind
speed (as shown by the shorter trajectories in 10 days), suggesting potential for greater
interaction with the land surface and more continental recycling as moisture source.
(3) In southern China (WR_3), trajectories come from the Southwest and South with
marine moisture sources from the Bay of Bengal (BoB).
During the monsoon period, we categorize our domain into 4 regions (Table 1):
(1) In northwestern China (SR_1), most air masses also spend considerable time over the
continent, suggesting some of the vapor can be recycled by continental recycling.
(2) In northeastern China (SR_2), trajectories mainly come from the North and though the
Westerlies.
(3) In central China (SR_3), both in its eastern (from Beijing to Harbin) and western part,
trajectories mainly come from the East. This suggests that vapor mainly comes from the Pacific
Ocean, or from continental recycling over eastern and central China.
(4) In southeastern China (SR_4), trajectories come from the South, suggesting marine
moisture sources from the Arabian Sea and the BoB.

Table 1. The dividing of the study zones based on moisture sources and corresponding
vapor $\delta^{18}$O-$\delta^2$H relationship

| Pre-monsoon period (2019) | | | | |
|---|---|---|---|---|
| | Water sources (Fig.3) | Region (China) | Climate background | $\delta^{18}$O-$\delta^2$H relationship |
| WR_1 | Westerlies | The north | Westerlies domain | $\delta^{18}$O=8.04$\delta^2$H+12.00 ($r^2$=0.99, n=750, q<0.01) |
| WR_2 | Westerlies and Continent | The middle | Transition domain | $\delta^{18}$O=8.26$\delta^2$H+23.15 ($r^2$=0.99, n=281, q<0.01) |
| WR_3 | Bay of Bengal (BoB) | The south | Monsoon domain | $\delta^{18}$O=7.98$\delta^2$H+17.13 ($r^2$=0.94, n=158, q<0.01) |
| Monsoon period (2018) | | | | |
| | Water sources (Fig.3) | Region (China) | Climate background | $\delta^{18}$O-$\delta^2$H relationship |
| SR_1 | Continent | The northwest | Transition domain | $\delta^{18}$O=8.31$\delta^2$H+20.92 ($r^2$=0.99, n=200, q<0.01) |
| SR_2 | Northern continent & Westerlies | The northeast | Transition domain | $\delta^{18}$O=7.53$\delta^2$H+5.13 ($r^2$=0.98, n=294, q<0.01) |
| SR_3 | Eastern continent & Pacific Ocean | The middle and west | Transition domain | $\delta^{18}$O=7.49$\delta^2$H+7.09 ($r^2$=0.97, n=271, q<0.01) |
| SR_4 | BoB & Arabian sea | The southeast | Monsoon domain | $\delta^{18}$O=8.21$\delta^2$H+17.81 ($r^2$=0.99, n=195, q<0.01) |


2.5 General circulation model simulation and satellite measurements
To disentangle the spatial and synoptic influences, we use surface layer variables from an
isotope-enabled general circulation model (Iso-GSM) simulations (Yoshimura and Kanamitsu,
2009) at 1.915° x 1.875° and the lowest level (the altitude are about 2950m) isotope retrievals

from satellite Infrared Atmospheric Sounding Interferometer (IASI) at 1°x1°. For both dataset, we use the outputs corresponding to the observation location and the observation date (daily outputs), and the multi-year monthly-mean outputs (March monthly for the pre-monsoon period and August monthly for the monsoon period) for each observation location from 2015 to 2020. When interpolating daily/multi-year monthly outputs, we select the nearest grid point for a given latitude and longitude of each measurement. For Iso-GSM simulations, because of the coarse resolution of the model, there is a difference between the altitude observed along the sampling route and that of the nearest grid point. Therefore, we correct the outputs of Iso-GSM for this altitude difference (the method is given in III. Supplementary Text). Since the satellite only retrieves $\delta^2H$, we just use $\delta^2H$ outputs of Iso-GSM and satellite to quantify the relative contributions of seasonal-mean and synoptic-scale variations (section 3.3). Other than that, our discussion focuses on $\delta^{18}O$ and d-excess. The variations of $\delta^2H$ are consistent with those of $\delta^{18}O$. We also interpret the biases in Iso-GSM after we understand the factors influencing the spatial and seasonal variation of vapor isotopes (section 4.6).

2.6 Method to decompose the observed daily variations

The temporal variations observed along the route for a given period represent a mixture of synoptic-scale perturbations, and of seasonal-mean spatial distribution:

$$\delta^2H_{\_daily} = \delta^2H_{\_seaso} + \delta^2H_{\_synoptic} \qquad (4)$$

The first term represents the contribution of seasonal-mean spatial variations, whereas the second term represents the contribution of synoptic-scale variations. Since these relative contributions are unknown, we use outputs from Iso-GSM and IASI. The daily variations of $\delta^2H$ simulated by Iso-GSM also represent a mixture of synoptic-scale perturbations and seasonal-mean spatial distribution, but with some errors relative to reality:

$$\delta^2H_{\_daily\_Iso\text{-}GSM} = \delta^2H_{\_seaso\_Iso\text{-}GSM} + \delta^2H_{\_synoptic\_Iso\text{-}GSM} \qquad (5)$$

where $\delta^2H_{\_daily\_Iso\text{-}GSM}$ is the daily outputs of $\delta^2H$ for each location, $\delta^2H_{\_seaso\_Iso\text{-}GSM}$ is the multi-year monthly outputs of $\delta^2H$ for each location, and $\delta^2H_{\_synoptic\_Iso\text{-}GSM} = \delta^2H_{\_daily\_Iso\text{-}GSM} - \delta^2H_{\_seaso\_Iso\text{-}GSM}$, each of these terms are affected by errors relative to observations:

$$\delta^2H_{\_daily\_Iso\text{-}GSM} = \delta^2H_{\_daily} + \in = (\delta^2H_{\_seaso} + \in_{\_seaso}) + (\delta^2H_{\_synoptic} + \in_{\_synoptic}) \qquad (6)$$

where $\in_{\_seaso}$ and $\in_{\_synoptic}$ are the errors on $\delta^2H_{\_seao\_Iso\text{-}GSM}$ and $\delta^2H_{\_synoptic\_Iso\text{-}GSM}$ relative to reality, respectively, $\in$ is the sum of $\in_{\_seaso}$ and $\in_{\_synoptic}$.

Correspondingly, $\delta^2H_{\_daily} = \delta^2H_{\_daily\_Iso\text{-}GSM} - \in = (\delta^2H_{\_seaso\_Iso\text{-}GSM} - \in_{\_seaso}) + (\delta^2H_{\_synoptic\_Iso\text{-}GSM} - \in_{\_synoptic}) \qquad (7)$

These individual error components $\in_{\_seaso}$ and $\in_{\_synoptic}$ are unknown, but we know the sum of them ($\in$), i.e. the difference between daily outputs and observations. For the decomposition, we made two extreme assumptions to estimate upper and lower bounds on the contribution values:

(1) If we assume that the error is purely synoptic, i.e. $\in = \in_{\_synoptic}$, and $\in_{\_seaso} = 0$, then:

$$\delta^2H_{\_daily} = \delta^2H_{\_seaso\_Iso\text{-}GSM} + (\delta^2H_{\_synoptic\_Iso\text{-}GSM} - \in) \qquad (8)$$

To evaluate the contribution of these two terms, we calculate the slopes of $\delta^2H_{\_daily}$ as a function of $\delta^2H_{\_seaso\_Iso\text{-}GSM}$ ($a_{\_seaso}$), and of $\delta^2H_{\_daily} - \delta^2H_{\_seaso\_Iso\text{-}GSM}$ ($a_{\_synoptic}$). The relative contributions of spatial and synoptic variations correspond to $a_{\_seaso}$ and $a_{\_synoptic}$ respectively. This will be the upper bound for the contribution of synoptic-scale variations, since some of

the systematic errors of Iso-GSM will be included in the synoptic component. This is equivalent
to using the seasonal-mean of Iso-GSM and the raw time series of observations.
(2) If we assume that the error is purely seasonal-mean, i.e. $\in = \in_{\_seaso}$, and $\in_{\_synoptic} = 0$,
then:
$\delta^2 H_{\_daily} = (\delta^2 H_{\_seaso\_Iso\text{-}GSM} - \in) + \delta^2 H_{\_synoptic\_Iso\text{-}GSM}.$          (9)
To evaluate the contribution of these two terms, we calculate the slopes of $\delta^2 H_{\_daily\_Iso\text{-}GSM}$
as a function of $\delta^2 H_{\_seaso\_Iso\text{-}GSM} - \in$ ($a_{\_seaso}$), and of $\delta^2 H_{\_daily} - (\delta^2 H_{\_seaso\_Iso\text{-}GSM} - \in)$ ($a_{\_synoptic}$).
This will be the lower bound for the contribution of synoptic-scale variations, since we expect
Iso-GSM to underestimate the synoptic variations.
The same analysis is also performed for $\delta^2 H$ retrieved from IASI, and the Iso-GSM
simulation q (Table 2) and reanalysis q (Table 3).
**3. Spatial and seasonal variations**
3.1 Raw time series
Our survey of the vapor isotopes yields two snapshots of the isotopic distribution along
the route (Fig.4 & Fig.5). Figure 4 shows the variations of observed 10-min averaged surface
vapor $\delta^{18}O$ and d-excess along the survey route across China during the pre-monsoon and
monsoon campaigns. The figure also shows the concurrent meteorological data from the
weather station installed on the vehicle and the water vapor content recorded by the Picarro
water vapor isotope analyzer as a comparison. We extract daily precipitation amount (P-daily)
and average precipitation amount over the entire observation period of about one month for
each observation location (P-mean) (mm/day) from GPCP. The vapor $\delta^{18}O$ shows high
magnitude variations in both seasons. A general decreasing-increasing trend overlapped with
short-term fluctuations is observed during the pre-monsoon period, whereas no general trend
but frequent fluctuations characterized the monsoon period. The $\delta^{18}O$ range is much larger
during the pre-monsoon period (varying between -44‰ and -8‰) than during the monsoon
period (from -11‰ to -23‰). Most measured vapor d-excess values ranges from 5 to 25‰
during the pre-monsoon period and from 10 to 22‰ during the monsoon period.
Comparison with the concurrently observed meteorological data shows a robust air
temperature (T) dependence of the vapor $\delta^{18}O$ variations. In particular, the general trend of $\delta^{18}O$
is roughly consistent with T variation during the pre-monsoon period (Fig.4a and g). During
the pre-monsoon period, humidity (Fig.4e and i), P-mean (Fig.4k) and vapor $\delta^{18}O$ (Fig.4a) are
much higher in southwestern China (at the beginning and end of the campaign) than in any
other regions. Humidity, q, and P-mean also vary consistently throughout the route during the
monsoon period (Fig.4f, j, l). Synoptic effects on the observed vapor isotopes are discussed in
detail in Section 4.3 and 4.6.

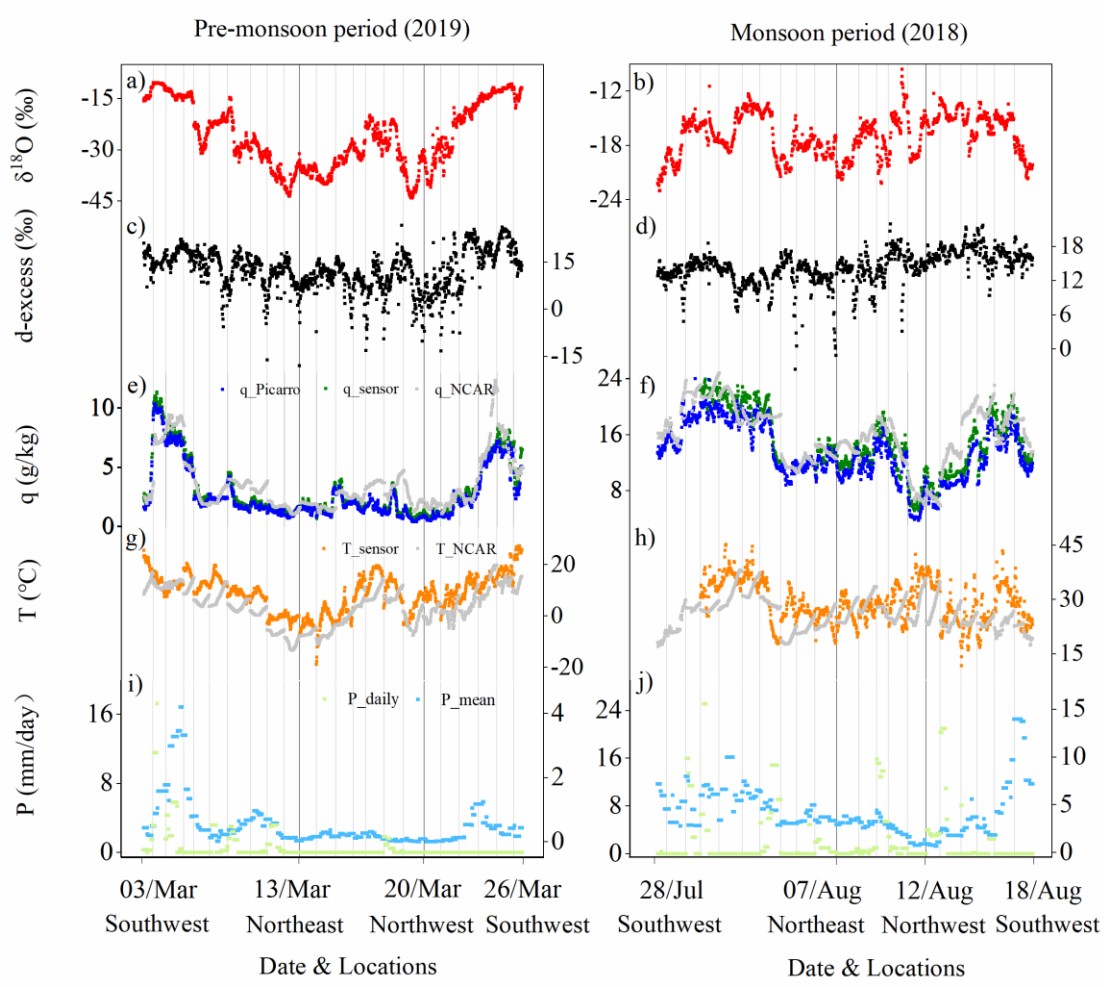

Fig.4. Measured vapor isotopic compositions and concurrent meteorological conditions along the survey routes during the pre-monsoon period (the left panel) and monsoon period (the right panel). (a, b) vapor $\delta^{18}O$ (‰); (c, d) vapor d-excess (‰); (e, f) specific humidity q (g/kg) measured by sensor (in green), measured by Picarro (in blue) and linearly interpolate from NCAR reanalysis (in grey); (g, h) air temperature T (°C) measured by Picarro (in orange) and linearly interpolate from NCAR reanalysis (in grey); (i, j) the daily precipitation amount P-daily (in green, mm/day) and average precipitation amount over the entire observation period of about one month for each observation location P-mean (in blue, mm/day) extract from GPCP. Notes: the gray dots are T and q linearly interpolate from NCAR reanalysis to compensate for missing observations; Gray vertical lines space the observations for one day.

3.2 Spatial variations

The spatial distribution of the observed vapor $\delta^{18}O$ and d-excess during the two surveys in different seasons are presented in Figure 5. During the pre-monsoon period, we find a south-north gradient of vapor $\delta^{18}O$ (Fig.5a). The vapor $\delta^{18}O$ ranges from -8~ -16‰ in southern China to as low as -24 ~ -44‰ in the North. A roughly similar spatial pattern is observed for the vapor d-excess during the pre-monsoon period (Fig.5c). The d-excess value ranges from 10 to 30‰ in southern China and from -10 to +20‰ (most observations with values from 5 to +20‰) in northern China. In previous studies, a higher precipitation d-excess during the pre-monsoon period was also observed in the Asian monsoon region owing to the lower relative humidity

(RH) at the surface in the moisture source region (Tian et al., 2007;Jouzel et al., 1997). The same reason probably explains the higher vapor d-excess in southern China observed here. Alternatively, the high d-excess in south China could also result from the moisture flow from Indian/Pacific Ocean, or from the deeper convective mixed layer in south China compared to north China. The lower d-excess values (as low as −10‰ to 10‰) in northern China (between 38°N and 51°N) have rarely been reported in earlier studies. The spatial distribution of the observed vapor d-excess could reflect the general latitudinal gradient of d-excess observed at the global-scale, with a strong poleward decrease in midlatitudes (between around 20 to 60°), which were found in previous studies on large-scale distribution of d-excess in vapor (Thurnherr et al., 2020;Benetti et al., 2017) and precipitation (Risi et al., 2013a;Terzer-Wassmuth et al., 2021;Pfahl and Sodemann, 2014;Bowen and Revenaugh, 2003), based on both observations and modelling. During the monsoon period, the lowest values of vapor $\delta^{18}$O are found in southwestern and northeastern China, with a range of -23‰ to -19‰ (Fig.5b). Higher vapor $\delta^{18}$O values up to -11‰ are founded in central China. The vapor d-excess values (Fig.5d) in western and northwestern China (91°E-109°E, 24°N-43°N) are roughly between 16 and 22‰, higher than in eastern China (mostly between 0 and 16‰).

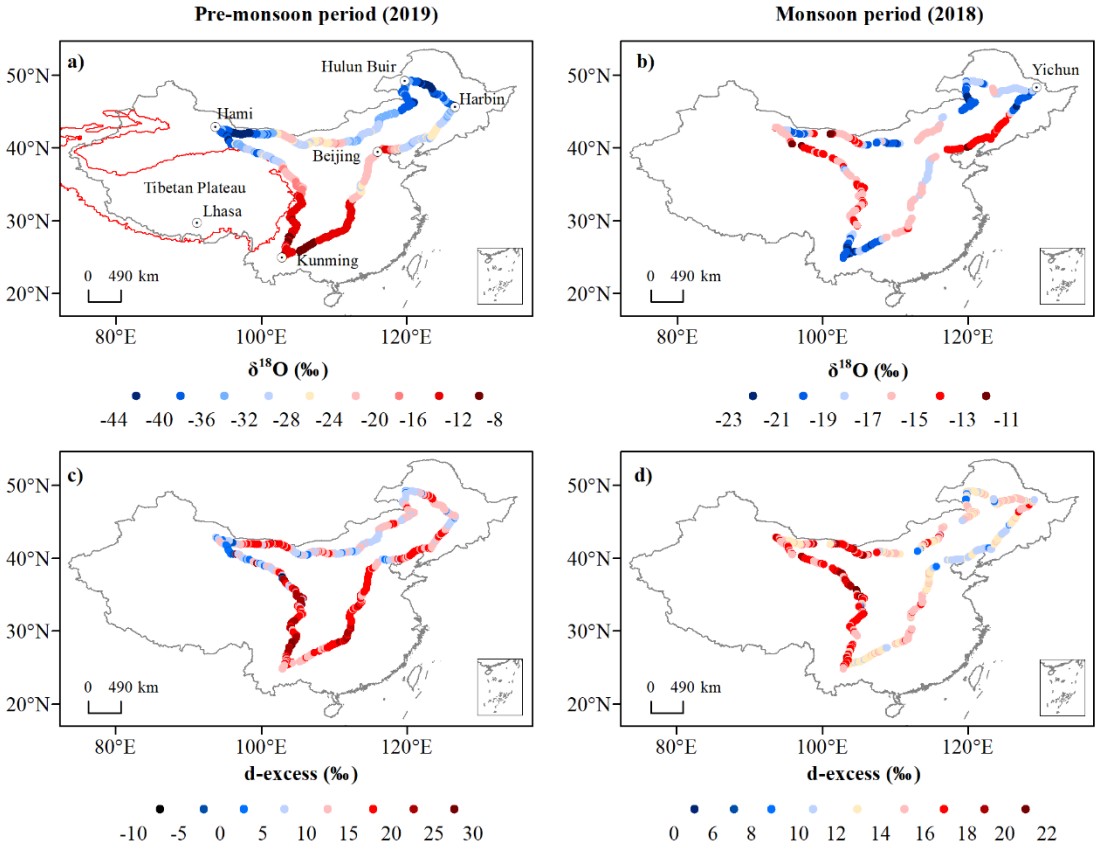

Fig.5. Spatial distribution of vapor $\delta^{18}$O (a, b) and d-excess (c, d) during the pre-monsoon period (the left panel) and monsoon period (the right panel).

We don't know whether these apparent spatial variations represent the seasonal-mean, or whether it is mainly affected by synoptic perturbations. We therefore use Iso-GSM simulation results and IASI satellite measurements to quantify the relative contributions of seasonal-mean and synoptic perturbations in section 3.3.

3.3 Disentangling seasonal-mean and synoptic variations

Figure 6 shows the comparison of the measured vapor $\delta^2H$, simulated $\delta^2H$ from Iso-GSM, and the $\delta^2H$ retrieves from IASI. Iso-GSM captures the variations in observed vapor $\delta^2H$ well during the pre-monsoon period, with correlation coefficient of r = 0.84 (p<0.01) (Table S3). The daily simulation results during the monsoon period are roughly in the range of observations, but detailed fluctuations are not well captured, with r = 0.24 (p>0.05) (Table S3). The largest differences occur in the SR_1 zone. IASI captures variations better than Iso-GSM during the monsoon period, with r = 0.42 (p>0.05). IASI observes over a broad range of altitudes above the ground level, so we expect lower $\delta^2H$ in IASI relative to ground-surface observations, but the variations of vapor isotopes are vertically coherent (Fig.6). The systematic differences between IASI and ground-level observations do not impact the slope of the correlation, and thus doesn't impact the contribution estimation.

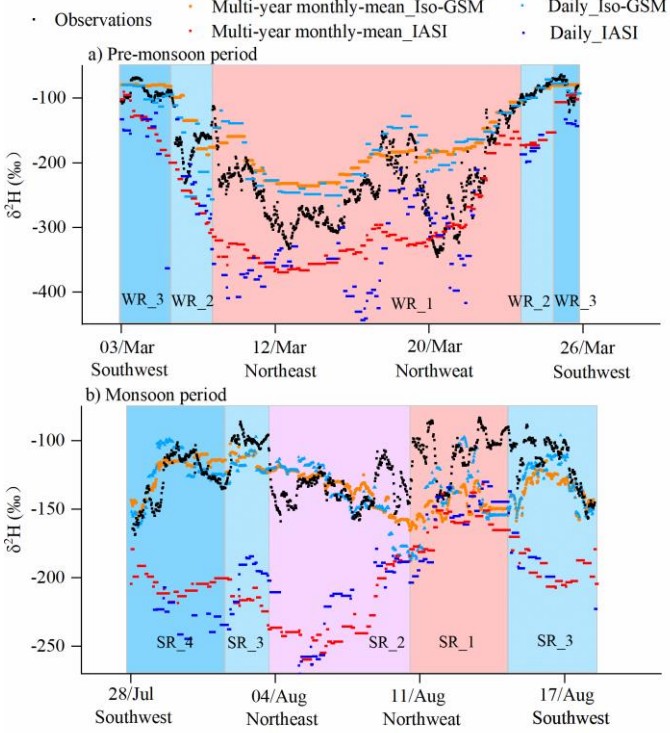

Fig.6 Comparison of observed vapor $\delta^2H$ (observations) with outputs of isotope-enabled general circulation model Iso-GSM and satellite IASI during the pre-monsoon period (a) and monsoon period (b). The results in this graph are from the daily and multi-year monthly outputs for the sampling locations.

The multi-year monthly-mean of $\delta^2H$ are smoother but similar to those for the daily outputs both from Iso-GSM and IASI (Fig.6). Using the method in section 2.6, taking into account the error, we calculate the relative contribution ranges of the seasonal-mean and synoptic-scale on our observed variations using q and $\delta^2H$ from Iso-GSM simulations, q from NCEP/NCAR reanalysis, and $\delta^2H$ from IASI.

**Table 2** The relative contribution ( in fraction) of spatial variations for a given season (a_seaso) and of synoptic-scale variations (a_synopic) to the daily variations of q and $\delta^2H$ simulated

by Iso-GSM. We checked that the sum of $a_{\_seaso}$ and $a_{\_synoptic}$ is always 1. The two values indicate the lower and upper bounds as calculated from equations 8 and 9.

| Period | Data | Variables | Controbutions | |
|---|---|---|---|---|
| | | | $a_{\_seaso}$ | $a_{\_synoptic}$ |
| Pre-monsoon (2019) | Iso_GSM | q | 0.73~1.02 | 0.27~-0.02 |
| | | $\delta^2H$ | 0.60 ~0.98 | 0.40~0.02 |
| | IASI | $\delta^2H$ | 1.06~0.94 | -0.06~0.06 |
| Monsoon (2018) | Iso_GSM | q | 0.71~0.82 | 0.29~0.18 |
| | | $\delta^2H$ | 0.09~0.87 | 0.91~0.13 |
| | IASI | $\delta^2H$ | 0.53~0.84 | 0.47~0.16 |

**Table 3** The same as Table 2, but for reanalysis q.

| Period | Variables | Controbutions | |
|---|---|---|---|
| | | $a_{\_seaso}$ | $a_{\_synoptic}$ |
| Pre-monsoon (2019) | q | 0.77~0.92 | 0.23~0.08 |
| Monsoon (2018) | q | 0.69~0.95 | 0.31~0.05 |

During the pre-monsoon period, based on both the Iso-GSM simulation and NCEP/NCAR reanalysis, we can find that the seasonal-mean contribution to the measured q is higher than the synoptic-scale contribution: $a_{\_seaso}$ is 73%~102% from Iso-GSM and 77%~92% from reanalysis, whereas $a_{\_synoptic}$ is 27% ~ -2% from Iso-GSM and 23% ~ 8% from reanalysis (Table 2 and Table3). The relative contribution of seasonal-mean spatial variations to the total measured variations in $\delta^2H$ (60% ~ 98%) is also higher than that of synoptic-scale variations (40% ~2%). This suggests that the observed variability in q and $\delta^2H$ is mainly due to spatial variability, and marginally due to synoptic-scale variability. During the monsoon, seasonal-mean spatial variations are also the main contributions to the observed variations of q ($a_{\_seaso}$ is 71% ~ 82% from Iso-GSM and 69% ~ 95% from reanalysis, whereas $a_{\_synoptic}$ is 18% ~ 29% from Iso-GSM and 5% ~ 31% from reanalysis). Since Iso-GSM doesn't capture daily variations of $\delta^2H$ very well during the monsoon period, the relative contribution has a large threshold range ($a_{\_seaso}$ is 9%~87%, $a_{\_synoptic}$ is 91% ~ 13%) after accounting for the errors. Therefore, we can not conclude the dominate contribution on $\delta^2H$ from Iso-GSM outputs. IASI, which has a higher correlation with observations, provides an more credible range of $a_{\_seaso}$ about 53% ~ 84%, and $a_{\_synoptic}$ 16% ~ 47%. These suggests that during the monsoon period, the synoptic contribution can be significant, but not dominate. Having understood the factors influencing the spatial and seasonal variation of vapor isotopes in section 4, we will be able to better understand the reasons for the inconsistent performance of Iso-GSM during the pre-monsoon and monsoon periods (in section 4.6).

3.4 Seasonal variations

During the monsoon season, synoptic-scale and intra-seasonal variations contribute significantly to the apparent spatial patterns. However, since these variations are not dominate, and have a smaller amplitude than seasonal differences, the comparison of the two snapshots do provide a representative picture of the climatological seasonal difference.

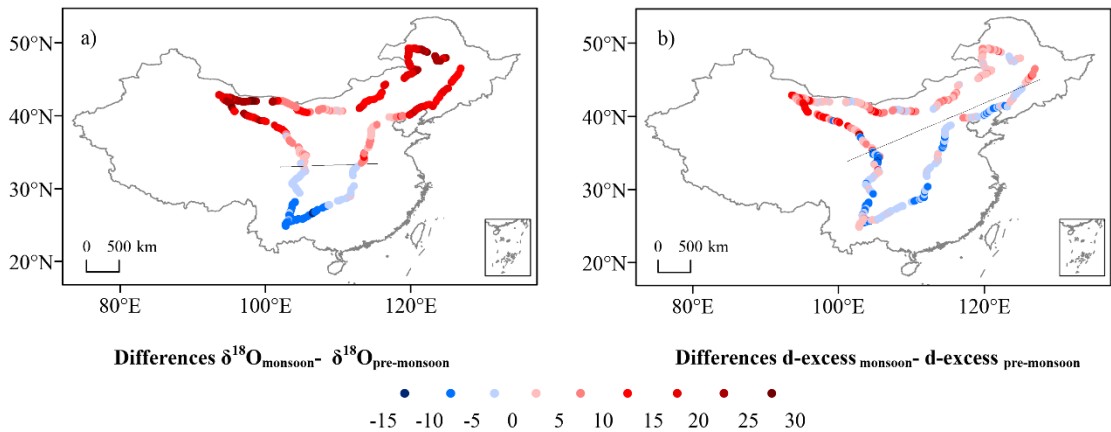

Fig.7 Spatial distribution of the isotope differences ($\delta^{18}O_{monsoon}$ - $\delta^{18}O_{pre-monsoon}$ (a) and d-excess$_{monsoon}$ - d-excess$_{pre-monsoon}$ (b)) for the observation locations. The solid black lines separate the areas of positive and negative values of the differences.

The climate in China features strong seasonality and it is captured in the snapshots of vapor isotopes (Fig.7). Since the observation routes of the two surveys are almost identical, we make a seasonal comparison of the observed vapor isotopes during the two surveys. The lines are drawn to distinguish between positive and negative values of seasonal isotopic differences. The seasonal differences $\delta^{18}O_{monsoon}$ - $\delta^{18}O_{pre-monsoon}$ (Fig.7a) show opposite sign in northern and southern China. In northern China, water vapor $\delta^{18}O$ values are higher during the monsoon period than during the pre-monsoon period, while the opposite are true in southern China. The boundary is located around 35 °N. The largest seasonal contrasts occur in southwest, northwest and northeast China, with seasonal $\delta^{18}O$ differences of -15 ‰, 30 ‰, and 30 ‰, respectively.

We also find a spatial pattern of vapor d-excess seasonality (Fig.7b). The line separating the areas of positive and negative values of the d-excess$_{monsoon}$ - d-excess$_{pre-monsoon}$ differences coincides with the 120 mm P-mean line (Fig.S2 f). In southeastern China, the water vapor d-excess is lower during the monsoon period than during the pre-monsoon period. The pattern of seasonal water vapor d-excess in northwestern China is the opposite. The two boundary lines separating the seasonal variations of $\delta^{18}O$ and d-excess do not overlap, suggesting different controls on water vapor $\delta^{18}O$ and d-excess.

## 4. Understanding the factors controlling the spatial and seasonal distributions

To interpret the spatial and seasonal variations observed both across China and in each region defined in section 2.4, we investigate q–$\delta$ diagrams (section 4.1), $\delta^{18}O$-$\delta^2H$ relationships (section 4.2), relationships with meteorological conditions at the local and regional scale (sections 4.3 and 4.4), the impact of air mass origin (section 4.5) and synoptic events (section 4.6).

4.1 q–δ diagrams

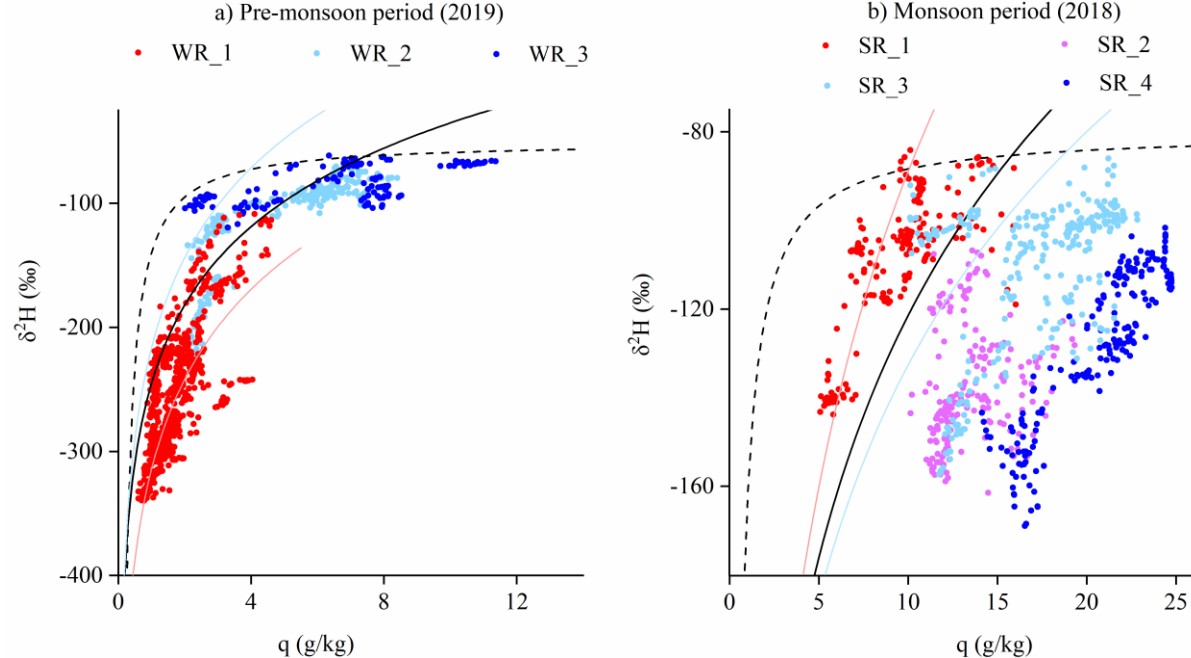

Fig.8 Scatterplot of observed vapor $\delta^2H$ (‰) versus specific humidity q (g/kg) during the pre-
monsoon period (a) and monsoon (b) period. The solid black curves show the Rayleigh
distillation line calculate for the initial conditions of $\delta^2H_0$ = -50‰, T=15 ℃ during the pre-
monsoon period and $\delta^2H_0$ = -80‰, T=25 ℃ during the monsoon period. The mixing lines
(dashed black curves) are calculated using a dry end-member with q = 0.2 g/kg and $\delta^2H$ = −
500 ‰ and air parcels for the corresponding Rayleigh curve as a wet end-member. The colored
solid curves show the uncertainty range of the Rayleigh curve, calculated for different initial
conditions of key moisture source regions: during March 2019, light red and light blue Rayleigh
curve are calculated for key moisture source regions of westerlies ( $\delta^2H_0$ = − 168.04‰, T=5℃)
and BoB ($\delta^2H_0$= -77.37‰, T=26.46℃) separately in (a); during July-August 2018, light red
and light blue Rayleigh curve are calculated for key moisture source regions of westerlies
($\delta^2H_0$= -149.64‰, T=6.16℃) and BoB ($\delta^2H_0$= -82.75‰, T=27.69℃) separately in (b). These
initial $\delta^2H$ are derived from Iso-GSM, the initial temperature and RH are derived from
NCAR/NCEP 2.5-deg global reanalysis data.

The progressive condensation of water vapor from an air parcel from the source region to
the sampling site and the subsequent removal of condensate results in a gradual reduction of
humidity and vapor isotope ratios. This relationship can be visualized in a q–δ diagram, which
has been used in many studies of the vapor isotopic composition (Noone, 2012;Galewsky et al.,
2016). Observations along the Rayleigh distillation line indicate progressive dehydration by
condensation. Observations above the Rayleigh line indicate either mixing between air masses
of contrasting humidity (Galewsky and Hurley, 2010) or evapotranspiration (Galewsky et al.,
2011;Samuels-Crow et al., 2015;Noone, 2012;Worden et al., 2007). Observations below the
Rayleigh line, even when considering the most depleted initial vapor conditions (light blue
Rayleigh curve in Fig 8b), indicate the influence of rain evaporation from depleted precipitation
(Noone, 2012;Worden et al., 2007). Figure 8 shows the observed vapor $q$–$\delta^2$H for different
regions during the pre-monsoon (a) and monsoon (b) period. This figure will be interpreted in
the light of meteorological variables along back-trajectories (Fig.2).
During the pre-monsoon period, most $q$-$\delta^2$H measurements are located surrounding or
overlapping the Rayleigh curve (the solid black curve in Fig.8a). Therefore, the observed spatial
pattern can mostly be explained by the gradual depletion of vapor isotopes by condensation.
The data for the three moisture sources are distributed in different positions of the Rayleigh
curve, relate to different moisture origins or different original vapor isotope values. This is
confirmed by the back-trajectory analysis: the Westerlies bring cold and dry air to northern
China (WR_1, Fig.3a, Fig.2a and c), consistent with the vapor further along the Rayleigh
distillation, and thus very depleted (Fig.5a). The observations in the WR_1 region (Fig.3c) are
closer to the q-$\delta^2$H Rayleigh distillation curve calculated for the key moisture source regions of
westerlies, providing further evidence of the influence of water vapor source on vapor isotopes.
The relatively high T and q along the ocean-sourced air trajectory reaching southern China
(WR_3, Fig.3c, Fig.2a and c) is consistent with an early Rayleigh distillation phase during
moisture transport, and thus higher water vapor $\delta^{18}$O in southern China (Fig.5a). Some
observations in the WR_3 region (Fig.3c) are located below the $q$-$\delta^2$H Rayleigh distillation
curve, indicating the influence of rain evaporation (Noone, 2012;Worden et al., 2007). This is
consistent with the fact that air originates from the BoB, where deep convection begins to be
active, and thus rain evaporation become a source of water vapor.
During the monsoon period, we find a scattered relationship in the $q$–$\delta^2$H diagram for
different regions, implying different moisture sources and/or water recycling patterns during
moisture transport. Data measured in the SR_1 region (Fig.3i) fall above the Rayleigh
distillation line (solid black curve in Fig.8b), likely due to the presence of moisture originating
from continental recycling. A larger number of $q$-$\delta^2$H measurements (most of the measurements
from the SR_2, SR_3, and SR_4 regions, Fig.3i) are located below the Rayleigh curve,
indicating moisture originating from the evaporation of rain drops within and below convective
systems (Noone, 2012;Worden et al., 2007). In SR_3 and SR_4 regions, this is consistent with
the high precipitation rate along Southerly and Easterly back-trajectories (Fig.2f). The
convection is active over the Bay of Bengal, Pacific Ocean and South-Eastern Asia, as shown
by the low OLR (<240W/m 2 ) in these regions (Fig.S3) (Wang and Xu, 1997). Therefore, a
significant fraction of the water vapor originates from the evaporation of rain drops in
convective systems. These results support recent studies showing that convective activity
depleted the vapor during transport by the Indian and East Asian monsoon flow (Cai et al.,
2018;He et al., 2015;Gao et al., 2013). In SR_2 region, the relatively low water vapor $\delta^{18}$O,
below the Rayleigh curve, is also probably associated with the evaporation of rain drop under
deep convective systems. This is confirmed by the high precipitation rates along Northerly
back-trajectories (Fig.2f), reflecting summer continental convection.
In northern China, q–$\delta$ diagrams show stronger distillation during the pre-monsoon period.
This suggests a "temperature dominated" control. Very low regional T during the pre-monsoon
period (Fig.S2 a and Fig.2a) are associated with low saturation vapor pressures and enhanced
distillation, producing lower vapor $\delta^{18}$O. The T in summer is higher (Fig.S2 b and Fig.2b),
allowing for higher vapor $\delta^{18}$O. The $\delta^{18}$O$_{monsoon}$ - $\delta^{18}$O$_{pre-monsoon}$ values in this region are therefore
positive (Fig.7a). In the South, q–$\delta$ diagrams suggest the stronger influence of rain evaporation
during the monsoon period. Higher precipitation amount significantly reduce $\delta^{18}O$ in the South
(Fig.2f), even though T was higher during the monsoon period than in pre-monsoon. This
suggests a "precipitation dominated" control in this region, explaining the negative values of
$\delta^{18}O_{monsoon}$ - $\delta^{18}O_{pre\text{-}monsoon}$. This seasonal pattern in $\delta^{18}O$ is consistent with the results in
precipitation isotopes (Araguás-Araguás et al., 1998;Wang and Wang, 2001). The boundary line
separating the seasonal variations of $\delta^{18}O$ is also consistent with previous study on seasonal
difference in vapor $\delta^2H$ retrieved by TES and GOSAT (Shi et al., 2020).
4.2 The $\delta^{18}O$-$\delta^2H$ relationship

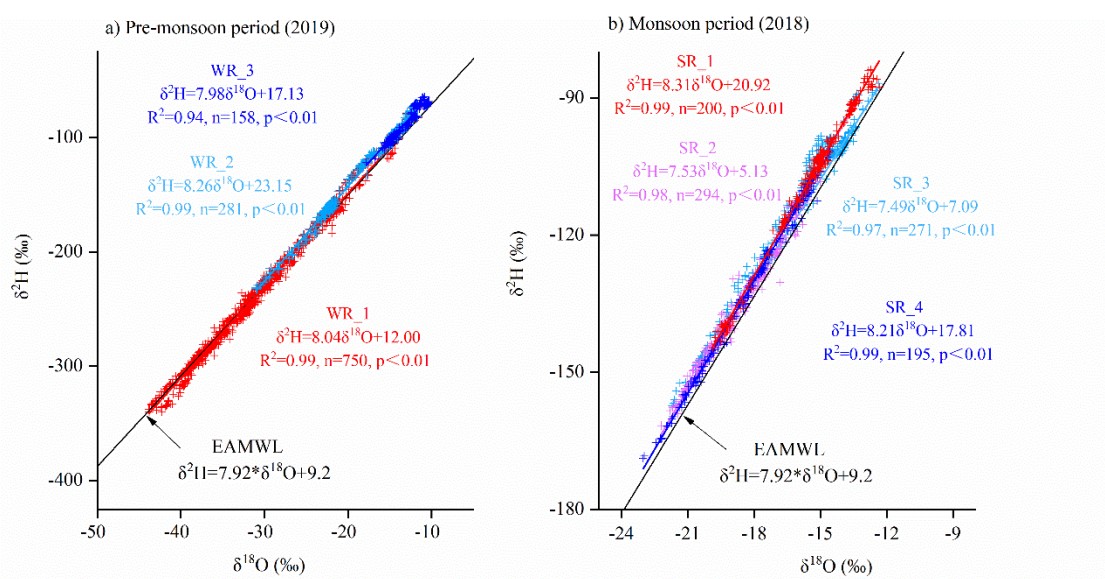

Fig.9 Regional patterns of vapor $\delta^{18}O$ - $\delta^2H$ relation during pre-monsoon period (a) and
monsoon (b) period, compared with the East Asia Meteoric Water Line (EAMWL) (Araguás-
Araguás et al., 1998).

The $\delta^{18}O$-$\delta^2H$ relationship is usually applied to diagnose the moisture source and water
cycling processes related to evaporation. Figure 9 and Table 1 show the $\delta^{18}O$-$\delta^2H$ relationship
for different regions in the two seasons. We also plot the East Asian Meteoric Water Line
(EAMWL) for a reference. Vapor $\delta^{18}O$-$\delta^2H$ is usually located above Meteoric Water Line owing
to the liquid water and vapor fractionation.
During the pre-monsoon period (Fig.9a), the data in northern China (WR_1, Fig.3a) are
located at the lower-left area in the $\delta^{18}O$-$\delta^2H$ graph, with similar slope and intercept as EAMWL
($\delta^2H = 8.04\ \delta^{18}O + 12.00$). This corresponds to air brought by the Westerlies and following
Rayleigh distillation. The linear relationship for the vapor in middle China (WR_2, Fig.3b) has
the steepest slope and highest intercept ($\delta^2H = 8.26\delta^{18}O + 23.15$). These properties are
associated with a high d-excess, consistent with strong continental recycling by
evapotranspiration (Aemisegger et al., 2014). As continental recycling is known to enrich the
water vapor (Salati et al., 1979) and is associated with high d-excess (Gat and Matsui,
1991;Winnick et al., 2014). The high intercept is further consistent with a correlation between
$\delta^{18}O$ and d-excess, which can typically result from continental recycling (Putman et al., 2019).
The data for vapor originating from the BoB (WR_3, Fig.3c) are located to the upper right of
the EAMWL. Their regression correlation shows similar features ($\delta^2H = 7.98\ \delta^{18}O + 17.13$) to

that of the monsoon season (with a slope of 8.21 and an intercept of 17.81). We find similar atmospheric conditions in the BoB (with the region marked as rectangle in Fig.3c and h) during the two observation periods, with T=26°C and RH=76% during pre-monsoon period and T=28°C and RH=78% during the monsoon period, suggesting that the BoB source may have similar signals on vapor $\delta^{18}O$ and $\delta^2H$ in both seasons. These observed vapor $\delta^{18}O$-$\delta^2H$ patterns are consistent with the back-trajectory results indicating that the Westerlies persist in northern China during the pre-monsoon period, while moisture from the BoB has already reached southern China.

During the monsoon period (Fig.9b), the data in northwestern China (SR_1, Fig.3e) with continental moisture sources is located in the upper right of the graph but above the EAMWL, with the steepest slope and highest intercept for the linear $\delta^{18}O$-$\delta^2H$ relationship ($\delta^2H$ = 8.31$\delta^{18}O$ +20.92). In contrast, the observations in southeastern China with BoB sources (SR_4, Fig.3h) are located in the lower left of the graph, with relatively lower intercept ($\delta^2H$ = 8.21$\delta^{18}O$ +17.81). This is the opposite pattern compared to the pre-monsoon season. The observations from the SR_3 region (Fig.3g) also have a low slope and low intercept ($\delta^2H$ = 7.49 $\delta^{18}O$ +7.09). This is consistent with the oceanic moisture from the Pacific Ocean. Also, these $\delta^{18}O$-$\delta^2H$ data are located in the upper right of the graph with more scattered relation (with the lowest correlation coefficient), suggesting more diverse moisture sources. This is consistent with the mixing of water vapor from continental recycling and Pacific Ocean (Fig.3g). The observations in northeastern China (SR_2, Fig.3f) are located at the lower left of the graph, suggesting the influence of condensation along trajectories in northern Asia (Fig.2f). Compared to the SR_3 and SR_4 regions, the slope and intercept of the observations in SR_2 region are lower ($\delta^2H$ = 7.53$\delta^{18}O$ +5.13), reflecting different origins of moisture.

4.3 Relationship with local meteorological variables

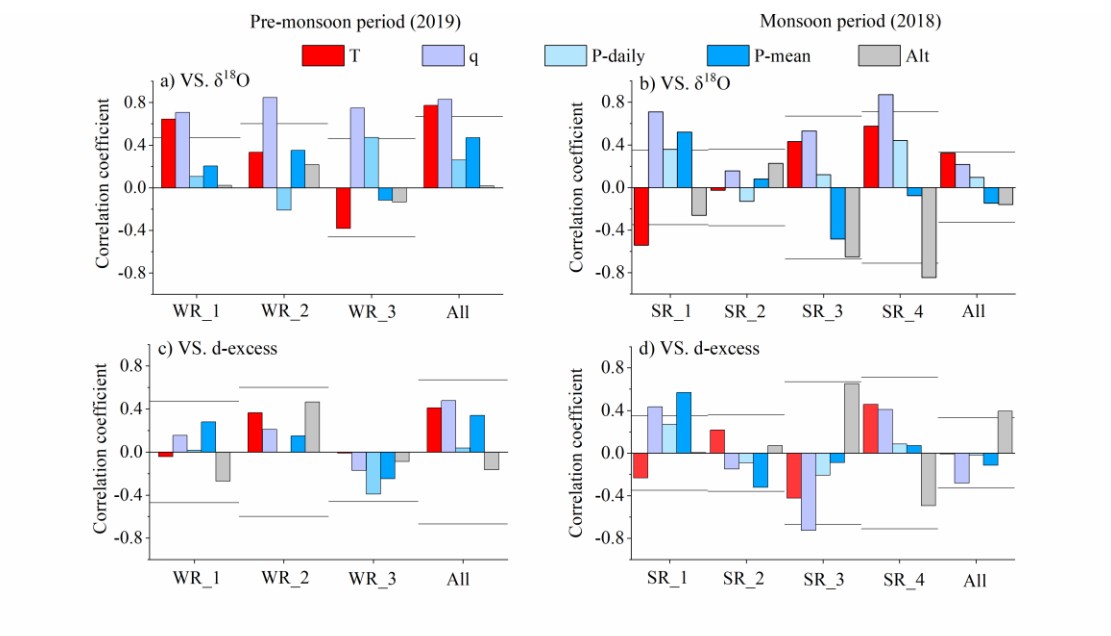

Fig.10 Regional patterns of the correlation between $\delta^{18}O$ (a, b), d-excess (c, d) and various local factors (temperature (T), specific humidity (q), daily precipitation amount (P-daily) and average precipitation amount over the entire observation period for each observation location   (P-

mean), and altitude (Alt)). The left panel is for the pre-monsoon period and the right is for the monsoon period. Horizontal lines indicate the correlation threshold for statistical significance (p<0.05), considered the degree of freedom.

Here we analyze the relationship between vapor $\delta^{18}$O, d-excess and local meteorological parameters, for all observations, and separately for the different regions (Fig.10 and Table S1).

We have taken particular care to estimate the statistical significance of the correlation coefficients. The statistical significance of a correlation depends on the correlation coefficient and on the degree of freedom D of the observed $\delta^{18}$O and d-excess time series. Since these variables evolve smoothly in time and are sampled at a high frequency, the total number of samples overestimates the degree of freedom D of the time series. We thus estimated the degree of freedom D as $\text{T}/\eta$, where $\text{T}$ is the length of the sampling period and $\eta$ is the characteristic auto-correlation time scale of the time series (an example of this calculation is given in III. Supplementary text). A similar method was used to calculate the degree of freedom of the signal in (Roca et al., 2010). Table S2 summarizes the threshold for the correlation coefficient to be statistically significant at 95%, for the two seasons, the different regions and the variable of interest.

During the pre-monsoon period, all observations taken together exhibit a "temperature effect" (the $\delta$'s decreasing with temperature, Dansgaard 1964 ) (Fig.10a), with significant and positive correlation between $\delta^{18}$O and T (r = 0.77, p<0.05, Table S1). This results from the high correlation between $\delta^{18}$O and q (r = 0.83, p<0.05, Table S1), consistent with the Rayleigh distillation, and between T and q (r = 0.54, p<0.05), consistent with the Clausius Clapeyron relationship. The vapor $\delta^{18}$O in the WR_1 (Fig.3a) region show similar correlations with T and q as for all observations. Rayleigh distillation thus contributes to the relationship between $\delta^{18}$O and T observed in northern China. In contrast, no significant positive correlation between vapor $\delta^{18}$O and T is observed in the WR_3 region with the BoB water source. This is consistent with the fact that the moisture from the BoB has already influenced southern China during the pre-monsoon period (Fig.3c). The weak positive correlation in most regions between $\delta^{18}$O and P-daily and P-mean might simply reflect the control of q on observed vapor $\delta^{18}$O, due to the relatively high correlation between observed P-mean and q, with r = 0.58 for all observations (Fig.4).

During the monsoon period (Fig 10b), no significant correlation emerges when considering all observations. Vapor $\delta^{18}$O is still significantly correlated with q in the SR_1 (Fig.3e, r = 0.71, p<0.05, Table S1) and SR_4 (Fig.3h, r = 0.87, p<0.05, Table S1) regions. This is consistent with different degree of rain-out. This may reflect the synoptic-scale variations of convection. The absence of correlation with T suggests that the variations in q mainly reflect variations in relative humidity that are associated with different air mass origins or rain evaporation. The $\delta^{18}$O is significantly anti-correlated with Alt in the SR_4 region (r = -0.85, p<0.05, Table S1), consistent with the "altitude effect" (the heavy isotope concentrations in fresh water decreasing with increasing altitude) in precipitation and water vapor (Dansgaard, 1964;Galewsky et al., 2016).

The vapor d-excess for all observations during the monsoon period (Fig.10d) is positively correlated with Alt (r = 0.39, p<0.05, Table S1). One possible reason is that the vapor d-excess is lower in coastal areas at lower altitudes, while at higher altitudes in the west, more continental

recycling of moisture leads to higher d-excess(Aemisegger et al., 2014). The positive
correlation between d-excess and altitude is consistent with previous studies in region (Acharya
et al., 2020). In the SR_1 region (Fig.3e), in arid northwestern China, vapor d-excess is
positively correlated with q (r = 0.43, p<0.05, Table S1) and P-mean (r = 0.57, p<0.05, Table
S1) during the monsoon period, suggesting that rain evaporation may also contribute to high d-
excess (Kong and Pang, 2016). Other than these examples, the correlation coefficients between
the d-excess and T, q, P, and Alt are not significant (Fig.10c and d), indicating that the local
meteorological variables are not strongly related to vapor d-excess, as was reported in previous
studies for precipitation isotopes (Guo et al., 2017;Tian et al., 2003).
4.4 Relationship with meteorological variables along trajectories

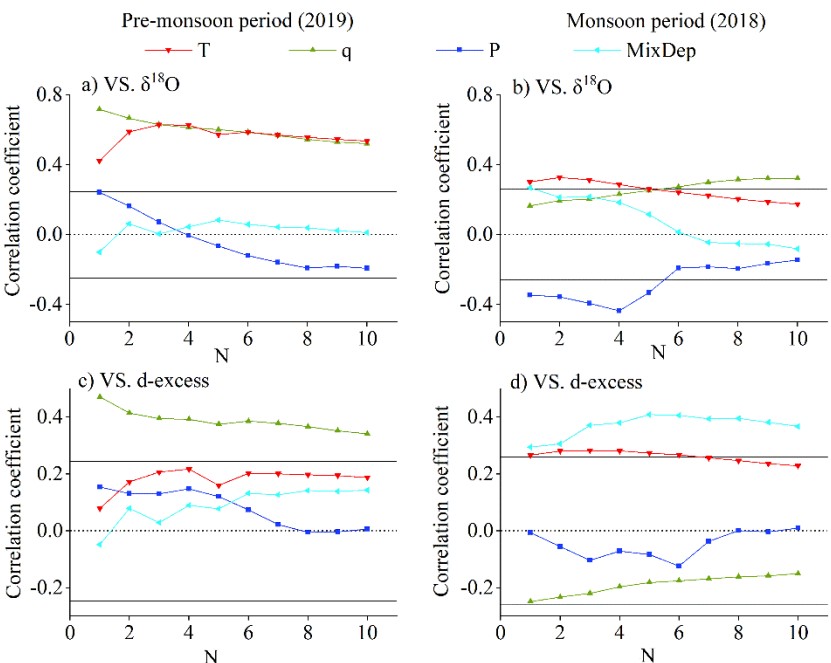

Fig.11 Correlation between $\delta^{18}$O (a, b), d-excess (c, d)), and various meteorological factors (air
temperature (T), specific humidity (q), precipitation (P), and mixing depth (MixDep)) along the
air mass trajectories during the pre-monsoon period (the left panel) and monsoon period (the
right panel). The x-axis "N" represents the number of days prior to the observations (from 1
to 10 days). For example, when the number of days is 2, the correlations is calculated with the
temporal mean of meteorological data along the air mass trajectories during the 2days before
the observations. Horizontal solid lines indicate the correlation threshold for statistical
significance (p<0.05).
Reconstructions of paleoclimates using ice core isotopes have relied on relationships with
local temperatures, but many previous studies have suggested that water isotopes are driven by
remote processes along air mass trajectories. In particular, they emphasized the importance of
upstream convection in controlling the isotopic composition of water (Gao et al., 2013;He et
al., 2015;Vimeux et al., 2005, Cai and Tian, 2016). We therefore perform a correlation analysis
between vapor isotope observations and the temporal mean meteorological conditions along air
mass trajectories. The meteorological conditions are averaged over the previous days (N form
1 to 10) prior to the observations.

The $\delta^{18}$O values have the strongest correlations with T and q along air mass trajectories
during the pre-monsoon period (Fig.11a). The results show gradually increasing positive
correlation coefficients as N changes from 10 to 3. This reflects the role of temperature and
humidity along air mass trajectories and the large spatial and temporal coherence of T variations
during the pre-monsoon period. During the monsoon period, the negative correlation
coefficients between $\delta^{18}$O and P (Fig.11b) become more significant as $N$ increases from 1 to 4
and less significant as $N$ increases from 5 to 10. This result indicates a maximum impact of P
during a few days prior to the observations, as observed also for precipitation isotopes (Gao et
al., 2013;Risi et al., 2008a). It is further consistent with the influence of precipitation along
back-trajectories (Fig.2f). Mixing depth (MixDep) is stably and positively correlated with d-
excess. A hypothesis to explain this correlation is that when the MixDep is higher, stronger
vertical mixing of convective system transports vapor with higher d-excess values from higher
altitude to the surface (Galewsky et al., 2016;Salmon et al., 2019).
4.5 Relationship between water vapor isotopes and moisture sources

In section 4.1 to 4.4, we have discussed that different moisture sources and corresponding
processes on transport pathways are related to the observed spatial patterns both in vapor $\delta^{18}$O
and d-excess.

We also identify different isotopic values of vapor from different ocean sources during the
monsoon period. The vapor $\delta^{18}$O in the zone from Beijing to Harbin and western China with
Pacific Ocean and continental origins (SR_3 region, about -17‰ to -13‰) are higher than those
in the Southeast with BoB sources (SR_4 region, about -23‰ to -15‰) (Fig.3i and Fig.5b). In
sections 4.1 and 4.2, we have shown that it is related to the extent of the Rayleigh distillation
and rain evaporation associated with convection along trajectories. Earlier studies suggest that
lower $\delta^{18}$O values were observed from the Indian monsoon source than from Pacific Asian
monsoon moisture due to the different original isotope values in the source regions (Araguás-
Araguás et al., 1998). To better isolate the direct effect of moisture sources, we extract the initial
vapor isotopes of the Indian and East Asian monsoon systems (the regions are marked as
annotated rectangles in Fig.3g and h) for the sampling dates of 2018 from the Iso-GSM model.
The values are about $\delta^{18}$O=-12‰ and $\delta^2$H =-83‰ in the northern BoB and $\delta^{18}$O =-14‰ and
$\delta^2$H =-97‰ in the eastern Pacific Ocean. The initial vapor isotope values of the two vapor
sources are not significantly different. The initial vapor isotopes in the BoB are even slightly
higher than those in the Pacific Ocean, contrary to moisture source hypothesis. The OLR was
significantly lower in the BoB than in the Pacific Ocean (Fig.S3). This suggests that the deeper
convection in the Indian Ocean leads to lower water vapor isotope ratios (Liebmann and Smith,
1996;Bony et al., 2008;Risi et al., 2008b;Risi et al., 2008a) in southeastern China, rather than
the initial composition of the moisture source.

Continental recycling probably also contribute to higher $\delta^{18}$O in the SR_3 region (Fig.3i
and Fig.5b) (Salati et al., 1979), especially in western China (Fig.3i), which can be confirmed
by the higher d-excess in this region (Fig.5d) (Gat and Matsui, 1991;Winnick et al., 2014).
Except SR_3 region, continental recycling also has a strong influence on isotopes in the WR2
and SR1 regions, which suggested by the high values of $\delta^{18}$O and d-excess, back-trajectories,
the location on the q-$\delta$ diagram, and the higher slopes and intercepts of $\delta^{18}$O-$\delta^2$H relationship.
In the opposite, in the zone from Beijing to Harbin (Fig.3i), greater proportion of water vapor
from Pacific sources than continental recycling and is in the early stage of Rayleigh distillation,
could result in high vapor $\delta^{18}O$ (Fig.5b) but relatively low d-excess (Fig.5d).
In previous studies, the d-excess has been interpreted as reflecting the moisture source and
evaporation conditions (Jouzel et al., 1997). During the pre-monsoon period, lower T and higher
RH over evaporative regions for the vapor transported by the Westerlies (Fig.2a and g, Fig.S2
a and g) reduces the non-equilibrium fractionation at the moisture source and produces lower
vapor d-excess in the WR_1 region (Fig.3a, Fig.5c) (Jouzel et al., 1997;Merlivat and Jouzel,
1979). In contrast, higher T and lower RH over evaporative regions (Fig.2 a and g, Fig.S2 a and
g) for the vapor coming from the South leads to higher d-excess in southern China (WR_3,
Fig.3c, Fig.5c). This is consistent with the global-scale poleward decrease in T and increase in
surface RH over the oceans (despite the occurrence of very low RH at the sea ice edge during
cold air outbreaks (Thurnherr et al., 2020;Aemisegger and Papritz, 2018)), resulting in global-
scale poleward decrease in d-excess at mid-latitudes (Risi et al., 2013a;Bowen and Revenaugh,
2003). Alternatively, the low d-excess during the night over the continent in Northern China
during the pre-monsoon could also have contributions (Li et al., 2021). During the monsoon
period, the lower vapor d-excess observed in eastern China (Fig.5d) is likely a sign of the
oceanic moisture, derived from source regions where RH at the surface is high (Fig.2h and
Fig.S2 h) and thus reduce non-equilibrium fractionation and lower d-excess. The high d-excess
values observed in western and northwestern China (Fig.5d) reflect the influence of continental
recycling (Fig.3e and g).
The seasonal variation of moisture sources also results in a seasonal difference in d-excess
(Fig.8b). In southeastern China, RH over the ocean surface in summer is higher than in winter
(Fig.S2 g and h, and Fig.2g and h ), resulting in negative values of d-excess$_{monsoon}$ - d-excess$_{pre-}$
$_{monsoon}$ (Fig.8b). Northwestern China has an opposite pattern of seasonal vapor d-excess. This
result largely due to the extremely low vapor d-excess during the pre-monsoon period (Fig.5c).
Also, we speculate that a greater contribution of continental recycling leads to higher d-excess
during the monsoon period than during the pre-monsoon period (Risi et al., 2013b) and the
positive values of the d-excess$_{monsoon}$ - d-excess$_{pre-monsoon}$ (Fig.8b).
4.6 Possible reasons for the biases in Iso-GSM
In section 3.3, we showed that Iso-GSM captured the isotopic variations during the pre-
monsoon season better than during the monsoon season. We hypothesize that this mainly could
be due to the larger contribution of synoptic-scale variations to the observed variations during
the monsoon season. Iso-GSM performs well during the pre-monsoon season, when seasonal
mean spatial variability dominates q and isotope. In contrast, it performs less well during the
monsoon season, when isotopic variations are significantly influenced by the synoptic-scale
variability. Among the synoptic influences, tropical cyclones, the Northern Summer Intra-
Seasonal Oscillation (BSISO) and local processes probably played a role. For example, during
our monsoon observations, landfall of tropical cyclones Jongdari and Yagi correspond to the
low values of $\delta^{18}O$ we observed in the eastern China (Fig.S7a). Bebinca corresponds to the low
values of $\delta^{18}O$ we observed in the southwestern China (Fig.S7a). Typhoons are known to be
associated with depleted rain and vapor (Bhattacharya et al., 2022;Gedzelman, 2003). Three
Northern Summer Intra-Seasonal Oscillation (BSISO) events occurred in China during about
July 28 - 31, August 5 - 8[th] and August 14 – 16 (Fig S8). The northward propagation of the
BSISO is associated with strong convection (Kikuchi, 2021) (Fig. S8). Moreover, short-lived
convective events that frequently occurred during our observation period (Wang et al., 2018).
It is possible that these rapid high-frequency synoptic events are not fully captured by Iso-GSM.
We expect that Iso-GSM captures the large-scale circulation. Yet, we notice that Iso-GSM
underestimates the depletion associated with tropical cyclones (Fig. 12b). We hypothesize that
given its coarse resolution, it underestimates the depletion associated with the meso-scale
structure. This might contribute to the overestimation of vapor $\delta^{18}O$ in southeastern China
(Fig.12b). In Northwestern China, Iso-GSM underestimates vapor $\delta^{18}O$, but also underestimates
precipitation, q and T (Fig.12b, d, f and h, Fig. S4). It is possible that Iso-GSM underestimates
the latitudinal extent of the monsoonal influence, which brings moist conditions, while
overestimating the influence of continental air, bringing dry conditions associated with depleted
vapor through Rayleigh distillation. It is also possible that Iso-GSM underestimates the
enriching effect of continental recycling. During the pre-monsoon period, Iso-GSM
overestimates the observed $\delta^{18}O$ along most of the survey route (Fig.12a), with the largest
difference in northwestern China, and underestimates the vapor $\delta^{18}O$ in the southern part of the
study region. Our results are consistent with previous studies showing that many models
underestimate the heavy isotope depletion in pre-monsoon seasons in subtropical and mid-
latitudes, especially in very dry regions (Risi et al., 2012). This was interpreted as overestimated
vertical mixing. The differences in $\delta^{18}O$ (Fig.12a) and q (Fig.12c) are spatially consistent. The
overestimation of $\delta^{18}O$ therefore could be due to the overestimation of q, and vice versa. These
biases could be associated with shortcomings in the representation of convection or in
continental recycling. Despite this, the good agreement during pre-monsoon period is probably
due to the dominant control by Rayleigh distillation on seasonal-mean spatial variations of
isotopes in this season, as concluded in the above. The q variation, in relation with T, drives
vapor isotope variations and is well captured by Iso-GSM spatially, with significant correlations
between observed and simulated q ( r = 0.84, slope=0.70 in Table S3) and T ( r = 0.87,
slope=0.70 in Table S3), though q is overestimated in the North and underestimated in the South.

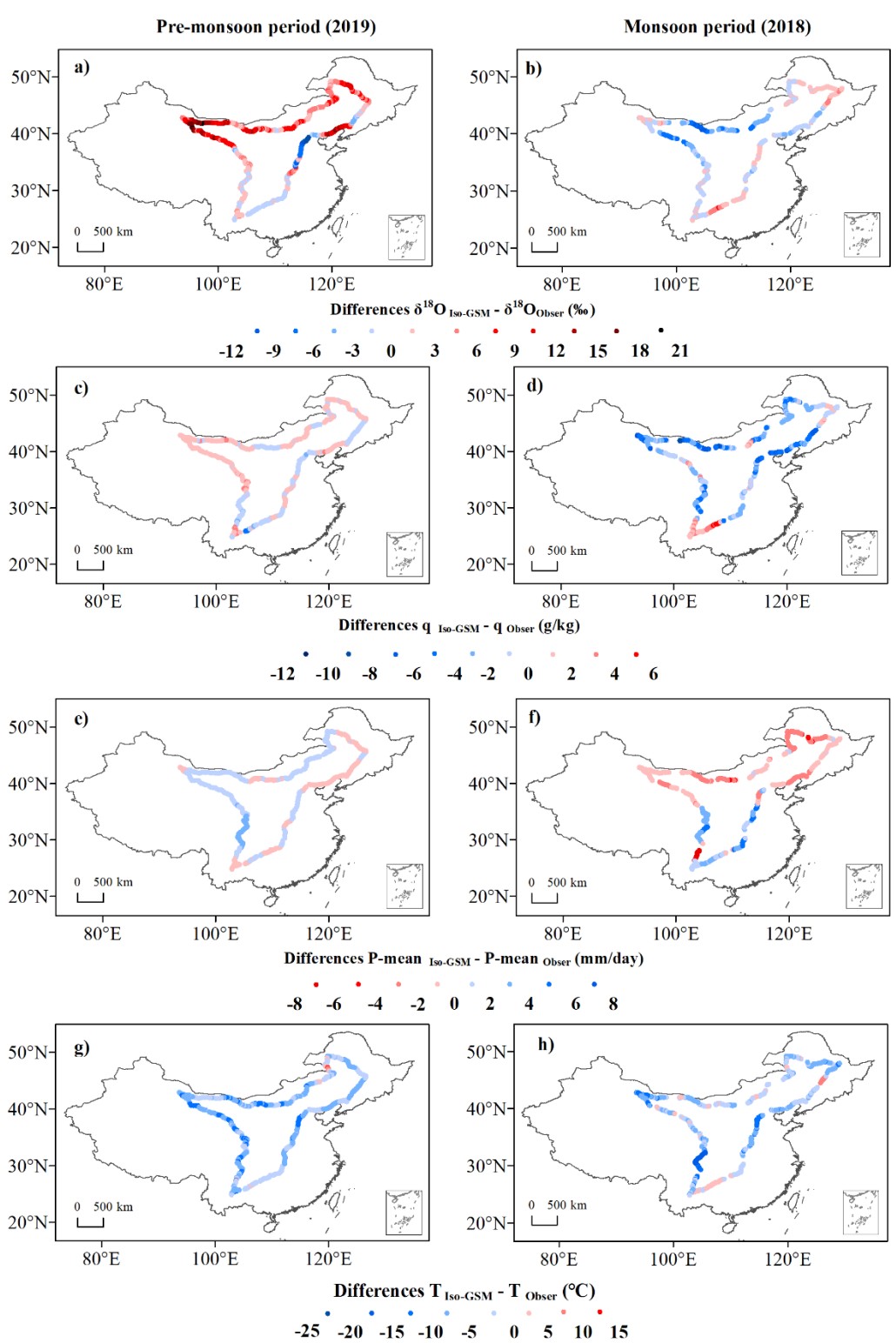


Fig.12 Spatial distribution of the differences between the outputs of Iso-GSM (subscripts are Iso-GSM) and observations (subscripts are Obser) during the pre-monsoon period (the left panel) and monsoon period (the right panel): $\delta^{18}O$ (a and b, ‰), specific humidity q (c and d, g/kg), P-mean for the sampling dates (e and f, mm/day), and temperature T (g and h, °C).

## 5 Conclusion

Our new, vehicle-based observations document spatial and seasonal variability in surface water vapor isotopic composition across a large part of China. Both during the pre-monsoon and monsoon periods, it is clear that different moisture sources and corresponding processes on transport pathways explain the spatial patterns both in vapor $\delta^{18}O$ and d-excess (summarized in Fig.13):

(1) During the pre-monsoon period (Fig.13a), the latitudinal gradient of vapor $\delta^{18}O$ and d-excess were observed. The gradient in $\delta^{18}O$ reflects the "temperature effect", Rayleigh distillation appears to be the dominant control, roughly consistent with earlier studies on precipitation. Vapor in northern China, derived from westerlies, and subject to stronger Rayleigh distillation (arrows fading from red to blue), is characterized by very low isotope ratios (blue shades). Less complete Rayleigh distillation (arrows fading from red to light red) results in less depleted vapor in southern China (light red shades). The vapor d-excess in northern China is low (green triangles series), probably due to the high RH over high-latitude oceanic moisture sources for the vapor transported by the Westerlies (green arrow), reducing the kinetic fractionation during ocean evaporation. In contrast, the lower RH over low-latitude moisture sources (yellow arrow) for the vapor transported to southern China leads to higher d-excess (yellow triangles series). Additional vapor sourced from continental recycling (orange twisted arrows), further increases the d-excess values in middle China. This distribution is consistent with the back-trajectory results showing that during the pre-monsoon period, the vapor in southwestern China comes from the BoB, whereas Westerly moisture sources still persist in northern China.

(2) During the monsoon period (Fig.13b), the lowest vapor $\delta^{18}O$ occurred in southwestern and northeastern China, and higher vapor $\delta^{18}O$ values were observed in between, while the d-excess features a west-east contrast. The relatively lower vapor $\delta^{18}O$ result from deep convection along the moisture transport pathway (blue clouds; arrows fading to blue). Meanwhile, the mixing with moisture from continental recycling (orange twisted arrows) increases the vapor $\delta^{18}O$ values in middle and northwestern China. We observed lower vapor $\delta^{18}O$ values when the moisture originates from the BoB than from the Pacific Ocean, consistent with stronger convection during transport. The dominance of oceanic-wet moisture (green arrows) results in the lower vapor d-excess (green triangles series) in eastern China, whereas continental recycling produces higher vapor d-excess in western and northwestern China (yellow triangles series).

(3) Variation in temperature drive the seasonal variations of vapor $\delta^{18}O$ in northern China, whereas convective activity along trajectories produces low vapor $\delta^{18}O$ curing the monsoon season and drive the seasonal variation in south China. Seasonal d-excess variation reflects different conditions in the sources of vapor: in southeastern China it is mainly due to differences in the RH over the adjacent ocean surface, while in northwestern China it is mainly due to the vapor transported by the Westerlies during the pre-monsoon period and a great contribution of continental recycling during the monsoon period.

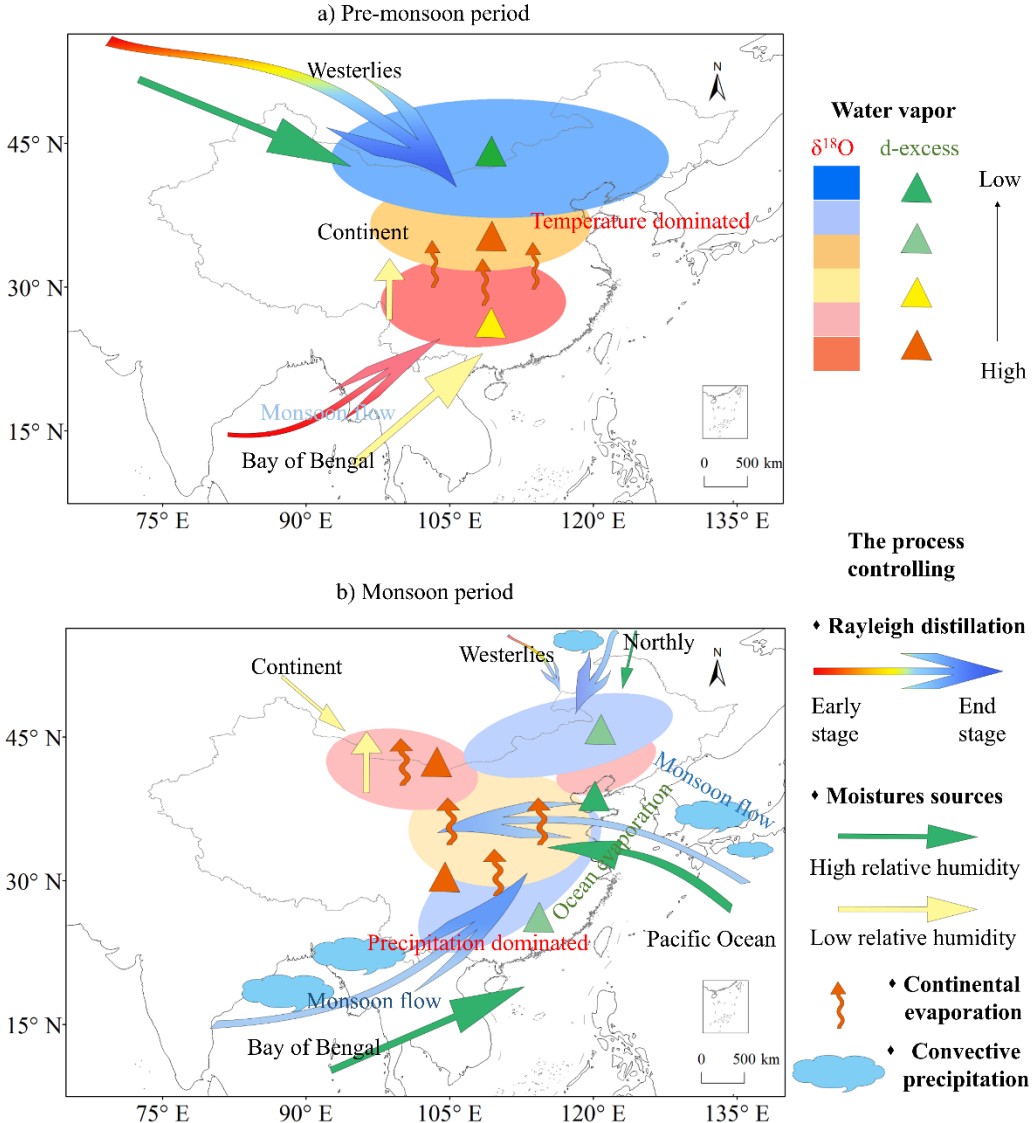


Fig.13 Schematic picture summarizing the different processes controlling the observed spatial
patterns and seasonality of vapor isotopes. Color gradient arrows from red to blue represent the
initial to subsequent extension of the Rayleigh distillation process along the water vapor
trajectory, corresponding to high to low values of δ18O; green arrows represent high relative
humidity and yellow arrows represent low relative humidity; orange twisted arrows represent
continental recycling; blue-sized clouds represent strong and weak convective processes; green
triangles series representing low values of d-excess; yellow triangles series representing high
values of d-excess.

Iso-GSM simulations and IASI satellite measurements indicate that during the pre-
monsoon period, the observed temporal variations along the route across China are mainly due
to multi-year seasonal-mean-spatial variations, and marginally due to synoptic-scale variations.
During the monsoon season, synoptic-scale and intra-seasonal variations might contribute
significantly to the apparent spatial patterns. However, since these variations have a smaller
amplitude than seasonal differences, the comparison of the two snapshots do provide a
representative picture of the climatological seasonal difference.

Our study on the processes governing water vapor isotopic composition at the regional
scale provides an overview of the spatial distribution and seasonal variability of water isotopes
and their controlling factors, providing an improved framework for interpreting the
paleoclimate proxy records of the hydrological cycle in low and mid-latitudes. In particular, our
results suggest a strong interaction between local factors and circulation, emphasizing the need
to interpret proxy records in the context of the regional system. This also suggests the potential
for changes in circulation to confound interpretations of proxy data.

**Data availability**

The data acquired during the field campaigns used can be accessed via the following link or
DOI: (1) Wang, Di; Tian, Lide (2022): Vehicle-based in-situ observations of the water vapor
isotopic composition across China during the monsoon season 2018. PANGAEA,
https://doi.org/10.1594/PANGAEA.947606; (2)Wang, Di; Tian, Lide (2022): Vehicle-based in-
situ observations of the water vapor isotopic composition across China during the pre-monsoon
season 2019. PANGAEA, https://doi.org/10.1594/PANGAEA.947627. . Other data used can be
downloaded from the corresponding website which were listed in the text.

**Author contributions**

L.T. and D.W. designed the research; D.W., and X.J. conducted to the field observations; J.C.
and J. B. contribute to the data calibration; Z.W. and K.Y performed Iso-GCM simulations;
D.W., C.R., and L.T. performed analysis; All authors contributed to the discussion of the results
and the final article; D.W. drafted the manuscript with contributions from all co-authors; C.R.,
L.T. and J. B. checked and modified the manuscript.

**Competing interests**

The authors declare that they have no conflict of interest.

**Acknowledgments**

The authors gratefully acknowledge NCAR/NCEP, GPCP and NOAA for provision of
regional and large-scale meteorological data. We are grateful to the NOAA Air Resources
Laboratory (ARL) that provided the HYSPLIT transport and dispersion model (http :/
/ready.arl .noaa .gov /HYSPLIT.php) and the HYSPLIT-compatible meteorological dataset
from GDAS. We thankfully acknowledge Yao Zhang, Xiaowen Zeng, Min Gan for technical
assistance. We thank Mingxing Tang and Ruoqun Zhang for partly participating in the field
observations. We thank to Zhaowei Jing for the discussions on Rayleigh distillation lines, and
thankfully acknowledge Yao Li, Zhongyin Cai and Rong Jiao for sharing some methods to use
Hysplit4 model. This work was supported by the Strategic Priority Research Program of
Chinese Academy of Sciences, Grant No. XDB40000000, the National Natural Science
Foundation of China (Grant 41771043, and 41701080), and Research Innovation Project for
Graduate Students of Yunnan University (Grant 2018Z098 and 2021Y040). Di Wang was
supported by the c.

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
