# Peer review of "Vehicle-based in-situ observations of the water vapor isotopic composition across China: spatial and seasonal distributions and controls"

_Atmospheric Chemistry and Physics, 2022_

## Referee Comment (RC1)

Review of "Vehicle-based in-situ observations of the water vapour isotopic composition across China: spatial and seasonal distributions and controls" by Di Wang et al. submitted to ACP

This paper presents a very interesting and impressive dataset of vehicle-based stable water isotope measurements in China with several new results and interpretations about the drivers of the variability observed in different regions. The observations cover two seasons: the pre-monsoon season 2019 and the monsoon season 2018, which are compared in terms of their different dynamics and isotope signals recorded along the route. I recommend publication of the paper after two major and several minor comments have been addressed:

**Major comments:**
A) **Synoptic vs. seasonal variations**: This analysis is very interesting.
   - However, it comes very late in the manuscript, even though the whole interpretation of the observations evolves along the main finding that the spatial (seasonal) variations dominate the variability of various isotope and meteorological variables. The results section even starts with referring to this analysis but asks the readers to postpone their curiosity to a much later section. Why not starting the results with what you show in Section 4.7?
   - What happens if you do the same regression analysis as presented in Section 4.7 using the actual observations for estimating the synoptic and seasonal components? Looking at the comparison of the simulation and the observations (Fig. 11), it strikes me that the daily model output follows the "temporal-mean" output much more closely than the actual measurements in both the pre-monsoon and the monsoon seasons. To me this shows that the Iso-GSM simulation on a relatively coarse grid is not capable of reproducing the observed mesoscale to large-scale variability in the water vapour isotope and meteorological fields. Therefore, in my view the finding that the isotope variability across China in the pre-monsoon and monsoon periods is mainly a result of seasonal/spatial variations and only marginally affected by synoptic-scale systems is biased towards what the Iso-GSM shows.
   - What exactly is the "temporal-mean" output? A multi-year seasonal mean? Or the mean over the considered 2018 and 2019 seasons? And over which spatial scale did you average?
   - I am not convinced that you can conduct this analysis in a robust way, given the few observations you have for a given region and season. A discussion on this sampling issue of the seasonal signal from only few synoptic events would be beneficial. Still, it is a valuable analysis especially if it is done with the measurements **and** the model, comparing and discussing the two results.

B) **Organization of the manuscript**: The reading of the manuscript would profit from a re-organization with a clearer separation of methodological aspects and results section. Many methodological aspects are in the results and interrupt the flow of the reader (regional analysis in Table 1, Urban emissions, methodological aspects on Eq. 4)

**Minor comments:**
1) L. 25: "large-scale (order 10000 km) continuous observations of near-surface vapor isotopes": can be misunderstood, reformulate. You made in-situ observations over a large area. Not continuous observations at multiple locations.
2) L. 28-29: "mainly due to spatial variations and marginally influenced by synoptic-scale variations": This is interesting! I was very curious about how you came to this conclusion, when starting to read your manuscript. When just reading the abstract, I found this statement and the following ones about the importance of Rayleigh distillation (cloud formation during large-scale ascent?), different moisture sources (variability induced by large-scale weather systems?), continental moisture recycling (transport regimes favouring oceanic vs. continental sources) and convection (mesoscale circulation) contradicting. First you write that the "spatial variations" dominate and then you mention different processes that are relevant at the synoptic- to meso- scale. The synoptic systems are very different in southern vs. northern China and these weather systems shape the spatial contrasts. It would be helpful

if you could concisely state in the abstract how you come to the conclusion that "seasonal"/"spatial" variations dominate the synoptic variability and think carefully about the best terminology to use.

3) L. 38-39: Why is the performance of the Iso-GSM model weaker over the monsoon period?
4) L. 46: The first sentence of the introduction is a bit heavy, could you think of a more general motivation for your study? And think about how to guide a non-isotope specialist reader into your subject.
5) L. 54: Are only Tibetan plateau ice cores questioned in terms of their use as past temperature records?
6) L. 56: "significant role of large-scale"
7) L. 57: is a specific teleconnection mode meant here?
8) L. 74-77: Isn't the Bailey et al. 2013 study from an oceanic environment? Aemisegger et al. 2014 ACP discusses the isotope variability of continental evaporation induced by synoptic-scale variability.
9) L. 97: Dry intrusions bring… cold and dry **upper tropospheric** airmasses
10) Fig. 1: the route is a bit difficult to make out from the otherwise very nice figure
11) L. 122: is computed
12) L. 164: no drift in the standards: could you indicate the standard deviation of your series of standard measurements?
13) Section 2.2.3: should be humidity-dependent isotope bias correction
14) Was the specific humidity calibrated as well using an independent sensor?
15) L. 212: why was only one air parcel started from 1000 m above the surface?
16) L. 229-234: This is a bit a difficult start of the results section. Could the analysis of the seasonal vs. synoptic drivers of isotope variability not come as a first result? It is otherwise very hard for the reader to follow the story.
17) L. 230: refer to Fig. 2a,b
18) L. 269: that could be the continental effect too… Should Dansgaard et al. 1964 be cited if you discuss the isotope effects? These different effects are not mutually exclusive, and in my opinion, they do not provide a direct mechanistic explanation for the correlation of the delta-values with temperature.
19) Section 3.2: I like the result about the fact that the monsoon counteracts the N-S gradient observed in the pre-Monsoon period very much. This could be included as a key result in the abstract.
20) L. 310-311: I don't understand this sentence. What is the separation line of the seasonal variation of d-excess? Reformulate.
21) P. 11-12: this is a methodological part and should not be in the results section.
22) L. 439: bring -> brought
23) L. 442: replace "strong" by "high". Actually, the fraction of transpiration in evapotranspiration is the key factor determining $d$ over the continent, when recycling is high (see Aemisegger et al. 2014, ACP).
24) L. 369: reduction of humidity and water isotope ratios.
25) Figure 6 is very interesting. Could it come earlier?
26) L. 391: this sounds a bit speculative: couldn't it be another Rayleigh line or mixing?
27) L. 493: "all observations taken together" because WR2 and WR3 do not show a "temperature effect".
28) L. 501: influence**d**
29) L. 509: degree of rain out
30) L. 517: why does the dexcess usually increase with altitude? Many recent studies show that it is more complicated than that (see, Salmon et al. 2019 ACP, Thurnherr et al. 2021 WCD). In many cases a decrease of dexcess towards the mid troposphere is observed.
31) Fig. 9: I am not sure this analysis is very useful. Why is the correlation of a local observation with conditions of air parcels upstream relevant? To me only the correlation between dexcess and RH normalized to surface temperature upstream would make sense. This analysis could be left out and the corresponding text removed, it would make the paper more concise and highlight more the key findings.
32) L. 552: I would rather say that convection transports moisture with high dexcess from lower altitudes to higher altitudes (see Thurnherr et al. 2021 WCD, Aemisegger et al. WCD).
33) L. 554: "deeper convection" instead of "more active convection" ?
34) L.. 593: non-equilibrium fractionation at the moisture source
35) L. 590-605: this is a bit repetitive, could be shortened
36) L. 597: it's mainly the decrease in $T$, RH can be very low at the sea ice edge during cold air outbreaks (see, Aemisegger and Papritz, 2018 JC, Thurnherr et al. 2020, ACP)
37) L. 610: what about the role of dew deposition during night over the continent in northern China? See Lee et al. 2021 HESS.

38) L. 621: To me it seems that Iso-GSM strongly underestimates the synoptic timescale variability. What happens to your synoptic vs. seasonal drivers of variability if you use the observations as a measure of synoptic variability?

39) L. 633: Could you speculate about the processes that are responsible for the model biases? Could it be linked to the representation of convection? Or evapotranspiration?

40) L. 633/634: I was slightly confused by the writing here: "overestimation (respectively underestimation)…", could this be written more clearly? Do you mean biases affecting both $\delta^{18}O$ and $q$?

41) L. 649: is the strongly smoothed topography in Iso-GSM the reason behind the bad representation of the altitude effect?

42) L. 664: what is the temporal-mean output of $\delta^{18}O$ from Iso-GSM? A multi-year seasonal mean? Or just the seasonal mean of that particular year?

43) Section 4.7: If possible, this should come much earlier in the manuscript to guide the reader in the interpretation of the observations. Also as mentioned in the major comments, I like the analysis, but I doubt that the number of observations in each region is large enough (sample size) to robustly characterize the synoptic variability encountered in that region. I also suspect that Iso-GSM underestimates the residual $\delta^{18}O_{synoptic}=\delta^{18}O_{daily}-\delta^{18}O_{seasonal}$. A critical discussion of the results would be very beneficial here.

44) Urban emissions: this is an interesting side-discussion. I would however rather see this as a methodological paragraph. Also, I strongly recommend discussing the potentially important baseline effects emerging from rapid changes in concentrations of different trace gases, which can lead to biases in the isotope observations from CRDS systems (see Johnson and Rella, 2017 AMT, Gralher et al., 2016).

**Technical comments:**
- The isotope observational dataset should also be made available online. It is only available upon request according to the data availability statement. A thorough documentation of the data (with adequate metadata description) is key, for making the data accessible to other scientists.
- All figures showing observational data: it would be nice to add the corresponding year in the captions (2019 & 2018).

**References:**

Aemisegger, F., Vogel, R., Graf, P., Dahinden, F., Villiger, L., Jansen, F., Bony, S., Stevens, B., and Wernli, H.: How Rossby wave breaking modulates the water cycle in the North Atlantic trade wind region, Weather Clim. Dynam., 2, 281–309, https://doi.org/10.5194/wcd-2-281-2021, 2021.

Aemisegger, F., and Papritz, L.: A Climatology of Strong Large-Scale Ocean Evaporation Events. Part I: Identification, Global Distribution, and Associated Climate Conditions, *Journal of Climate*, *31*(18), 7287-7312, 2018.

Aemisegger, F., Pfahl, S., Sodemann, H., Lehner, I., Seneviratne, S. I., and Wernli, H.: Deuterium excess as a proxy for continental moisture recycling and plant transpiration, Atmos. Chem. Phys., 14, 4029–4054, https://doi.org/10.5194/acp-14-4029-2014, 2014.

Dansgaard, W., Stable isotopes in precipitation, Tellus, 16:4, 436-468, doi: 10.3402/tellusa.v16i4.8993, 1964.

Gralher, B., Herbstritt, B., Weiler, M., Wassenaar, L. I., and Stumpp, C.: Correcting Laser-Based Water Stable Isotope Readings Biased by Carrier Gas Changes, Environ. Sci. Technol., 50, 7074– 7081, https://doi.org/10.1021/acs.est.6b01124, 2016.

Johnson, J. E. and Rella, C. W.: Effects of variation in background mixing ratios of N2, O2, and Ar on the measurement of δ18O–H2O and δ2H–H2O values by cavity ring-down spectroscopy, Atmos. Meas. Tech., 10, 3073–3091, https://doi.org/10.5194/amt-10-3073-2017, 2017.

Li, Y., Aemisegger, F., Riedl, A., Buchmann, N., and Eugster, W.: The role of dew and radiation fog inputs in the local water cycling of a temperate grassland during dry spells in central Europe, Hydrol. Earth Syst. Sci., 25, 2617–2648, https://doi.org/10.5194/hess-25-2617-2021, 2021.

Salmon, O. E., Welp, L. R., Baldwin, M. E., Hajny, K. D., Stirm, B. H., and Shepson, P. B.: Vertical profile observations of water vapor deuterium excess in the lower troposphere, Atmos. Chem. Phys., 19, 11525–11543, https://doi.org/10.5194/acp-19-11525-2019, 2019.

Thurnherr, I., Hartmuth, K., Jansing, L., Gehring, J., Boettcher, M., Gorodetskaya, I., Werner, M., Wernli, H., and Aemisegger, F.: The role of air–sea fluxes for the water vapour isotope signals in the cold and warm sectors of extratropical cyclones over the Southern Ocean, Weather Clim. Dynam., 2, 331–357, https://doi.org/10.5194/wcd-2-331-2021, 2021.

Thurnherr, I., Kozachek, A., Graf, P., Weng, Y., Bolshiyanov, D., Landwehr, S., Pfahl, S., Schmale, J., Sodemann, H., Steen-Larsen, H. C., Toffoli, A., Wernli, H., and Aemisegger, F.: Meridional and vertical variations of the water vapour isotopic composition in the marine boundary layer over the Atlantic and Southern Ocean, Atmos. Chem. Phys., 20, 5811–5835, https://doi.org/10.5194/acp-20-5811-2020, 2020.

---

## Referee Comment (RC2)

**Review of "Vehicle-based in-situ observations of the water vapour isotopic composition across China: spatial and seasonal distributions and controls" by Di Wang et al. submitted to ACP**

Summary:
This paper serves really well in characterizing the seasonal and spatial variability over a broad region of China over the pre-monsoon and monsoon periods using a vehicle-based isotope analyzer and other meteorological measurements. The synoptic effects from the continents and oceans along with their entanglement with seasonal characteristics are thoroughly investigated using water vapor isotope observations. This paper would be a good fit for ACP publication after the authors address some of the outstanding major and few minor issues.

Major comments:
1) The start of the introduction is strong but it still needs to incorporate a discussion about the current knowledge of synoptic and seasonal processes within the continental region. Why is it important and what do the previous studies broadly say? What is the isotope perspective adding to the traditional methods of study? An expanded version of the first paragraph of Section 2.1 is more fitting for the introduction in my opinion.

2) Since synoptic vs seasonal influences on the isotope observations is the major conclusion of this paper (it literally is in the title of the manuscript), it would help to talk and explain these influences in the introduction. Mention all the factors that you are examining for the two categories. I see that you have classified them in the abstract but it would serve the readers well to know about them from early on. Are there earlier studies that you can compare your results with? Also, whenever you use terms like "altitude effect" and "continental-recycling", use a line or two to give the readers a gist. This would make it convenient for the readers lest they need a jolt to their memory or are not acquainted with the terms well.

3) Another point regarding the synoptic vs seasonal comparison is that although this is the focal point of the paper, and it is only talked about formally in section 4.7. This leaves the readers hanging for the major part of results and discussions. I understand that you talk about the seasonal effects followed by the synoptic effects individually, and then you use Iso-GSM to compare the two effects. However, for more clarity, once you end discussing the seasonal/synoptic effects, write a summary sentence or two to tie up the results to the final seasonal vs synoptic comparison. This will keep the audience' attention till they reach section 4.7. No major editing is necessary. Just adding/shuffling some summary sentences at the end of subsections of sections 3 and 4 would do.

4) Paragraph 3 of Introduction is important since it provides a transition from previous work to your work. However, the phrasing seems weak. If it is a first of its kind study conducted using the isotope analyzers over the continental China, then you should

emphasize it, by all means. Readers would appreciate the new science and techniques that this work is doing.

5) It is scientifically sound to not use the city datasets that may have been impacted by the traffic pollution. However, I am curious, if you can totally eliminate the role of water vapor emitted from country sources like irrigation, farms, power plants etc. in affecting the water vapor isotope or the humidity measurements in any way? If this is an assumption made, then please state it somewhere.

6) I have some concerns regarding the use of Iso-GSM for the disentanglement of seasonal and synoptic influences. From figures 11 and 12, the model does not seem to simulate the observations well, especially in the monsoon period. Moreover, the grid spacing for the models is way coarser compared to the observations. I would suggest using the observations to perform the same analysis discussed in 4.7, and only use model inputs where observations are not available. Then in table 2, compare the Iso-GSM vs observation-based fractions. This would be far more accurate. Since the main result of this paper is disentangling the seasonal vs synoptic effects, I think it is crucial to be thorough with your methods here.

Minor comments:

1) Line 76: Isn't Bailey et al. (2013) a Hawaii-based study? Are there other studies based on larger continental settings?
2) Section 2.1, 1st paragraph: Include continental recycling as well.
3)
4) Line 124: Maybe write a few lines about the advantage of using d-excess, how does d-excess vary in comparison with d18O.
5) Line 146: What is the value of calibration humidity correction term f? Is equation 3 a standard equation for these calculations? Any references?
6) Figure 2,3: Have you tried using the same y-scale for pre-monsoon and monsoon period?
7) Figure 2: Are the meteorological observations also 10-minute average? What is the resolution of P-daily and P-mean respectively? It is not clear to me how P-mean is calculated. If P-mean is the temporal-mean of the precipitation amount over the sampling days, then P-mean should have a single value for each season. Please clarify. Also, mention the grid size of the GPCP precipitation. How is the grid average precipitation calculated?
8) Is the humidity (figure 2e,f) and q (figure 2i,j) obtained from the Picarro sampler and the rooftop weather station, respectively? Do they correlate well? Can the humidity (ppm) also be expressed in (g/kg) for ease of comparison?
9) Line 249: "During the pre-monsoon…..than in any other regions". Here you say that in the pre-monsoon period, the humidity, P-mean and d18O are higher in southwestern

China than in other regions. However, since the data is not available for the entire pre-monsoon period in the northeast and northwest regions, are you assuming that the humidity, p-mean and d18O remain consistent for the entire pre-monsoon period? Have you estimated a sampling bias? It would be good to write this assumption too.

10) Line 277: Alternatively, the high d-excess in south China could also be coming from the moisture flow from Indian/Pacific Ocean as is talked about later. Or, it could be resulting from the deeper convective mixed layer in south China compared to north China where vapor with high d-excess is transported towards the surface as shown in figure 9.

11) Line 314: I find it exciting to see that the controls on d18O (temperature correlation) and d-excess (precipitation-line correlation) can be differentiated so well just based on the observations alone.

12) Change reference Noone and David, 2012 to Noone, 2012.

13) Figure 6: Can you explain the reason for a significant number of WR_1 dried and depleted dots below Rayleigh curve? Are they within the uncertainty range of the Rayleigh curve?

14) Since figure 10 is referred at multiple places in the paragraph starting at line 413, maybe change index of figure 10 to 7 to maintain the flow of the paper.

15) Line 439: replace 'bring' with 'brought'.

16) Line 442: "As continental….". Wrong grammar.

17) Continental recycling from WR2, SR1 and SR3 is a strong factor influencing the isotope ratios. Perhaps, defining it in one line or so will help the readers.

18) Line 458: "…with a steepest slope….". Replace 'a' with 'the'.

19) Line 493: Not all observations exhibit temperature effect. WR3 does not.

20) Line 509: Another factor for the dominating effect of q at least for SR2,3,4 can be rain evaporation.

21) Line 512: Explain the altitude effect in a separate line.

22) What is the difference between daily precipitation amount (P-daily) and temporal-mean precip amount for sampled dates (P-mean)?

23) Line 540: Define 'dtra'.

24) Section 4.7: Going back to my major comment, why start the spatial vs synoptic comparison so late in 4.6 when you introduced the concept at the beginning of section 3? Tie this section with the previous sections.

25) Table 2 description is incomplete.

26) I really like the summary plot 13. It provides a summary of all the meteorological processes described in the paper. Two questions: What is the significance of upward and downward pointing triangles for d-excess? Please explain all the colored arrows in the figure description.

27) 'Bay of Bengal' spelt wrong in the figures 5 and 13.

---

## Author Comment (AC1)

**Response to Reviewers**

We appreciate your insightful comments and suggestions. These have greatly helped us refine our manuscript and improve the clarity of our key points. We also thank you for your positive feedback.

In this response letter, we have addressed each comment in detail below. Our responses are in blue, with specific changes to the text highlighted in *blue italics*. All line numbers in this document correspond to the line numbers in the updated version of the manuscript.

With best regards, Di WANG On behalf of all authors

**AC1: 'Comment on acp-2022-223', 27 Jun 2022**

Review of "Vehicle-based in-situ observations of the water vapour isotopic composition across China: spatial and seasonal distributions and controls" by Di Wang et al. submitted to ACP

This paper presents a very interesting and impressive dataset of vehicle-based stable water isotope measurements in China with several new results and interpretations about the drivers of the variability observed in different regions. The observations cover two seasons: the pre-monsoon season 2019 and the monsoon season 2018, which are compared in terms of their different dynamics and isotope signals recorded along the route. I recommend publication of the paper after two major and several minor comments have been addressed:

We appreciate your positive comment. We also appreciate your detailed suggestions.

**Major comments:**

A) Synoptic vs. seasonal variations: This analysis is very interesting. - However, it comes very late in the manuscript, even though the whole interpretation of the observations evolves along the main finding that the spatial (seasonal) variations dominate the variability of various isotope and meteorological variables. The results section even starts with referring to this analysis but asks the readers to postpone their curiosity to a much later section. Why not starting the results with what you show in Section 4.7?

We agree that the analysis of the seasonal vs synoptic should be earlier in the manuscript. We put it as section 3.3, just after the description of the raw data in section 3.1 and 3.2, and before the seasonal variations in section 3.4 and analyzing the factors controlling the spatial and seasonal distributions in section 4.

- What happens if you do the same regression analysis as presented in Section 4.7 using

the actual observations for estimating the synoptic and seasonal components? Looking at the comparison of the simulation and the observations (Fig. 11), it strikes me that the daily model output follows the "temporal-mean" output much more closely than the actual measurements in both the premonsoon and the monsoon seasons. To me this shows that the Iso-GSM simulation on a relatively coarse grid is not capable of reproducing the observed mesoscale to large-scale variability in the water vapour isotope and meteorological fields. Therefore, in my view the finding that the isotope variability across China in the pre-monsoon and monsoon periods is mainly a result of seasonal/spatial variations and only marginally affected by synoptic-scale systems is biased towards what the Iso-GSM shows.

We accepted your suggestion and assessed the relative contribution using multiyear averages from 2010 to 2020 and a new estimation method that takes into account model bias and gives upper and lower bounds on the value of the contribution.

We modified (lines 327-361):

"2.6 Method to decompose the observed daily variations

The temporal variations observed along the route for a given period represent a mixture of synoptic-scale perturbations, and of seasonal-mean spatial distribution:

(4)

 $\delta^{18}O_{daily} = \delta^{18}O_{seaso} + \delta^{18}O_{synoptic}$

The first term represents the contribution of seasonal-mean spatial variations, whereas the second term represents the contribution of synoptic-scale variations. Since these relative contributions are unknown, we use outputs from Iso-GSM. The daily variations of  $\delta^{18}O$  simulated by Iso-GSM also represent a mixture of synoptic-scale perturbations and seasonal-mean spatial distribution, but with some errors relative to reality:

 $\delta^{18}O\_daily\_Iso\_GSM = \delta^{18}O\_daily\_Iso\_GSM + \in =(\delta^{18}O\_seaso\_Iso\_GSM + \in\_seaso] + (\delta^{18}O\_synoptic\_Iso\_GSM + \in\_synoptic]$   $GSM + \in\_synoptic]$  (5)

where  $\delta^{18}O_{daily\_Iso-GSM}$  is the daily outputs of  $\delta^{18}O$  for each location,  $\delta^{18}O_{seaso\_Iso-GSM}$  is the multi-year monthly outputs of  $\delta^{18}O$  for each location, and  $\delta^{18}O_{synoptic\_Iso-GSM} = \delta^{18}O_{daily\_Iso-GSM} - \delta^{18}O_{seaso\_Iso-GSM}$ ,  $\epsilon_{seaso}$  and  $\epsilon_{synoptic}$  are the errors on  $\delta^{18}O_{seaso\_Iso-GSM}$  and  $\delta^{18}O_{synoptic\_Iso-GSM}$  relative to reality, respectively,  $\epsilon$  is the sum of  $\epsilon_{seaso}$  and  $\epsilon_{synoptic}$ .

These individual error components  $\in_{seaso}$  and  $\in_{synoptic}$  are unknown, but we know the sum of them ( $\in$ ), i.e. the difference between daily outputs and observations. For the decomposition, we made two extreme assumptions to estimate upper and lower bounds on the contribution values:

1) We assume that the error is purely seasonal-mean, i.e.  $\in = \in_{seaso}$ , and  $\in_{synoptic}=0$ :

 $\delta^{18}O_{daily} = \delta^{18}O_{seaso\_Iso\_GSM} + (\delta^{18}O_{synoptic\_Iso\_GSM} + \epsilon)$ (6)

To evaluate the contribution of these two terms, we calculate the slopes of  $\delta^{18}O_{-daily}$ as a function of  $\delta^{18}O_{-seaso_{-}Iso-GSM}$  ( $a_{-seaso}$ ), and of  $\delta^{18}O_{-daily}$ - $\delta^{18}O_{-seaso_{-}Iso-GSM}$  ( $a_{-synoptic}$ ). The relative contributions of spatial and synoptic variations correspond to  $a_{-seaso}$  and  $a_{-synoptic}$  respectively. This will be the upper bound for the contribution of synoptic-scale variations, since some of the systematic errors of Iso-GSM will be included in the synoptic component. This is equivalent to using the seasonal-mean of Iso-GSM and the raw time series of observations.

2) We assume that the error is purely synoptic, i.e.  $\epsilon = \epsilon_{synoptic}$ , and  $\epsilon_{seaso} = 0$ : Then  $\delta^{18}O_{daily} = (\delta^{18}O_{seaso_{Iso-GSM}} + \epsilon) + \delta^{18}O_{synoptic_{Iso-GSM}}$ . (7)

To evaluate the contribution of these two terms, we calculate the slopes of  $\delta^{18}O_{daily\_Iso-GSM}$  as a function of  $\delta^{18}O_{seaso\_Iso-GSM}$  ( $a_{seaso}$ ), and of  $\delta^{18}O_{daily\_Iso-GSM}$  -  $\delta^{18}O_{seaso\_Iso-GSM}$  ( $a_{synoptic}$ ). This will be the lower bound for the contribution of synoptic-scale variations, since we expect Iso-GSM to underestimate the synoptic variations.

*The same analysis is also performed for the Iso-GSM simulation q (Table 2) and reanalysis q (Table 3).*"

We updated the discussion on disentangling seasonal-mean and synoptic variations in section 3.3 (lines 446-466):

"**Table 2** The relative contribution (in fraction) of spatial variations for a given season ( $a_{seaso}$ ) and of synoptic-scale variations ( $a_{synoptic}$ ) to the daily variations of qand  $\delta^{18}O$  simulated by Iso-GSM. We checked that the sum of  $a_{seaso}$  and  $a_{synoptic}$  is always 1.

| Period      | variables      | Controbutions |            |  |
|-------------|----------------|---------------|------------|--|
|             |                | a_seaso       | a_synoptic |  |
| Pre-monsoon | q              | 0.73~1.05     | 0.27~-0.05 |  |
|             | $\delta^{18}O$ | 0.54~0.70     | 0.46~0.30  |  |
| Monsoon     | q              | 0.63~0.71     | 0.37~0.29  |  |
|             | $\delta^{18}O$ | -0.50~0.10    | 1.5~0.9    |  |

*Table 3 The same as Table 2, but for reanalysis q.*

| Period      | variables | Controbutions |            |
|-------------|-----------|---------------|------------|
|             |           | a_seaso       | a_synoptic |
| Pre-monsoon | q         | 0.77~0.88     | 0.23~0.12  |
| Monsoon     | q         | 0.69~0.88     | 0.31~0.12  |

During the pre-monsoon period, based on both the Iso-GSM simulation and NCEP/NCAR reanalysis, we can find that the seasonal-mean contribution to q is higher than the synoptic-scale contribution:  $a_{seaso}$  is 73%-105% for Iso-GSM and 77%-88% for reanalysis, whereas  $a_{synoptic}$  is 27%~ -5% for Iso-GSM and 23%~12% for reanalysis (Table 2 and Table3). The relative contribution of seasonal-mean spatial variations to the total simulated variations of  $\delta^{18}O$  (54%-70%) is also much higher than that of synoptic-scale variations (27%- -5%). This suggests that the observed variability in q and  $\delta^{18}O$  is mainly due to spatial variability, and marginally due to synoptic-scale variability. During the monsoon, seasonal-mean spatial variations is also the main contribution to the observed q variations. But it is different for  $\delta^{18}O$ : the relative contribution of synoptic-scale variations (90%-150%) dominates the total simulated variations of  $\delta^{18}O$ ."

- What exactly is the "temporal-mean" output? A multi-year seasonal mean? Or the

**mean over the considered 2018 and 2019 seasons? And over which spatial scale did you average?**

We have added a description and application of the "temporal-mean outputs" in the manuscript to make it clearer (lines 319-322):

"We also use the multi-year monthly-mean outputs from Iso-GSM (March monthly for the pre-monsoon period and August monthly for the monsoon period) for each observation location from 2011 to 2020 to quantify the relative contributions of seasonal-mean and synoptic-scale variations."

- I am not convinced that you can conduct this analysis in a robust way, given the few observations you have for a given region and season. A discussion on this sampling issue of the seasonal signal from only few synoptic events would be beneficial. Still, it is a valuable analysis especially if it is done with the measurements and the model, comparing and discussing the two results.

We carried out this analysis on a spatial scale for a dense set of observations using the method in the response to comment 1. Comparison with multi-year averages reveals that during the pre-monsoon, spatial variability dominates for both q and  $\delta^{18}O$ ; during the monsoon, q still mainly reflects seasonally averaged spatial variability, however,  $\delta^{18}O$  is more influenced by synoptic-scale processes that may include different sources of water vapor, convective processes, etc., as discussed in the manuscript.

We added more discussion on the impact of possible synoptic-scale processes (lines 468-485):

**"3.3.3 Diagnosing the reasons for the GCMs performance**

"Since typhoons are known to be associated with depleted rain and vapor (Gedzelman et al 2003, Xu et al 2019, Battacharya et al 2022), during our monsoon observations, landfall of tropical cyclones Jongdari and Yagi correspond to the low values of  $\delta^{18}$ O we observed in the eastern China (Fig.S7a), Bebinca corresponds to the low values of  $\delta^{18}$ O we observed in the southwestern China (Fig.S7a). Iso-GSM captures the large-scale circulation associated with a tropical cyclone, but given its coarse resolution, it underestimates the depletion associated with the meso-scale structure of the cyclone (Fig.7). The northward propagation of the Northern Summer Intra-Seasonal Oscillation (BSISO) during July-August 2018 allowed for strong convection over large areas of China (Susskind et al., 2011;Kikuchi, 2021) (Fig. S8). Moreover, short-lived convective events that frequently occurred during our observation period (Jing Wang et al., 2018). It is possible that these rapid high-frequency synoptic events are not fully captured by Iso-GSM. Iso-GSM underestimates the observed variability, especially synoptic timescale variability. Thus, Iso-GSM performs well during the pre-monsoon season, when seasonal mean spatial variability dominates q and  $\delta^{18}$ O. In contrast, it performs less well during the monsoon season, when  $\delta^{18}$ O variations are mainly due to the synoptic-scale variability. This could explain the different performances of Iso-GSM during pre-monsoon and monsoon periods."

B) Organization of the manuscript: The reading of the manuscript would profit from a re-organization with a clearer separation of methodological aspects and results section.

Many methodological aspects are in the results and interrupt the flow of the reader (regional analysis in Table 1, Urban emissions, methodological aspects on Eq. 4)

We accepted your suggestion, we have moved regional analysis in Table 1 to section 2.4, moved section "4.7 urban emissions" to section 2.2.6, and moved methodological aspects on Eq. 4 to section 2.5.

**Minor comments:**

 L. 25: "large-scale (order 10000 km) continuous observations of near-surface vapor isotopes": can be misunderstood, reformulate. You made in-situ observations over a large area. Not continuous observations at multiple locations.

We have reworded as "in-situ observations of near-surface vapor isotopes over a large region (order 10000 km) across China" to avoid misunderstandings.

2) L. 28-29: "mainly due to spatial variations and marginally influenced by synoptic-scale variations": This is interesting! I was very curious about how you came to this conclusion, when starting to read your manuscript. When just reading the abstract, I found this statement and the following ones about the importance of Rayleigh distillation (cloud formation during large-scale ascent?), different moisture sources (variability induced by large-scale weather systems?), continental moisture recycling (transport regimes favouring oceanic vs. continental sources) and convection (mesoscale circulation) contradicting. First you write that the "spatial variations" dominate and then you mention different processes that are relevant at the synoptic- to meso- scale. The synoptic systems are very different in southern vs. northern China and these weather systems shape the spatial contrasts. It would be helpful if you could concisely state in the abstract how you come to the conclusion that "seasonal"/"spatial" variations dominate the synoptic variability and think carefully about the best terminology to use.

We have reworded the abstract to clarify what processes play a role at the seasonal-mean and synoptic-scale:

"The spatial variations during the pre-monsoon period represent mainly seasonal-mean variations, but significantly influenced by synoptic-scale variations during the monsoon period. During the pre-monsoon period, the spatial variations of vapor  $\delta^{18}O$  are mainly controlled by Rayleigh distillation along air mass trajectories. The North-South gradient observed during the pre-monsoon period is counteracted by different moisture sources, continental recycling processes and convection during moisture transport during the monsoon period."

**3) L. 38-39: Why is the performance of the Iso-GSM model weaker over the monsoon period?**

We point out the main reasons for the different performance of Iso-GSM in the pre-monsoon and monsoon periods (lines 462-466):

"3.3.3 Diagnosing the reasons for the GCMs performance

During our monsoon observations, Super Typhoon Jundari, Super Tropical Storms Yagi and Bebinca, and Super Tropical Storm Lumbia occurred in the China region. Since typhoons are known to be associated with depleted rain (Gedzelman et al 2003, Xu et al 2019, Battacharya et al 2022), they probably contributed to the low values of  $\delta^{18}O$  we observed during the corresponding period (Fig. S7). In addition, the northward propagation of the Northern Summer Intra-Seasonal Oscillation (BSISO) during July-August 2018 allowed for strong convection over large areas of China (Susskind et al., 2011;Kikuchi, 2021), particularly in the south (from latitudes 30N to 45N) (Fig. S8). It is possible that these rapid high-frequency synoptic events are not fully captured by Iso-GSM. Iso-GSM underestimates the observed variability, especially synoptic timescale variability. Thus, Iso-GSM performs well during the premonsoon season, when seasonal mean spatial variability dominates q and  $\delta^{18}O$ . In contrast, it performs less well during the monsoon season, when  $\delta^{18}O$  variations are mainly due to the synoptic-scale variability. This could explain the different performances of Iso-GSM during pre-monsoon and monsoon periods."

4) L. 46: The first sentence of the introduction is a bit heavy, could you think of a more general motivation for your study? And think about how to guide a non-isotope specialist reader into your subject.

We have reworded the beginning of the introduction (lines 49-52):

"Stable water isotopes have been applied to study a wide range of hydrological and climatic processes (Gat, 1996;Bowen et al., 2019;West et al., 2009). This is because water isotopes vary with changes in the water phases (e.g., evaporation, condensation), and therefore produce a natural labeling effect within the global water cycle."

5) L. 54: Are only Tibetan plateau ice cores questioned in terms of their use as past temperature records?

We modified "isotopic records in Tibetan ice cores" to "isotopic records in ice cores from low and middle latitudes regions".

6) L. 56: "significant role of large-scale"

We have modified "the significant role of large regional atmospheric circulation" to "the significant role of large-scale atmospheric circulation".

**7) L. 57: is a specific teleconnection mode meant here?**

Based on suggestions from the other reviewer, we have reorganized the introduction and removed this sentence.

8) L. 74-77: Isn't the Bailey et al. 2013 study from an oceanic environment? Aemisegger et al. 2014 ACP discusses the isotope variability of continental evaporation induced by synoptic-scale variability.

We modified as (104-105):

"One study made vehicle-based in-situ observations to document spatial variations, but this was restricted to the Hawaii island (Bailey et al., 2013)."

- 9) L. 97: Dry intrusions bring... cold and dry upper tropospheric airmasses We have modified "Dry intrusions" to "Westerlies".
- 10) Fig. 1: the route is a bit difficult to make out from the otherwise very nice figure Thank you for your suggestion, we highlighted the observation routes.

**11) L. 122: is computed**

We have modified "are computed" to "is computed".

**12) L. 164: no drift in the standards: could you indicate the standard deviation of your series of standard measurements?**

We added the standard deviation of standard measurements during our campaigns (lines 197-200):

"For two standard waters, the standard deviation of standard measurements are 0.2‰ and 0.11‰ for  $\delta^{18}$ O, and 1.16‰ and 1.2‰ for  $\delta^{2}$ H during the pre-monsoon period of 2019. During the monsoon period of 2018, the standard deviation of standard measurements are 0.09‰ and 0.06‰ for  $\delta^{18}$ O, and 0.6‰ and 0.33‰ for  $\delta^{2}$ H."

**13) Section 2.2.3: should be humidity-dependent isotope bias correction**

We have modified "Humidity calibration" to "Humidity-dependent isotope bias correction".

**14) Was the specific humidity calibrated as well using an independent sensor?**

Thank you for your suggestions, we added (lines 165-169):

"The specific humidity measured by Picarro is very close to that measured by an independent sensor installed in the vehicle (Fig.4). The correlation between the humidity measured by the Picarro and the independent sensor are over 0.99, the slopes are approximately 1 and the average deviation are less than 1 g/kg both during premonsoon and monsoon periods."

**15) L. 212: why was only one air parcel started from 1000 m above the surface?**

Tracing parcels starting from different altitudes would provide information on air mass mixing. However, due to the large number of our observation points, this would reduce the visibility of the backward trajectory results in Figure 2 and Figure 4. Therefore, we follow the conventional setup that people usually use in tracking near-surface water vapor sources, i.e., a tracking starting point of 1000 m from the ground (Guo et al., 2017;Bershaw et al., 2012),. We believe that this is representative of the backward trajectory of near-surface air, since most water vapor in the atmosphere is within 0–2 km above ground level (Wallace and Hobbs, 2006).

We added (lines 273-275):

"This is representative of the water vapor near the ground (Guo et al., 2017;Bershaw et al., 2012), since most water vapor in the atmosphere is within 0–2 km above ground level (Wallace and Hobbs, 2006)."

16) L. 229-234: This is a bit a difficult start of the results section. Could the analysis of the seasonal vs. synoptic drivers of isotope variability not come as a first result? It is otherwise very hard for the reader to follow the story.

We accepted your suggestion to place the analysis of seasonal versus synoptic drivers of isotopic variation earlier in the manuscript (section 3.3).

**17) L. 230: refer to Fig. 2a,b**

We have referred to these two figures in this sentence. As the order of the figures has changed, it is currently Figure 4 and Figure 5 (lines 364-365):

"Our survey of the vapor isotopes yields two snapshots of the isotopic distribution along the route (Fig.4 & Fig.5)."

18) L. 269: that could be the continental effect too... Should Dansgaard et al. 1964 be cited if you discuss the isotope effects? These different effects are not mutually exclusive, and in my opinion, they do not provide a direct mechanistic explanation for the correlation of the delta-values with temperature.

We cited Dansgaard et al. 1964 in the manuscript when we discuss the isotope effects:

*Lines* 513-514: *"the altitude effect (the heavy isotope concentrations in fresh water decreasing with increasing altitude, Dansgaard 1954)"*

*Lines* 690-691: "During the pre-monsoon period, all observations taken together exhibit a "temperature effect" (the  $\delta$ 's decreasing with temperature, Dansgaard 1964)"

We moved the discussion of the spatial variation of water vapor isotopes with temperature to Section 4. This discussion is further developed in the context of temperature data and correlation coefficients. In this section, we focus on the description of the observed data.

19) Section 3.2: I like the result about the fact that the monsoon counteracts the N-S gradient observed in the pre-Monsoon period very much. This could be included as a key result in the abstract.

We incorporated this great idea in the abstract, we modified lines 30-36 as:

"During the pre-monsoon period, the spatial variations of vapor  $\delta^{18}O$  are mainly controlled by Rayleigh distillation along air mass trajectories. The North-South gradient observed during the pre-monsoon period is counteracted by different moisture sources, continental recycling processes and convection during moisture transport during the monsoon period. The seasonal variation of vapor  $\delta^{18}O$  reflects the influence of the summer monsoon convective precipitation in southern China, and a dependence on temperature in the North."

20) L. 310-311: I don't understand this sentence. What is the separation line of the seasonal variation of dexcess? Reformulate.

We modified as (lines 532-534):

"The line separating the areas of positive and negative values of the dexcessmonsoon - d-excesspre-monsoon differences coincides with the 120 mm P-mean line (Fig.S2 f)."

**21) P. 11-12: this is a methodological part and should not be in the results section.**

We have now moved pages 11-12 to section 2.4 untitled "Back-trajectory calculation and categorizing regions based on air mass origin".

**22) L. 439: bring -> brought**

We have modified "bring" to "brought".

23) L. 442: replace "strong" by "high". Actually, the fraction of transpiration in evapotranspiration is the key factor determining d over the continent, when recycling is high (see Aemisegger et al. 2014, ACP).

We have modified "strong" to "high", and also added the reference for discussion (lines 637-639):

"These properties are associated with a high d-excess, consistent with strong continental recycling and by evapotranspiration (Aemisegger et al., 2014)."

**24) L. 369: reduction of humidity and water isotope ratios.**

We have modified "reduction of vapor isotope ratios" to "reduction of humidity and water isotope ratios" (line 163).

**25) Figure 6 is very interesting. Could it come earlier?**

We have tried many ways to reorganize the paper, but since following reviewers' comment we have moved the section disentangling seasonal/synoptic variations earlier, and the moisture source categorization to the method section, this figure actually comes a bit later. However, it comes as the first figure of the discussion section 4, untitled "Understanding the factors controlling the spatial and seasonal distributions".

**26) L. 391: this sounds a bit speculative: couldn't it be another Rayleigh line or mixing?**

In current Figure 9, we added the uncertainty range of the Rayleigh curve calculated for different initial conditions of key moisture source regions: during March 2019, light red and light blue Rayleigh curve are calculated for key moisture source regions of westerlies ( $\delta^2 H_0 = -168.04\%$ , T=5°C) and BoB ( $\delta^2 H_0 = -77.37\%$ , T=26.46°C) separately in (a); during July-August 2018, light red and light blue Rayleigh curve are calculated for key moisture source regions of westerlies ( $\delta^2 H_0 = -82.75\%$ , T=27.69°C) separately in (b). These initial  $\delta^2 H$  are derived from iso-GSM, the initial temperature and RH are derived from NCAR/NCEP 2.5-deg global reanalysis data.

We can find that observations in the WR\_3 region (Fig.3c) are truly located below the q- $\delta^2$ H Rayleigh distillation curve calculated for general values for pre-monsoon period and calculated for it's key moisture source region of BoB, we modified as (lines 568-571) :

"Observations below the Rayleigh line, even when considering the most depleted initial vapor conditions (light blue Rayleigh curve in Fig 9b), indicate the influence of rain evaporation from depleted precipitation (Noone, 2012; Worden et al., 2007)."

27) L. 493: "all observations taken together" because WR2 and WR3 do not show a "temperature effect".

We modified "all observations" to "all observations taken together" (line 690).

**28) L. 501: influenced**

We have modified "influence" to "influenced".

**29) L. 509: degree of rain out**

We modified "degrees along the Rayleigh distillation" to "degree of rain out" (line 707).

30) L. 517: why does the dexcess usually increase with altitude? Many recent studies show that it is more complicated than that (see, Salmon et al. 2019 ACP, Thurnherr et al. 2021 WCD). In many cases a decrease of dexcess towards the mid troposphere is observed.

We agree that our observations of altitude variation within 2km are not sufficient to discuss the relationship between d-excess and altitude. We deleted the sentence "Alternatively, this may reflect the fact that the d-excess generally increases with altitude (Galewsky et al., 2016)".

We explain the positive correction between d-excess and altitude by the following sentence: "The vapor d-excess for all observations during the monsoon period (Fig.8d) is positively correlated with Alt (r = 0.39, p<0.05, Table S1). One possible reason is that the vapor d-excess is lower in coastal areas at lower altitudes, while at higher altitudes in the west, more recycling moisture leads to higher d-excess. The positive correlation between d-excess and altitude is consistent with previous studies in region (Acharya et al 2020)."

31) Fig. 9: I am not sure this analysis is very useful. Why is the correlation of a local observation with conditions of air parcels upstream relevant? To me only the correlation between dexcess and RH normalized to surface temperature upstream would make sense. This analysis could be left out and the corresponding text removed, it would make the paper more concise and highlight more the key findings.

We would like to keep this short discussion because many previous studies have shown that convection has a stronger impact on the vapor isotopic composition when integrated along trajectories than when considered locally in monsoon region. Therefore, we keep this analysis for two proxies of convection: precipitation (proxy for deep convection) and boundary layer mixing depth (proxy for both shallow and deep convection).

We added a few sentences to introduce the purpose of this section (lines 735-739):

"Past reconstructions of paleoclimate using ice core isotopes have relied on relationships with local temperatures, but many previous studies have suggested that water isotopes are driven by remote processes along air mass trajectories. In particular, they emphasized the importance of upstream convection in controlling the isotopic composition of water vapor (Gao et al., 2013;He et al., 2015;Vimeux et al., 2005, Cai and Tian, 2016)"

We pointed out the relevance of the relationship between water vapor isotopes and temperature and humidity on water vapor transport pathways (lines 744-747) :

"The results show gradually increasing positive correlation coefficients as dtra changes from 10 to 3. This reflects the role of temperature and humidity along air mass trajectories and the large spatial and temporal coherence of T variations during the pre-monsoon period."

32) L. 552: I would rather say that convection transports moisture with high dexcess from lower altitudes to higher altitudes (see Thurnherr et al. 2021 WCD, Aemisegger et al. WCD).

Many studies documents an increase of d-excess with altitude in the free troposphere (Galewsky et al 2016, Salmon et al 2019), though the vertical profiles can be more complex in the boundary layer and just above it (Salmon et al 2010, Thurnerr et al 2021) or over oceanic regions in dry conditions (Aemisegger et al 2021). Vertical profiles measured in China (unpublished) confirm the increase of d-excess with altitude in our geographical setting.

33) L. 554: "deeper convection" instead of "more active convection" ? We modified "more active convection" to "deeper convection".

**34) L. 593: non-equilibrium fractionation at the moisture source**

We modified "kinetic fractionation" to "non-equilibrium fractionation at the moisture source".

35) L. 590-605: this is a bit repetitive, could be shortened We have abbreviated this paragraph.

36) L. 597: it's mainly the decrease in T, RH can be very low at the sea ice edge during cold air outbreaks (see, Aemisegger and Papritz, 2018 JC, Thurnherr et al. 2020, ACP)

We add this reference (lines 795-799):

"This is consistent with the global-scale poleward decrease in T and increase in surface RH over the oceans (despite the occurrence of very low RH at the sea ice edge during cold air outbreaks (Aemisegger and Papritz, 2018, Thurnherr et al. 2020), resulting in global-scale poleward decrease in d-excess at mid-latitudes (Risi et al., 2013a; Bowen and Revenaugh, 2003)."

37) L. 610: what about the role of dew deposition during night over the continent in northern China? See Lee et al. 2021 HESS.

We agree that dew deposition during night could also contribute to the low dexcess over the continent in northern China. We add this reference (lines 799-800): "Alternatively, the low-dexcess during the night over the continent in Northern China during the pre-monsoon could also contribute to the low-dexcess (Lee et al 2021)."

38) L. 621: To me it seems that Iso-GSM strongly underestimates the synoptic timescale variability. What happens to your synoptic vs. seasonal drivers of variability if you use the observations as a measure of synoptic variability?

As in the response to the first major comment, we now use observations to estimate the relative contributions, giving the upper and lower bounds on the values of the contribution accounting for errors in Iso-GSM.

39) L. 633: Could you speculate about the processes that are responsible for the model biases? Could it be linked to the representation of convection? Or evapotranspiration?

We added the interpterion for the bias (lines 500-503):

"The differences in  $\delta^{18}O$  (Fig.7a) and q (Fig.7c) are spatially consistent. The overestimation of  $\delta^{18}O$  therefore could be due to the overestimation of q, and vice versa. These biases could be associated with deviations in the representation of convection or in continental recycling."

40) L. 633/634: I was slightly confused by the writing here: "overestimation (respectively underestimation)...", could this be written more clearly? Do you mean biases affecting both  $^{18}$ O and q?

What I would like to explain by this is that regions that overestimate q also overestimate  $\delta^{18}$ O, and conversely, regions that underestimate q also underestimate  $\delta^{18}$ O. I modified it (lines 500-501):

"The overestimation of  $\delta^{18}O$  therefore could be due to the overestimation of q, and vice versa."

41) L. 649: is the strongly smoothed topography in Iso-GSM the reason behind the bad representation of the altitude effect?

We corrected for the altitude difference between observations and Iso-GSM using the method given in III. Supplementary Text. We have the same results. Therefore, we think that the smoothed topography is not the main reason.

**42) L. 664: what is the temporal-mean output of 18O from Iso-GSM? A multi-year seasonal mean? Or just the seasonal mean of that particular year?**

We have made the definition clearer in the manuscript (lines 319-322):

"We also use the multi-year monthly-mean outputs from iso-GSM (March monthly for the pre-monsoon period and August monthly for the monsoon period) for each observation location from 2011 to 2020 to quantify the relative contributions of seasonal-mean and synoptic-scale variations."

43) Section 4.7: If possible, this should come much earlier in the manuscript to guide the reader in the interpretation of the observations. Also as mentioned in the major

comments, I like the analysis, but I doubt that the number of observations in each region is large enough (sample size) to robustly characterize the synoptic variability encountered in that region. I also suspect that Iso-GSM underestimates the residual 18Osynoptic= 18Odaily- 18Oseasonal. A critical discussion of the results would be very beneficial here.

We have placed the analysis of seasonal versus synoptic drivers of isotopic variation earlier in the manuscript (in section 3.3).

We have now improved our decomposition method to account for errors in Iso-GSM for both seasonal and synoptic-scale variations. Taking into account these errors, we slightly modify one of our conclusions by noting the greater influence of synopticscale variability during the monsoon period than seasonally averaged spatial variability.

44) Urban emissions: this is an interesting side-discussion. I would however rather see this as a methodological paragraph.

We accepted your suggestion and moved this section to data and method section (section 2.2.6 Data processing).

Also, I strongly recommend discussing the potentially important baseline effects emerging from rapid changes in concentrations of different trace gases, which can lead to biases in the isotope observations from CRDS systems (see Johnson and Rella, 2017 AMT, Gralher et al., 2016).

We added a short discussion about the potentially important baseline effects emerging from rapid changes in concentrations of different trace gases (lines 228-230):

"Some of these d-excess anomalies are not excluded from being affected by the baseline effects emerging from rapid changes in concentrations of different trace gases. (Johnson and Rella, 2017; Gralher et al., 2016)"

Technical comments: - The isotope observational dataset should also be made available online. It is only available upon request according to the data availability statement. A thorough documentation of the data (with adequate metadata description) is key, for making the data accessible to other scientists.

Thank you for your reminder and detailed suggestions. We had submitted the isotope observational dataset to the PANGEA repository. We also described the data in detail and indicate the time of observation. However, publishing data in this database requires a long review period. Therefore the data was not available to readers online at the time we submitted this article. The datasets are now available.

We modified the data availability section as (lines 874-880):

"The data acquired during the field campaigns used can be accessed via the following link or DOI: (1) Wang, Di; Tian, Lide (2022): Vehicle-based in-situ observations of the water vapor isotopic composition across China during the monsoon season 2018. PANGAEA, https://doi.org/10.1594/PANGAEA.947606; (2)Wang, Di; Tian, Lide (2022): Vehicle-based in-situ observations of the water vapor isotopic composition across China during the pre-monsoon season 2019. PANGAEA, https://doi.org/10.1594/PANGAEA.947627."

- All figures showing observational data: it would be nice to add the corresponding

year in the captions (2019 & 2018).

We accepted your suggestion and add the corresponding year in the captions (2019 & 2018).

---

## Author Comment (AC2)

**Response to Reviewers**

We appreciate your insightful comments and suggestions. These have greatly helped us refine our manuscript and improve the clarity of our key points. We also thank you for your positive feedback.

In this response letter, we have addressed each comment in detail below. Our responses are in blue, with specific changes to the text highlighted in *blue italics*. All line numbers in this document correspond to the line numbers in the updated version of the manuscript.

With best regards,
Di WANG
On behalf of all authors

**AC2: 'Comment on acp-2022-223', 30 Sep 2022**

Review of "Vehicle-based in-situ observations of the water vapour isotopic composition
across China: spatial and seasonal distributions and controls" by Di Wang et al. submitted to ACP

Summary:
This paper serves really well in characterizing the seasonal and spatial variability over a broad region of China over the pre-monsoon and monsoon periods using a vehicle-based isotope analyzer and other meteorological measurements. The synoptic effects from the continents and oceans along with their entanglement with seasonal characteristics are thoroughly investigated using water vapor isotope observations. This paper would be a good fit for ACP publication after the authors address some of the outstanding major and few minor issues.

Thank you very much for reviewing our paper. We appreciate your positive feedback and detailed suggestions.

Major comments:
1) The start of the introduction is strong but it still needs to incorporate a discussion about the current knowledge of synoptic and seasonal processes within the continental region. Why is it important and what do the previous studies broadly say? What is the isotope perspective adding to the traditional methods of study? An expanded version of the first paragraph of Section 2.1 is more fitting for the introduction in my opinion.

Thank you for your suggestions to make our introduction section clearer. We have integrated the first paragraph of section 2.1 into the introduction, and added the introduction about the seasonal and synoptic-scale variations (lines 66-90):

*"In general, large parts of the country are affected by the Indian monsoon and the East Asian monsoon in summer, which bring humid marine moisture from the Indian Ocean, South China Sea, and Northwestern Pacific Ocean (Fig.1). During the non-*

*monsoon seasons, the Westerlies influence most of northern China (Fig.1). Westerlies brings extremely cold and dry air masses. Occasional moisture flow from the Indian Ocean and/or Pacific Ocean brings moisture to southern China. Continental recycling, i.e. the moistening of the near-surface air by the evapo-transpiration from the land surface (transpiration by plants, evaporation of bare soil or standing water bodies, Brubaker et al 2013), is also an important source of water vapor in both seasons. Some of the spatial and seasonal patterns of water vapor transport are imprinted in the observed station-based precipitation isotopes (Araguás-Araguás et al., 1998;Tian et al., 2007;Wright, 1993;Mei'e et al., 1985;Tan, 2014). However, precipitation isotopes can only be obtained at a limited number of stations and on rainy days. The lack of information makes it limited to analyze the effects of water vapor propagation, alternating monsoon and westerlies, and synoptic-scale processes on the spatial and seasonal variation of water isotopes. In particular, the seasonal pattern and the spatial variation of water isotopes can strongly influenced by synoptic-scale processes, through their influence on moisture source, transport, convection and mixing processes (Wang et al., 2021;Sánchez-Murillo et al., 2019;Klein et al., 2015), which requires higher frequency observations. For example, some studies founded the impact of tropical cyclones (Gedzelman et al 2003, Xu et al 2019, Battacharya et al 2022) the Northern Summer Intra-Seasonal Oscillation (BSISO) (Susskind et al., 2011;Kikuchi, 2021), local or large-scale convections (Shi et al., 2020), cold front passages (Aemisegger et al., 2015), depressions(Saranya et al., 2018), and anticyclones (Khaykin et al., 2022) on water isotopes in the Asian region. Additional data and analysis refining our understanding of controls on the spatial and temporal variation of water isotopes in low-latitude regions therefore are needed."*

2) Since synoptic vs seasonal influences on the isotope observations is the major conclusion of this paper (it literally is in the title of the manuscript), it would help to talk and explain these influences in the introduction. Mention all the factors that you are examining for the two categories. I see that you have classified them in the abstract but it would serve the readers well to know about them from early on. Are there earlier studies that you can compare your results with?

We introduced the seasonal and synoptic-scale variations and the factors that we are examining for the two categories. (Same lines as for the last response. Because the responses to the first and second advice are interrelated, they are not listed separately).

Also, whenever you use terms like "altitude effect" and "continental-recycling", use a line or two to give the readers a gist. This would make it convenient for the readers lest they need a jolt to their memory or are not acquainted with the terms well.

We added the gist for "altitude effect", "continental-recycling" and "temperature effect":

*Lines 513-514: "the altitude effect (the heavy isotope concentrations in fresh water decreasing with increasing altitude, Dansgaard 1954)".*

*Lines690-691: "During the pre-monsoon period, all observations taken together*

*exhibit a "temperature effect" (the δ's decreasing with temperature, Dansgaard 1964 )".*

*Lines 72-74: "Continental recycling , i.e. the moistening of the near-surface air by the evapo-transpiration sfrom the land surface (transpiration by plants, evaporation of bare soil or standing water bodies, Brubaker et al 2013)".*

3) Another point regarding the synoptic vs seasonal comparison is that although this is the focal point of the paper, and it is only talked about formally in section 4.7. This leaves the readers hanging for the major part of results and discussions. I understand that you talk about the seasonal effects followed by the synoptic effects individually, and then you use Iso-GSM to compare the two effects. However, for more clarity, once you end discussing the seasonal/synoptic effects, write a summary sentence or two to tie up the results to the final seasonal vs synoptic comparison. This will keep the audience' attention till they reach section 4.7. No major editing is necessary. Just adding/shuffling some summary sentences at the end of subsections of sections 3 and 4 would do.

We agree that the analysis of the seasonal vs synoptic is important to interpret the observed variations. We now moved this section earlier in the manuscript. We put it as section 3.3, just after the description of the raw data in section 3.1 and 3.2, on this basis, and before the seasonal variations in section 3.4 and analyzing the factors controlling the spatial and seasonal distributions in section 4.

For our main results, we also focus on a better understanding the factors controlling the spatial and seasonal distributions, not only the relative contribution of seasonal-mean and synoptic variations. Synoptic vs seasonal influences is part of result and a basis for the analysis of controlling factors.

To clarify this we have reworded the last paragraph of introduction (lines 112-119):

*"After describing our observed time series along the route (section 3.1 and 3.2), we quantify the relative contributions of seasonal-mean spatial variations and synoptic-scale variations that locally disturb the seasonal-mean to our observed time series (section 3.3). We show that our observed variations during the pre-monsoon period are dominated by spatial variations but significantly influenced by synoptic-scale variations during the monsoon period. On the basis of this, we then focus on analyzing the main mechanisms underlying these distributions (section 4)."*

4) Paragraph 3 of Introduction is important since it provides a transition from previous work to your work. However, the phrasing seems weak. If it is a first of its kind study conducted using the isotope analyzers over the continental China, then you should emphasize it, by all means. Readers would appreciate the new science and techniques that this work is doing.

Thank you very much for your suggestion. Now we place more emphasis on the novelty of our measurements (lines 107-111):

*"However, in situ observations documenting continuous spatial variations at the continental scale do not exist. This paper presents the first isotope dataset*

*documenting the spatial variations of vapor isotopes over a large continental region (order 10000 km) both during the pre-monsoon and monsoon periods, based on vehicle-based in-situ observations."*

(5) It is scientifically sound to not use the city datasets that may have been impacted by the traffic pollution. However, I am curious, if you can totally eliminate the role of water vapor emitted from country sources like irrigation, farms, power plants etc. in affecting the water vapor isotope or the humidity measurements in any way? If this is an assumption made, then please state it somewhere.

Thank you for your suggestion, we added (lines 233-235):

*"Outside towns, country sources, such as irrigation, farms, and power plants, cannot be completely ruled out. However, we expect their influence to be much smaller than large-scale spatial variations."*

(6) I have some concerns regarding the use of Iso-GSM for the disentanglement of seasonal and synoptic influences. From figures 11 and 12, the model does not seem to simulate the observations well, especially in the monsoon period. Moreover, the grid spacing for the models is way coarser compared to the observations. I would suggest using the observations to perform the same analysis discussed in 4.7, and only use model inputs where observations are not available. Then in table 2, compare the Iso-GSM vs observation-based fractions. This would be far more accurate. Since the main result of this paper is disentangling the seasonal vs synoptic effects, I think it is crucial to be thorough with your methods here.

We assessed the relative contribution using multi-year averages from 2010 to 2020 and a new estimation method that takes into account model bias and gives upper and lower bounds on the value of the contribution.

We modified (lines 327-361):

*"2.6 Method to decompose the observed daily variations*

*The temporal variations observed along the route for a given period represent a mixture of synoptic-scale perturbations, and of seasonal-mean spatial distribution:*

$$\delta^{18}O_{\_daily} = \delta^{18}O_{\_seaso} + \delta^{18}O_{\_synoptic} \qquad (4)$$

*The first term represents the contribution of seasonal-mean spatial variations, whereas the second term represents the contribution of synoptic-scale variations. Since these relative contributions are unknown, we use outputs from Iso-GSM. The daily variations of $\delta^{18}O$ simulated by Iso-GSM also represent a mixture of synoptic-scale perturbations and seasonal-mean spatial distribution, but with some errors relative to reality:*

$$\delta^{18}O_{\_daily\_Iso\text{-}GSM} = \delta^{18}O_{\_daily\_Iso\text{-}GSM} + \epsilon = (\delta^{18}O_{\_seaso\_Iso\text{-}GSM} + \epsilon_{\_seaso}) + (\delta^{18}O_{\_synoptic\_Iso\text{-}GSM} + \epsilon_{\_synoptic}) \qquad (5)$$

*where $\delta^{18}O_{\_daily\_Iso\text{-}GSM}$ is the daily outputs of $\delta^{18}O$ for each location, $\delta^{18}O_{\_seaso\_Iso\text{-}GSM}$ is the multi-year monthly outputs of $\delta^{18}O$ for each location, and $\delta^{18}O_{\_synoptic\_Iso\text{-}GSM} = \delta^{18}O_{\_daily\_Iso\text{-}GSM} - \delta^{18}O_{\_seaso\_Iso\text{-}GSM}$, $\epsilon_{\_seaso}$ and $\epsilon_{\_synoptic}$ are the errors on $\delta^{18}O_{\_seao\_Iso\text{-}GSM}$ and $\delta^{18}O_{\_synoptic\_Iso\text{-}GSM}$ relative to reality, respectively, $\epsilon$ is the sum of $\epsilon_{\_seaso}$ and $\epsilon_{\_synoptic}$.*

*These individual error components $\in_{\_seaso}$ and $\in_{\_synoptic}$ are unknown, but we know the sum of them ($\in$), i.e. the difference between daily outputs and observations. For the decomposition, we made two extreme assumptions to estimate upper and lower bounds on the contribution values:*

*1)We assume that the error is purely seasonal-mean, i.e. $\in = \in_{\_seaso}$, and $\in_{\_synoptic}=0$ :*

$$\delta^{18}O_{\_daily} = \delta^{18}O_{\_seaso\_Iso\text{-}GSM}+(\delta^{18}O_{\_synoptic\_Iso\text{-}GSM}+ \in) \qquad (6)$$

*To evaluate the contribution of these two terms, we calculate the slopes of $\delta^{18}O_{\_daily}$ as a function of $\delta^{18}O_{\_seaso\_Iso\text{-}GSM}$ ($a_{\_seaso}$), and of $\delta^{18}O_{\_daily}$- $\delta^{18}O_{\_seaso\_Iso\text{-}GSM}$ ($a_{\_synoptic}$). The relative contributions of spatial and synoptic variations correspond to $a_{\_seaso}$ and $a_{\_synoptic}$ respectively. This will be the upper bound for the contribution of synoptic-scale variations, since some of the systematic errors of Iso-GSM will be included in the synoptic component. This is equivalent to using the seasonal-mean of Iso-GSM and the raw time series of observations.*

*2) We assume that the error is purely synoptic, i.e. $\in = \in_{\_synoptic}$, and $\in_{\_seaso} =0$ :*

*Then $\delta^{18}O_{\_daily} = (\delta^{18}O_{\_seaso\_Iso\text{-}GSM}+ \in) + \delta^{18}O_{\_synoptic\_Iso\text{-}GSM}$. $\qquad (7)$*

*To evaluate the contribution of these two terms, we calculate the slopes of $\delta^{18}O_{\_daily\_Iso\text{-}GSM}$ as a function of $\delta^{18}O_{\_seaso\_Iso\text{-}GSM}$ ($a_{\_seaso}$), and of $\delta^{18}O_{\_daily\_Iso\text{-}GSM}$ - $\delta^{18}O_{\_seaso\_Iso\text{-}GSM}$ ($a_{\_synoptic}$). This will be the lower bound for the contribution of synoptic-scale variations, since we expect Iso-GSM to underestimate the synoptic variations.*

*The same analysis is also performed for the Iso-GSM simulation q (Table 2) and reanalysis q (Table 3)."*

We updated the discussion on disentangling seasonal-mean and synoptic variations in section 3.3 (lines 446-466):

" **Table 2** *The relative contribution ( in fraction) of spatial variations for a given season ($a_{\_seaso}$) and of synoptic-scale variations ($a_{\_synoptic}$) to the daily variations of q and $\delta^{18}O$ simulated by Iso-GSM. We checked that the sum of $a_{\_seaso}$ and $a_{\_synoptic}$ is always 1.*

| Period | variables | Controbutions | |
|---|---|---|---|
| | | $a_{\_seaso}$ | $a_{\_synoptic}$ |
| Pre-monsoon | q | 0.73~1.05 | 0.27~-0.05 |
| | $\delta^{18}O$ | 0.54~0.70 | 0.46~0.30 |
| Monsoon | q | 0.63~0.71 | 0.37~0.29 |
| | $\delta^{18}O$ | -0.50~0.10 | 1.5~0.9 |

**Table 3** *The same as Table 2, but for reanalysis q.*

| Period | variables | Controbutions | |
|---|---|---|---|
| | | $a_{\_seaso}$ | $a_{\_synoptic}$ |
| Pre-monsoon | q | 0.77~0.88 | 0.23~0.12 |
| Monsoon | q | 0.69~0.88 | 0.31~0.12 |

*During the pre-monsoon period, based on both the Iso-GSM simulation and NCEP/NCAR reanalysis, we can find that the seasonal-mean contribution to q is higher than the synoptic-scale contribution: a_seaso is 73%-105% for Iso-GSM and 77%-88% for reanalysis, whereas a_synoptic is 27%~ -5% for Iso-GSM and 23%~12% for reanalysis (Table 2 and Table3). The relative contribution of seasonal-mean spatial variations to the total simulated variations of $\delta^{18}O$ (54%-70%) is also much higher than that of synoptic-scale variations (27%- -5%). This suggests that the observed variability in q and $\delta^{18}O$ is mainly due to spatial variability, and marginally due to synoptic-scale variability. During the monsoon, seasonal-mean spatial variations is also the main contribution to the observed q variations. But it is different for $\delta^{18}O$: the relative contribution of synoptic-scale variations (90%-150%) dominates the total simulated variations of $\delta^{18}O$."*

Minor comments:
1) Line 76: Isn't Bailey et al. (2013) a Hawaii-based study? Are there other studies based on larger continental settings?

*We think we are the first to acquire large-scale in-situ observations. We modified as (lines 104-105):*

*"One study made vehicle-based in-situ observations to document spatial variations, but this was restricted to the Hawaii island (Bailey et al., 2013)."*

2) Section 2.1, 1st paragraph: Include continental recycling as well.

*We added the role of continental recycling (lines 72-74):*

*"Continental recycling, i.e. the moistening of the near-surface air by the evapo-transpiration from the land surface (transpiration by plants, evaporation of bare soil or standing water bodies, Brubaker et al 2013), is also an important source of water vapor in both seasons."*

4) Line 124: Maybe write a few lines about the advantage of using d-excess, how does d-excess vary in comparison with d18O.

*We added how people usually interpret d-excess (lines 141-144):*

*"The second-order d-excess parameter is computed based on the commonly used definition (Dansgaard, 1964). The d-excess is usually interpreted as reflecting the moisture source and evaporation conditions (Jouzel et al., 1997), since the d-excess is more sensitive to non-equilibrium fractionation occurs than $\delta^{18}O$".*

5) Line 146: What is the value of calibration humidity correction term f? Is equation 3 a standard equation for these calculations? Any references?

*We added the expiation and reference for f (lines 177-180) :*

*" f is the equation of $\delta$ as a function of humidity, and humidity is in ppm. E.g., if we measured that f is $\delta = a*\ln (humidity)+b$) by measuring standard water with different humidity, then the full equation for humidity-dependent isotope bias correction would be $\delta_{measured} - \delta_{humidity\ calibration} = a*\ln (humidity_{measured})+b - (a*\ln (20000)+b)$."*

We added following reference for f:

(Schmidt et al., 2010;JingfengLiu et al., 2014)

JingfengLiu, CundeXiao, MinghuDing, and JiawenRen: Variationsinstablehydrogenandoxygenisotopesinatmosphericwatervaporinthemarineboundarylayeracrossawidelatituderange, Journal of Environmental Sciences, 26, 2266-2276, 2014.

Schmidt, M., Maseyk, K., Lett, C., Biron, P., Richard, P., Bariac, T., and Seibt, U.: Concentration effects on laser-based $\delta^{18}O$ and $\delta^{2}H$ measurements and implications for the calibration of vapour measurements with liquid standards, Rapid Communications in Mass Spectrometry, 24, 3553-3561, 2010.

6) Figure 2,3: Have you tried using the same y-scale for pre-monsoon and monsoon period?

We tried, but it was too hard to see the spatial variations for the monsoon period.

7) Figure 2: Are the meteorological observations also 10-minute average?

We clarify the resolution of meteorological observations (Lines 241-242):

*"And all of them also had been averaged to a 10-min temporal resolution."*

What is the resolution of P-daily and P-mean respectively? It is not clear to me how P-mean is calculated. If P-mean is the temporal-mean of the precipitation amount over the sampling days, then P-mean should have a single value for each season. Please clarify. Also, mention the grid size of the GPCP precipitation. How is the grid average precipitation calculated?

We clarified the grid size (1-deg) of the GPCP in line 253.

P-mean is the average value over the entire observation period of about one month for each observation location, so it is not a single value for each season but for each position. We clarified the method to calculate the P-mean in lines 256-259:

*"When compare the time series of GPCP data with our observed isotopes, we linearly interpolate the daily GPCP data to the location of each observation location (P-daily).We also used the average of the GPCP precipitation over the entire observation period of about one month for each observation location (P-mean)."*

8) Is the humidity (figure 2e,f) and q (figure 2i,j) obtained from the Picarro sampler and the rooftop weather station, respectively? Do they correlate well? Can the humidity (ppm) also be expressed in (g/kg) for ease of comparison?

Yes, the humidity (figure 2e,f) and q (figure 2i,j) is obtained from the Picarro sampler and the rooftop weather station, respectively. We expressed the humidity (ppm) in (g/kg) and compare the q obtained from the Picarro, the rooftop weather station and NCAR in Figure 1. We added (lines 165-169):

*"The specific humidity measured by Picarro is close to that measured by an independent sensor installed in the vehicle (Fig.4). The correlation between the*

*humidity measured by the Picarro and the independent sensor are over 0.99, the slopes are approximately 1 and the average deviation are less than 1 g/kg both during pre-monsoon and monsoon periods."*

9) Line 249: "During the pre-monsoon…..than in any other regions". Here you say that

in the pre-monsoon period, the humidity, P-mean and d18O are higher in southwestern China than in other regions. However, since the data is not available for the entire pre-monsoon period in the northeast and northwest regions, are you assuming that the humidity, p-mean and d18O remain consistent for the entire pre-monsoon period? Have you estimated a sampling bias? It would be good to write this assumption too.

The time solution is 10min for humidity for each observation location. The P-mean is the average of the GPCP precipitation over the entire observation period of about one month for each observation location. The temporal 10-min average humidity when we measured have the same variation with the P-mean, so that we would like to show that the humidity we observed could reflect the information of seasonal mean precipitation. Since we have clarified the method to calculate the P-mean, I believe that this part should have been accurately expressed.

10) Line 277: Alternatively, the high d-excess in south China could also be coming

from the moisture flow from Indian/Pacific Ocean as is talked about later. Or, it could be resulting from the deeper convective mixed layer in south China compared to north China where vapor with high d-excess is transported towards the surface as shown in figure 9.

We added it as the potential reason for the high d-excess in south China (lines 412-414):

*"Alternatively, the high d-excess in south China could also result from the moisture flow from Indian/Pacific Ocean, or from the deeper convective mixed layer in south China compared to north China."*

11) Line 314: I find it exciting to see that the controls on d18O (temperature correlation)

and d-excess (precipitation-line correlation) can be differentiated so well just based on the observations alone.

Thank you. We highlighted this in the abstract by modifying lines 30-34 as:

*"During the pre-monsoon period, the spatial variations of vapor $\delta18O$ are mainly controlled by Rayleigh distillation along air mass trajectories. The North-South gradient observed during the pre-monsoon period is counteracted by different moisture sources, continental recycling processes and convection during moisture transport during the monsoon period."*

12) Change reference Noone and David, 2012 to Noone, 2012.

We have modified this reference.

13) Figure 6: Can you explain the reason for a significant number of WR_1 dried and depleted dots below Rayleigh curve? Are they within the uncertainty range of the Rayleigh curve?

Thank you for your suggestion. In current Figure 9, we added the uncertainty range of the Rayleigh curve calculated for different initial conditions of key moisture source regions: during March 2019, light red and light blue Rayleigh curve are calculated for key moisture source regions of westerlies ( $\delta^2H_0 = -168.04‰$, T=5°C) and BoB ($\delta^2H_0$= -77.37‰, T=26.46°C) separately in (a); during July-August 2018, ight red and light blue Rayleigh curve are calculated for key moisture source regions of westerlies ($\delta^2H_0$= -149.64‰, T=6.16°C) and BoB ($\delta^2H_0$= -82.75‰, T=27.69°C) separately in (b). These initial $\delta^2H$ are derived from iso-GSM, the initial temperature and RH are derived from NCAR/NCEP 2.5-deg global reanalysis data.

We added following discussion in lines 581-583:

"The observations in the WR_1 region (Fig.3c) are closer to the q-$\delta^2H$ Rayleigh distillation curve calculated for the key moisture source regions of westerlies, providing further evidence of the influence of water vapor source on vapor isotopes."

14) Since figure 10 is referred at multiple places in the paragraph starting at line 413, maybe change index of figure 10 to 7 to maintain the flow of the paper.

We moved Figure 10 to an earlier position in the manuscript (Section 2.4) due to the multiple references to the figure in the analysis and the fact that it is one of the results of the backward trajectory tracking, i.e., the meteorological data on the backward trajectory.

15) Line 439: replace 'bring' with 'brought'.

We have modified 'bring' to 'brought'.

16) Line 442: "As continental….". Wrong grammar.

We modified the grammar (lines 639-641):

*"As continental recycling is known to enrich the water vapor (Salati et al., 1979) and is associated with high d-excess (Gat and Matsui, 1991;Winnick et al., 2014)."*

17) Continental recycling from WR2, SR1 and SR3 is a strong factor influencing the isotope ratios. Perhaps, defining it in one line or so will help the readers.

We summarized the influence of continental recycling onWR2, SR1 and SR3 (lines 782-784):

*"Except SR_3 region, continental recycling also has a strong influence on isotopes in the WR2 and SR1 regions, which suggested by the high values of $\delta^{18}O$ and d-excess, back-trajectories, the location on the q-$\delta$ diagram, and the higher slopes and intercepts of $\delta^{18}O$-$\delta^2H$ relationship."*

18) Line 458: "…with a steepest slope….". Replace 'a' with 'the'.

We have modified "with a steepest slope" to "with the steepest slope".

19) Line 493: Not all observations exhibit temperature effect. WR3 does not.
    We modified "all observations" to "all observations taken together".

20) Line 509: Another factor for the dominating effect of q at least for SR2,3,4 can be rain evaporation.
    We incorporated this idea (lines 707-709):
    *"The absence of correlation with T suggests that the variations in q mainly reflect variations in relative humidity that are associated with different air mass origins or rain evaporation."*

21) Line 512: Explain the altitude effect in a separate line.
    We added the explanation of altitude effect when it first occur in lines 513-514:
    *"the altitude effect (the heavy isotope concentrations in fresh water decreasing with increasing altitude, Dansgaard 1954)"*

22) What is the difference between daily precipitation amount (P-daily) and temporal-mean precip amount for sampled dates (P-mean)?
    We clarified the definition and method to calculate the P-daily and P-mean in lines 256-259:
    *"When comparing the time series of GPCP data with our observations, we linearly interpolate the daily GPCP data to the location of each observation location (P-daily).We also used the average of the GPCP precipitation over the entire observation period of about one month for each observation location (P-mean)."*

23) Line 540: Define 'dtra'.
    We defined "dtra" in the figure description of Figure 9 (lines 729-732):
    *"The x-axis "dtra" represents the number of days prior to the observations (from 1 to 10 days). For example, when the number of days is 2, the correlations is calculated with the temporal mean of meteorological data along the air mass trajectories during the 2 days before the observations."*

24) Section 4.7: Going back to my major comment, why start the spatial vs synoptic comparison so late in 4.6 when you introduced the concept at the beginning of section 3? Tie this section with the previous sections.
    We re-organized the manuscript and put the spatial vs synoptic comparison as section 3.3.

25) Table 2 description is incomplete.
    We added "period" as the title of the 1st column, and "variables" as the title for the 2nd column (line 449).

26) I really like the summary plot 13. It provides a summary of all the meteorological processes described in the paper. Two questions: What is the significance of upward

and downward pointing triangles for d-excess?

We would like to use the upward and downward pointing triangles to represent high and low values of d-excess, different colours correspond to high and low relative humidity. The idea is that the high relative humidity source of water vapor leads to low values of d-excess and vice versa.

Please explain all the colored arrows in the figure description.

We explained all the colored arrows in the figure description (lines 852-857):

*"Colored arrows from red to blue represent the initial to subsequent extension of the Rayleigh distillation process along the water vapor trajectory, corresponding to high to low values of $\delta^{18}O$; green arrows represent high relative humidity and yellow arrows represent low relative humidity; orange twisted arrows represent continental recycling; blue-sized clouds represent strong and weak convective processes; triangle colours representing high and low values of d-excess correspond to the corresponding relative humidity."*

27) 'Bay of Bengal' spelt wrong in the figures 5 and 13

Thanks you, we corrected the spelling.

---

## Author Response (AR1)

**Response to Reviewers**

We appreciate your insightful comments and suggestions. These have greatly helped us refine our manuscript and improve the clarity of our key points. We also thank you for your positive feedback.

In this response letter, we have addressed each comment in detail below. Our responses are in blue, with specific changes to the text highlighted in *blue italics*. All line numbers in this document correspond to the line numbers in the updated version of the manuscript.

With best regards, Di WANG On behalf of all authors

**AC1: 'Comment on acp-2022-223', 27 Jun 2022**

Review of "Vehicle-based in-situ observations of the water vapour isotopic composition across China: spatial and seasonal distributions and controls" by Di Wang et al. submitted to ACP

This paper presents a very interesting and impressive dataset of vehicle-based stable water isotope measurements in China with several new results and interpretations about the drivers of the variability observed in different regions. The observations cover two seasons: the pre-monsoon season 2019 and the monsoon season 2018, which are compared in terms of their different dynamics and isotope signals recorded along the route. I recommend publication of the paper after two major and several minor comments have been addressed:

We appreciate your positive comment. We also appreciate your detailed suggestions.

**Major comments:**

A) Synoptic vs. seasonal variations: This analysis is very interesting. - However, it comes very late in the manuscript, even though the whole interpretation of the observations evolves along the main finding that the spatial (seasonal) variations dominate the variability of various isotope and meteorological variables. The results section even starts with referring to this analysis but asks the readers to postpone their curiosity to a much later section. Why not starting the results with what you show in Section 4.7?

We agree that the analysis of the seasonal vs synoptic should be earlier in the manuscript. We put it as section 3.3, just after the description of the raw data in section 3.1 and 3.2, and before the seasonal variations in section 3.4 and analyzing the factors controlling the spatial and seasonal distributions in section 4.

- What happens if you do the same regression analysis as presented in Section 4.7 using the actual observations for estimating the synoptic and seasonal components? Looking at the comparison of the simulation and the observations (Fig. 11), it strikes me that the daily model output follows the "temporal-mean" output much more closely than the actual measurements in both the premonsoon and the monsoon seasons. To me this shows that the Iso-GSM simulation on a relatively coarse grid is not capable of reproducing the observed mesoscale to large-scale variability in the water vapour isotope and meteorological fields. Therefore, in my view the finding that the isotope variability across China in the pre-monsoon and monsoon periods is mainly a result of seasonal/spatial variations and only marginally affected by synoptic-scale systems is biased towards what the Iso-GSM shows.

We accepted your suggestion and assessed the relative contribution using multiyear averages from 2015 to 2020 and a new estimation method that takes into account bias and gives upper and lower bounds on the value of the contribution. We also added IASI satellite measurements to estimate the contribution:

We modified (lines 331-370):

"2.6 Method to decompose the observed daily variations

The temporal variations observed along the route for a given period represent a mixture of synoptic-scale perturbations, and of seasonal-mean spatial distribution:

 $\delta^{2} \mathbf{H}_{\text{daily}} = \delta^{2} \mathbf{H}_{\text{seaso}} + \delta^{2} \mathbf{H}_{\text{synoptic}}$ (4)

The first term represents the contribution of seasonal-mean spatial variations, whereas the second term represents the contribution of synoptic-scale variations. Since these relative contributions are unknown, we use outputs from Iso-GSM and IASI. The daily variations of  $\delta^2$ H simulated by Iso-GSM also represent a mixture of synoptic-scale perturbations and seasonal-mean spatial distribution, but with some errors relative to reality:

 $\delta^{2}H_{\text{_daily\_Iso-GSM}} = \delta^{2}H_{\text{\_seaso\_Iso-GSM}} + \delta^{2}H_{\text{\_synoptic\_Iso-GSM}}$ (5)

where  $\delta^2 H_{daily\_Iso-GSM}$  is the daily outputs of  $\delta^2 H$  for each location,  $\delta^2 H_{\_seaso\_Iso-GSM}$  is the multi-year monthly outputs of  $\delta^2 H$  for each location, and  $\delta^2 H_{\_synoptic\_Iso-GSM} = \delta^2 H_{\_daily\_Iso-GSM} - \delta^2 H_{\_seaso\_Iso-GSM}$ , each of these terms are affected by errors relative to observations:

$$\delta^{2}H_{\text{_daily_Iso-GSM}} = \delta^{2}H_{\text{_daily}} + \epsilon = (\delta^{2}H_{\text{_seaso}} + \epsilon_{\text{_seaso}}) + (\delta^{2}H_{\text{_synoptic}} + \epsilon_{\text{_synoptic}})$$
(6)

where  $\in_{seaso}$  and  $\in_{synoptic}$  are the errors on  $\delta^2 H_{seao\_Iso-GSM}$  and  $\delta^2 H_{synoptic\_Iso-GSM}$  relative to reality, respectively,  $\in$  is the sum of  $\in_{seaso}$  and  $\in_{synoptic}$ .

Correspondingly,  $\delta^{2}H_{\text{daily}} = \delta^{2}H_{\text{daily}_{\text{ISO-GSM}}} - \epsilon = (\delta^{2}H_{\text{seaso}_{\text{ISO-GSM}}} - \epsilon_{\text{seaso}}) + (\delta^{2}H_{\text{synoptic}_{\text{ISO-GSM}}} - \epsilon_{\text{synoptic}})$  (7)

These individual error components  $\in_{seaso}$  and  $\in_{synoptic}$  are unknown, but we know the sum of them ( $\in$ ), i.e. the difference between daily outputs and observations. For the decomposition, we made two extreme assumptions to estimate upper and lower bounds on the contribution values:

- (1) If we assume that the error is purely synoptic, i.e. ∈ = ∈\_synoptic, and ∈\_seaso =0, then:
- $\delta^{2} \mathbf{H}_{\text{daily}} = \delta^{2} \mathbf{H}_{\text{seaso}_{\text{Iso-GSM}}} + (\delta^{2} \mathbf{H}_{\text{synoptic}_{\text{Iso-GSM}}} \epsilon)$ (8)

To evaluate the contribution of these two terms, we calculate the slopes of  $\delta^2 H_{daily}$  as a function of  $\delta^2 H_{seaso\_Iso-GSM}$  (a\_seaso ), and of  $\delta^2 H_{daily}$  -  $\delta^2 H_{seaso\_Iso-GSM}$  (a\_synoptic ). The relative contributions of spatial and synoptic variations correspond to a\_seaso and a\_synoptic respectively. This will be the upper bound for the contribution of synoptic-scale variations, since some of the systematic errors of Iso-GSM will be included in the synoptic component. This is equivalent to using the seasonal-mean of Iso-GSM and the raw time series of observations.

(2) If we assume that the error is purely seasonal-mean, i.e.  $\in = \in \_$  seaso, and  $\in \_$  synoptic =0, then:

 $\delta^{2} \mathbf{H}_{\text{daily}} = (\delta^{2} \mathbf{H}_{\text{seaso}_{\text{Iso-GSM}}} - \epsilon) + \delta^{2} \mathbf{H}_{\text{synoptic}_{\text{Iso-GSM}}}.$  (9)

To evaluate the contribution of these two terms, we calculate the slopes of  $\delta^2 H_{\_daily\_Iso-GSM}$  as a function of  $\delta^2 H_{\_seaso\_Iso-GSM} - \in (a_{\_seaso})$ , and of  $\delta^2 H_{\_daily} - (\delta^2 H_{\_seaso\_Iso-GSM} - \in)$  ( $a_{\_synoptic}$ ). This will be the lower bound for the contribution of synoptic-scale variations, since we expect Iso-GSM to underestimate the synoptic variations.

The same analysis is also performed for  $\delta^2$ H retrieved from IASI, and the Iso-GSM simulation q (Table 2) and reanalysis q (Table 3)."

We updated the discussion on disentangling seasonal-mean and synoptic variations in section 3.3 (lines 462-486):

"**Table 2** The relative contribution ( in fraction) of spatial variations for a given season ( $a_{seaso}$ ) and of synoptic-scale variations ( $a_{synopic}$ ) to the daily variations of qand  $\delta^2 H$  simulated by Iso-GSM. We checked that the sum of  $a_{seaso}$  and  $a_{synoptic}$  is always 1. The two values indicate the lower and upper bounds as calculated from equations 7 and 8.

| Period                    | Data                      | Variable     | Controbutions |               |            |
|---------------------------|---------------------------|--------------|----------------------|---------------|------------|
|                           |                           |              | a_seaso              | a_syn         | optic      |
| Pre-
monsoon
(2019) | Iso_GSM                   | q            | 0.73~1.02            | 2 0.27~-      | 0.02       |
|                           |                           | $\delta^2 H$ | 0.60 ~0.9            | 8 0.46~       | 0.09       |
|                           | IASI                      | $\delta^2 H$ | 1.06~0.94            | 4 -0.06~      | -0.06~0.06 |
| Monsoon
(2018)         | Iso_GSM                   | q            | 0.71~0.82            | 2 0.37~       | 0.18       |
|                           |                           | $\delta^2 H$ | 0.07~0.80            | 0 0.93~       | 0.20       |
|                           | IASI                      | $\delta^2 H$ | 0.53~0.84            | 4 0.47~       | 0.47~0.16  |
| Та                        | ble 3 The so              | ame as Tab   | le 2, but for r      | eanalysis q.  |            |
|                           | Period                    | Variables    | Contro               | Controbutions |            |
|                           |                           |              | a_seaso              | a_synoptic    |            |
| п                         | Pre-
10nsoon
(2019) | q            | 0.77~0.92            | 0.23~0.08     |            |
| 1                         | Monsoon
(2018)         | q            | 0.69~0.95            | 0.31~0.05     |            |

During the pre-monsoon period, based on both the Iso-GSM simulation and NCEP/NCAR reanalysis, we can find that the seasonal-mean contribution to the measured q is higher than the synoptic-scale contribution:  $a_{seaso}$  is 73%~102% from Iso-GSM and 77%~92% from reanalysis, whereas  $a_{synoptic}$  is 27% ~ -2% from Iso-

GSM and 23% ~ 8% from reanalysis (Table 2 and Table3). The relative contribution of seasonal-mean spatial variations to the total measured variations in  $\delta^2 H$  (60% ~ 98%) is also higher than that of synoptic-scale variations (46% ~ 9%). This suggests that the observed variability in q and  $\delta^2 H$  is mainly due to spatial variability, and marginally due to synoptic-scale variability. During the monsoon, seasonal-mean spatial variations are also the main contributions to the observed variations of q (a seaso is 71% ~ 82% from Iso-GSM and 69% ~ 95% from reanalysis, whereas a\_synoptic is 18% ~ 37% from Iso-GSM and 5% ~ 31% from reanalysis). Since Iso-GSM doesn't capture daily variations of  $\delta^2 H$  very well during the monsoon period, the relative contribution has a large threshold range (a\_seaso is 7%~80%, a\_synoptic is 20% ~ 93%) after accounting for the errors. Therefore, we can not conclude the dominate contribution on  $\delta^2 H$  from Iso-GSM outputs. IASI, which has a higher correlation with observations, provides an more credible range of  $a_{seaso}$  about 53% ~ 84%, and  $a_{synoptic}$  16% ~ 47%. These suggests that during the monsoon period, the synoptic contribution can be significant. Having understood the factors influencing the spatial and seasonal variation of vapor isotopes in section 4, we will be able to better understand the reasons for the inconsistent performance of Iso-GSM during the pre-monsoon and monsoon periods (in section 4.6)."

- What exactly is the "temporal-mean" output? A multi-year seasonal mean? Or the mean over the considered 2018 and 2019 seasons? And over which spatial scale did you average?

We modified "temporal-mean outputs" to "multi-year monthly-mean" and added a description in the manuscript (lines 317-320):

"For both dataset, we use the outputs corresponding to the observation location and the observation date (daily outputs), and the multi-year monthly-mean outputs (March monthly for the pre-monsoon period and August monthly for the monsoon period) for each observation location from 2015 to 2020"

- I am not convinced that you can conduct this analysis in a robust way, given the few observations you have for a given region and season. A discussion on this sampling issue of the seasonal signal from only few synoptic events would be beneficial. Still, it is a valuable analysis especially if it is done with the measurements and the model, comparing and discussing the two results.

We carried out this analysis on a spatial scale for a dense set of observations using the method in the response to comment 1. Comparison with multi-year averages reveals that during the pre-monsoon, spatial variability dominates for both q and  $\delta^2 H$ ; during the monsoon, q still mainly reflects seasonally averaged spatial variability, however,  $\delta^2 H$  is more influenced by synoptic-scale processes that may include different sources of water vapor, convective processes, etc., as discussed in the manuscript.

We added more discussion on the impact of possible synoptic-scale processes (lines 799-814):

"Among the synoptic influences, tropical cyclones, the Northern Summer Intra-Seasonal Oscillation (BSISO) and local processes probably played a role. For example, during our monsoon observations, landfall of tropical cyclones Jongdari and Yagi correspond to the low values of  $\delta^{18}O$  we observed in the eastern China (Fig.S7a). Bebinca corresponds to the low values of  $\delta^{18}O$  we observed in the southwestern China (Fig.S7a). Typhoons are known to be associated with depleted rain and vapor (Gedzelman et al 2003, Xu et al 2019, Battacharya et al 2022). Three Northern Summer Intra-Seasonal Oscillation (BSISO) events occurred in China during about July 28 -*31*, August 5 - 8th and August 14 – 16 (Fig S8). The northward propagation of the BSISO is associated with strong convection (Susskind et al., 2011;Kikuchi, 2021) (Fig. S8). Moreover, short-lived convective events that frequently occurred during our observation period (Wang et al., 2018). It is possible that these rapid high-frequency synoptic events are not fully captured by Iso-GSM. We expect that Iso-GSM captures the large-scale circulation. Yet, we notice that Iso-GSM underestimates the depletion associated with tropical cyclones (Fig. 12b). We hypothesize that given its coarse resolution, it underestimates the depletion associated with the meso-scale structure. This might contribute to the overestimation of vapor  $\delta^{18}O$  in southeastern China (*Fig.12b*). "

B) Organization of the manuscript: The reading of the manuscript would profit from a re-organization with a clearer separation of methodological aspects and results section. Many methodological aspects are in the results and interrupt the flow of the reader (regional analysis in Table 1, Urban emissions, methodological aspects on Eq. 4)

We accepted your suggestion, we have moved regional analysis in Table 1 to section 2.4, moved section "4.7 urban emissions" to section 2.2.6, and moved methodological aspects on Eq. 4 to section 2.5.

Minor comments:

1) L. 25: "large-scale (order 10000 km) continuous observations of near-surface vapor isotopes": can be misunderstood, reformulate. You made in-situ observations over a large area. Not continuous observations at multiple locations.

We have reworded as "in-situ observations of near-surface vapor isotopes over a large region" to avoid misunderstandings.

2) L. 28-29: "mainly due to spatial variations and marginally influenced by synoptic-scale variations": This is interesting! I was very curious about how you came to this conclusion, when starting to read your manuscript. When just reading the abstract, I found this statement and the following ones about the importance of Rayleigh distillation (cloud formation during large-scale ascent?), different moisture sources (variability induced by large-scale weather systems?), continental moisture recycling (transport regimes favouring oceanic vs. continental sources) and convection (mesoscale circulation) contradicting. First you write that the "spatial variations" dominate and then you mention different processes that are relevant at the synoptic- to meso- scale. The synoptic systems are very different in southern vs. northern China and these weather systems shape the spatial contrasts. It would be helpful if you could concisely state in the abstract how you come to the conclusion

that "seasonal"/"spatial" variations dominate the synoptic variability and think carefully about the best terminology to use.

We reworded the abstract to clarify the method (lines 29-33):

"Combined with daily and multi-year monthly mean outputs from the isotopeincorporated global spectral model (Iso-GSM) and IASI satellite to calculate the relative contribution, we found that the observed spatial variations in both periods represent mainly seasonal-mean spatial variations, but are influenced by more significant synoptic-scale variations during the monsoon period."

We have reworded the last paragraph of introduction to sort out the structure of the article (lines 112-119):

"After describing our observed time series along the route (section 3.1 and 3.2), we quantify the relative contributions of seasonal-mean spatial variations and synoptic-scale variations that locally disturb the seasonal-mean to our observed time series (section 3.3). We show that our observed variations in both seasons are dominated by spatial variations, but are influenced by significant synoptic-scale variations during the monsoon period. On the basis of this, we then focus on analysing the main mechanisms underlying these distributions (section 4). Collectively, these data and analyses provide refined understanding of how the interaction of the summer monsoon and westerly circulation control water isotope ratios in East Asia."

3) L. 38-39: Why is the performance of the Iso-GSM model weaker over the monsoon period?

We point out the main reasons for the different performance of Iso-GSM in the pre-monsoon and monsoon periods (lines 793-799):

"In section 3.3, we showed that Iso-GSM captured the isotopic variations during the pre-monsoon season better than during the monsoon season. We hypothesize that this mainly could be due to the larger contribution of synoptic-scale variations to the observed variations during the monsoon season. Iso-GSM performs well during the pre-monsoon season, when seasonal mean spatial variability dominates q and isotope. In contrast, it performs less well during the monsoon season, when isotopic variations are significantly influenced by the synoptic-scale variability."

4) L. 46: The first sentence of the introduction is a bit heavy, could you think of a more general motivation for your study? And think about how to guide a non-isotope specialist reader into your subject.

We have reworded the beginning of the introduction (lines 51-54):

"Stable water isotopes have been applied to study a wide range of hydrological and climatic processes (Gat, 1996;Bowen et al., 2019;West et al., 2009). This is because water isotopes vary with changes in the water phases (e.g., evaporation, condensation), and therefore produce a natural labeling effect within the global water cycle." 5) L. 54: Are only Tibetan plateau ice cores questioned in terms of their use as past temperature records?

We modified "isotopic records in Tibetan ice cores" to "isotopic records in ice cores from low and middle latitudes regions" (line59).

6) L. 56: "significant role of large-scale"

We have modified "the significant role of large regional atmospheric circulation" to "the significant role of large-scale atmospheric circulation".

7) L. 57: is a specific teleconnection mode meant here?

Based on suggestions from the other reviewer, we have reorganized the introduction and removed this sentence.

8) L. 74-77: Isn't the Bailey et al. 2013 study from an oceanic environment? Aemisegger et al. 2014 ACP discusses the isotope variability of continental evaporation induced by synoptic-scale variability.

We modified as (104-105): "One study made vehicle-based in-situ observations to document spatial variations, but this was restricted to the Hawaii island (Bailey et al., 2013)."

- 9) L. 97: Dry intrusions bring... cold and dry upper tropospheric airmasses We have modified "Dry intrusions" to "Westerlies" (line 69).
- 10) Fig. 1: the route is a bit difficult to make out from the otherwise very nice figure Thank you for your suggestion, we highlighted the observation routes.
- 11) L. 122: is computed We have modified "are computed" to "is computed" (line 142).

**12) L. 164: no drift in the standards: could you indicate the standard deviation of your series of standard measurements?**

We added the standard deviation of standard measurements during our campaigns (lines 198-201):

"For two standard waters, the standard deviation of standard measurements are 0.2‰ and 0.11‰ for  $\delta^{18}$ O, and 1.16‰ and 1.2‰ for  $\delta^{2}$ H during the pre-monsoon period of 2019. During the monsoon period of 2018, the standard deviation of standard measurements are 0.09‰ and 0.06‰ for  $\delta^{18}$ O, and 0.6‰ and 0.33‰ for  $\delta^{2}$ H."

**13) Section 2.2.3: should be humidity-dependent isotope bias correction**

We have modified "Humidity calibration" to "Humidity-dependent isotope bias correction" (line 164).

14) Was the specific humidity calibrated as well using an independent sensor? Thank you for your suggestions, we added (lines 166-170): "The specific humidity measured by Picarro is very close to that measured by an independent sensor installed in the vehicle (Fig.4). The correlation between the humidity measured by the Picarro and the independent sensor are over 0.99, the slopes are approximately 1 and the average deviation are less than 1 g/kg both during premonsoon and monsoon periods."

**15) L. 212: why was only one air parcel started from 1000 m above the surface?**

Tracing parcels starting from different altitudes would provide information on air mass mixing. However, due to the large number of our observation points, this would reduce the visibility of the backward trajectory results in Figure 2 and Figure 4. Therefore, we follow the conventional setup that people usually use in tracking near-surface water vapor sources, i.e., a tracking starting point of 1000 m from the ground (Guo et al., 2017;Bershaw et al., 2012),. We believe that this is representative of the backward trajectory of near-surface air, since most water vapor in the atmosphere is within 0–2 km above ground level (Wallace and Hobbs, 2006).

We added (lines 274-277):

"This is representative of the water vapor near the ground (Guo et al., 2017;Bershaw et al., 2012), since most water vapor in the atmosphere is within 0-2 km above ground level (Wallace and Hobbs, 2006)."

16) L. 229-234: This is a bit a difficult start of the results section. Could the analysis of the seasonal vs. synoptic drivers of isotope variability not come as a first result? It is otherwise very hard for the reader to follow the story.

We accepted your suggestion to place the analysis of seasonal versus synoptic drivers of isotopic variation earlier in the manuscript (section 3.3).

**17) L. 230: refer to Fig. 2a,b**

We have referred to these two figures in this sentence. As the order of the figures has changed, it is currently Figure 4 and Figure 5 (lines 373-374):

"Our survey of the vapor isotopes yields two snapshots of the isotopic distribution along the route (Fig.4 & Fig.5)."

18) L. 269: that could be the continental effect too... Should Dansgaard et al. 1964 be cited if you discuss the isotope effects? These different effects are not mutually exclusive, and in my opinion, they do not provide a direct mechanistic explanation for the correlation of the delta-values with temperature.

We cited Dansgaard et al. 1964 in the manuscript when we discuss the isotope effects:

Lines 687-690: "The  $\delta^{18}O$  is significantly anti-correlated with Alt in the SR\_4 region (r = -0.85, p < 0.05, Table S1), consistent with the "altitude effect" (the heavy isotope concentrations in fresh water decreasing with increasing altitude) in precipitation and water vapor (Dansgaard, 1964; Galewsky et al., 2016)."

*Lines* 667-668: "During the pre-monsoon period, all observations taken together exhibit a "temperature effect" (the  $\delta$ 's decreasing with temperature, Dansgaard 1964)"

We moved the discussion of the spatial variation of water vapor isotopes with temperature to Section 4. This discussion is further developed in the context of temperature data and correlation coefficients. In this section, we focus on the description of the observed data.

19) Section 3.2: I like the result about the fact that the monsoon counteracts the N-S gradient observed in the pre-Monsoon period very much. This could be included as a key result in the abstract.

We incorporated this great idea in the abstract, we modified lines 33-37 as:

"The spatial variations of vapor  $\delta^{18}O$  are mainly controlled by Rayleigh distillation along air mass trajectories during the pre-monsoon period, but are significantly influenced by different moisture sources, continental recycling processes and convection during moisture transport during the monsoon period. Thus, the North-South gradient observed during the pre-monsoon period is counteracted during the monsoon period."

20) L. 310-311: I don't understand this sentence. What is the separation line of the seasonal variation of dexcess? Reformulate.

We modified as (lines 506-508):

"The line separating the areas of positive and negative values of the dexcessmonsoon - d-excesspre-monsoon differences coincides with the 120 mm P-mean line (Fig.S2 f)."

21) P. 11-12: this is a methodological part and should not be in the results section.

We have now moved pages 11-12 to section 2.4 untitled "Back-trajectory calculation and categorizing regions based on air mass origin".

**22) L. 439: bring -> brought**

We have modified "bring" to "brought".

23) L. 442: replace "strong" by "high". Actually, the fraction of transpiration in evapotranspiration is the key factor determining d over the continent, when recycling is high (see Aemisegger et al. 2014, ACP).

We have modified "strong" to "high", and also added the reference for discussion (lines 614-616):

"These properties are associated with a high d-excess, consistent with strong continental recycling and by evapotranspiration (Aemisegger et al., 2014)."

**24) L. 369: reduction of humidity and water isotope ratios.**

We have modified "reduction of vapor isotope ratios" to "reduction of humidity and water isotope ratios" (line 539-540).

**25) Figure 6 is very interesting. Could it come earlier?**

We have tried many ways to reorganize the paper, but since following reviewers'

comment we have moved the section disentangling seasonal/synoptic variations earlier, and the moisture source categorization to the method section, this figure actually comes a bit later. However, it comes as the first figure of the discussion section 4, untitled "Understanding the factors controlling the spatial and seasonal distributions".

**26) L. 391: this sounds a bit speculative: couldn't it be another Rayleigh line or mixing?**

In current Figure 8, we added the uncertainty range of the Rayleigh curve calculated for different initial conditions of key moisture source regions: during March 2019, light red and light blue Rayleigh curve are calculated for key moisture source regions of westerlies ( $\delta^2 H_0 = -168.04\%$ , T=5°C) and BoB ( $\delta^2 H_0 = -77.37\%$ , T=26.46°C) separately in (a); during July-August 2018, light red and light blue Rayleigh curve are calculated for key moisture source regions of westerlies ( $\delta^2 H_0 = -149.64\%$ , T=6.16°C) and BoB ( $\delta^2 H_0 = -82.75\%$ , T=27.69°C) separately in (b). These initial  $\delta^2$ H are derived from Iso-GSM, the initial temperature and RH are derived from NCAR/NCEP 2.5-deg global reanalysis data.

We can find that observations in the WR\_3 region (Fig.3c) are truly located below the q- $\delta^2$ H Rayleigh distillation curve calculated for general values for pre-monsoon period and calculated for it's key moisture source region of BoB, we modified as (lines 545-548) :

"Observations below the Rayleigh line, even when considering the most depleted initial vapor conditions (light blue Rayleigh curve in Fig 9b), indicate the influence of rain evaporation from depleted precipitation (Noone, 2012; Worden et al., 2007)."

**27) L. 493: "all observations taken together" because WR2 and WR3 do not show a "temperature effect".**

We modified "all observations" to "all observations taken together" (line 667).

**28) L. 501: influenced**

We have modified "influence" to "influenced".

**29) L. 509: degree of rain out**

We modified "degrees along the Rayleigh distillation" to "degree of rain-out" (line 684).

30) L. 517: why does the dexcess usually increase with altitude? Many recent studies show that it is more complicated than that (see, Salmon et al. 2019 ACP, Thurnherr et al. 2021 WCD). In many cases a decrease of dexcess towards the mid troposphere is observed.

We agree that our observations of altitude variation within 2km are not sufficient to discuss the relationship between d-excess and altitude. We deleted the sentence "Alternatively, this may reflect the fact that the d-excess generally increases with altitude (Galewsky et al., 2016)".

We explain the positive correction between d-excess and altitude by the following sentence (lines 691-696): "The vapor d-excess for all observations during the monsoon

period (Fig.8d) is positively correlated with Alt (r = 0.39, p<0.05, Table S1). One possible reason is that the vapor d-excess is lower in coastal areas at lower altitudes, while at higher altitudes in the west, more recycling moisture leads to higher d-excess. The positive correlation between d-excess and altitude is consistent with previous studies in region (Acharya et al 2020)."

31) Fig. 9: I am not sure this analysis is very useful. Why is the correlation of a local observation with conditions of air parcels upstream relevant? To me only the correlation between dexcess and RH normalized to surface temperature upstream would make sense. This analysis could be left out and the corresponding text removed, it would make the paper more concise and highlight more the key findings.

We would like to keep this short discussion because many previous studies have shown that convection has a stronger impact on the vapor isotopic composition when integrated along trajectories than when considered locally in monsoon region. Therefore, we keep this analysis for two proxies of convection: precipitation (proxy for deep convection) and boundary layer mixing depth (proxy for both shallow and deep convection).

We added a few sentences to introduce the purpose of this section (lines 714-718):

"Reconstructions of paleoclimate using ice core isotopes have relied on relationships with local temperatures, but many previous studies have suggested that water isotopes are driven by remote processes along air mass trajectories. In particular, they emphasized the importance of upstream convection in controlling the isotopic composition of water vapor (Gao et al., 2013;He et al., 2015;Vimeux et al., 2005, Cai and Tian, 2016)"

We pointed out the relevance of the relationship between water vapor isotopes and temperature and humidity on water vapor transport pathways (lines 723-726) :

"The results show gradually increasing positive correlation coefficients as dtra changes from 10 to 3. This reflects the role of temperature and humidity along air mass trajectories and the large spatial and temporal coherence of T variations during the pre-monsoon period."

32) L. 552: I would rather say that convection transports moisture with high dexcess from lower altitudes to higher altitudes (see Thurnherr et al. 2021 WCD, Aemisegger et al. WCD).

Many studies documents an increase of d-excess with altitude in the free troposphere (Galewsky et al 2016, Salmon et al 2019), though the vertical profiles can be more complex in the boundary layer and just above it (Salmon et al 2010, Thurnerr et al 2021) or over oceanic regions in dry conditions (Aemisegger et al 2021). Vertical profiles measured in China (unpublished) confirm the increase of d-excess with altitude in our geographical setting.

33) L. 554: "deeper convection" instead of "more active convection" ?We modified "more active convection" to "deeper convection" (line 755).

**34) L. 593: non-equilibrium fractionation at the moisture source**

We modified "kinetic fractionation" to "non-equilibrium fractionation at the moisture source" (line 770).

35) L. 590-605: this is a bit repetitive, could be shortened We have abbreviated this paragraph.

36) L. 597: it's mainly the decrease in T, RH can be very low at the sea ice edge during cold air outbreaks (see, Aemisegger and Papritz, 2018 JC, Thurnherr et al. 2020, ACP)

We add following discussion (lines 774-778):

"This is consistent with the global-scale poleward decrease in T and increase in surface RH over the oceans (despite the occurrence of very low RH at the sea ice edge during cold air outbreaks (Aemisegger and Papritz, 2018, Thurnherr et al. 2020), resulting in global-scale poleward decrease in d-excess at mid-latitudes (Risi et al., 2013a; Bowen and Revenaugh, 2003)."

37) L. 610: what about the role of dew deposition during night over the continent in northern China? See Lee et al. 2021 HESS.

We agree that dew deposition during night could also contribute to the low dexcess over the continent in northern China. We add this reference (lines 778-779):

*"Alternatively, the low d-excess during the night over the continent in Northern China during the pre-monsoon could also have contributions (Li et al 2021)."*

38) L. 621: To me it seems that Iso-GSM strongly underestimates the synoptic timescale variability. What happens to your synoptic vs. seasonal drivers of variability if you use the observations as a measure of synoptic variability?

As in the response to the first major comment, we now use observations to estimate the relative contributions, giving the upper and lower bounds on the values of the contribution accounting for errors in Iso-GSM and IASI. We found that the synoptic contribution is significant during the monsoon season, but not dominant.

39) L. 633: Could you speculate about the processes that are responsible for the model biases? Could it be linked to the representation of convection? Or evapotranspiration? We added the interpretion for the bias (lines 826-829):

"The differences in  $\delta^{18}O$  (Fig.12a) and q (Fig.12c) are spatially consistent. The overestimation of  $\delta^{18}O$  therefore could be due to the overestimation of q, and vice versa. These biases could be associated with deviations in the representation of convection or in continental recycling."

40) L. 633/634: I was slightly confused by the writing here: "overestimation (respectively underestimation)...", could this be written more clearly? Do you mean biases affecting both  $^{18}$ O and q?

What I would like to explain by this is that regions that overestimate q also

overestimate  $\delta^{18}$ O, and conversely, regions that underestimate q also underestimate  $\delta^{18}$ O. I modified it (lines 827):

"The overestimation of  $\delta^{18}O$  therefore could be due to the overestimation of q, and vice versa."

**41) L. 649: is the strongly smoothed topography in Iso-GSM the reason behind the bad representation of the altitude effect?**

We corrected for the altitude difference between observations and Iso-GSM using the method given in III. Supplementary Text. We have the same results. Therefore, we think that the smoothed topography is not the main reason.

**42) L. 664: what is the temporal-mean output of $^{18}$ O from Iso-GSM? A multi-year seasonal mean? Or just the seasonal mean of that particular year?**

We have made the definition clearer in the manuscript (lines 317-320):

"For both dataset, we use the outputs corresponding to the observation location and the observation date (daily outputs), and the multi-year monthly-mean outputs (March monthly for the pre-monsoon period and August monthly for the monsoon period) for each observation location from 2015 to 2020."

43) Section 4.7: If possible, this should come much earlier in the manuscript to guide the reader in the interpretation of the observations. Also as mentioned in the major comments, I like the analysis, but I doubt that the number of observations in each region is large enough (sample size) to robustly characterize the synoptic variability encountered in that region. I also suspect that Iso-GSM underestimates the residual 18Osynoptic= 18Odaily- 18Oseasonal. A critical discussion of the results would be very beneficial here.

We have placed the analysis of seasonal versus synoptic drivers of isotopic variation earlier in the manuscript (in section 3.3).

We have now improved our decomposition method to account for errors in Iso-GSM and IASI for both seasonal and synoptic-scale variations. Taking into account these errors, we slightly modify one of our conclusions by noting the great influence of synoptic-scale variability during the monsoon period.

44) Urban emissions: this is an interesting side-discussion. I would however rather see this as a methodological paragraph.

We accepted your suggestion and moved this section to data and method section (section 2.2.6 Data processing).

Also, I strongly recommend discussing the potentially important baseline effects emerging from rapid changes in concentrations of different trace gases, which can lead to biases in the isotope observations from CRDS systems (see Johnson and Rella, 2017 AMT, Gralher et al., 2016).

We added a short discussion about the potentially important baseline effects emerging from rapid changes in concentrations of different trace gases (lines 229-231): *"Some of these d-excess anomalies are not excluded from being affected by the*

**baseline effects emerging from rapid changes in concentrations of different trace gases. (Johnson and Rella, 2017; Gralher et al., 2016)"**

Technical comments: - The isotope observational dataset should also be made available online. It is only available upon request according to the data availability statement. A thorough documentation of the data (with adequate metadata description) is key, for making the data accessible to other scientists.

Thank you for your reminder and detailed suggestions. We had submitted the isotope observational dataset to the PANGEA repository. We also described the data in detail and indicate the time of observation. However, publishing data in this database requires a long review period. Therefore the data was not available to readers online at the time we submitted this article. The datasets are now available.

We modified the data availability section as (lines 905-911):

"The data acquired during the field campaigns used can be accessed via the following link or DOI: (1) Wang, Di; Tian, Lide (2022): Vehicle-based in-situ observations of the water vapor isotopic composition across China during the monsoon season 2018. PANGAEA, https://doi.org/10.1594/PANGAEA.947606; (2)Wang, Di; Tian, Lide (2022): Vehicle-based in-situ observations of the water vapor isotopic composition across China during the pre-monsoon season 2019. PANGAEA, https://doi.org/10.1594/PANGAEA.947627."

- All figures showing observational data: it would be nice to add the corresponding year in the captions (2019 & 2018).

We accepted your suggestion and add the corresponding year in the captions (2019 & 2018).

**AC2: 'Comment on acp-2022-223', 30 Sep 2022**

Review of "Vehicle-based in-situ observations of the water vapour isotopic composition

across China: spatial and seasonal distributions and controls" by Di Wang et al. submitted to ACP

**Summary:**

This paper serves really well in characterizing the seasonal and spatial variability over a broad region of China over the pre-monsoon and monsoon periods using a vehiclebased isotope analyzer and other meteorological measurements. The synoptic effects from the continents and oceans along with their entanglement with seasonal characteristics are thoroughly investigated using water vapor isotope observations. This paper would be a good fit for ACP publication after the authors address some of the outstanding major and few minor issues.

Thank you very much for reviewing our paper. We appreciate your positive feedback and detailed suggestions.

Major comments:

1) The start of the introduction is strong but it still needs to incorporate a discussion about the current knowledge of synoptic and seasonal processes within the continental region. Why is it important and what do the previous studies broadly say? What is the isotope perspective adding to the traditional methods of study? An expanded version of the first paragraph of Section 2.1 is more fitting for the introduction in my opinion.

Thank you for your suggestions to make our introduction section clearer. We have integrated the first paragraph of section 2.1 into the introduction, and added the introduction about the seasonal and synoptic-scale variations (lines 68-90):

"In general, large parts of the country are affected by the Indian monsoon and the East Asian monsoon in summer, which bring humid marine moisture from the Indian Ocean, South China Sea, and Northwestern Pacific Ocean (Fig.1). During the nonmonsoon seasons, the Westerlies influence most of northern China (Fig.1). Westerlies brings extremely cold and dry air masses. Occasional moisture flow from the Indian Ocean and/or Pacific Ocean brings moisture to southern China. Continental recycling, *i.e.* the moistening of the near-surface air by the evapo-transpiration from the land surface (transpiration by plants, evaporation of bare soil or standing water bodies, Brubaker et al 2013), is also an important source of water vapor in both seasons. Some of the spatial and seasonal patterns of water vapor transport are imprinted in the observed station-based precipitation isotopes (Araguás-Araguás et al., 1998; Tian et al., 2007; Wright, 1993; Mei'e et al., 1985; Tan, 2014). However, precipitation isotopes can only be obtained at a limited number of stations and on rainy days. The lack of information makes it limited to analyze the effects of water vapor propagation and alternating monsoon and westerlies. In particular, the seasonal pattern and the spatial variation of water isotopes can strongly influenced by synoptic-scale processes,

through their influence on moisture source, transport, convection and mixing processes (Wang et al., 2021;Sánchez-Murillo et al., 2019;Klein et al., 2015), which requires higher frequency observations. For example, some studies founded the impact of tropical cyclones (Gedzelman et al 2003, Xu et al 2019, Battacharya et al 2022) the Northern Summer Intra-Seasonal Oscillation (BSISO) (Susskind et al., 2011;Kikuchi, 2021), local or large-scale convections (Shi et al., 2020), cold front passages (Aemisegger et al., 2015), depressions(Saranya et al., 2018), and anticyclones (Khaykin et al., 2022) on water isotopes in the Asian region. Additional data and analysis refining our understanding of controls on the spatial and temporal variation of water isotopes in low-latitude regions therefore are needed."

2) Since synoptic vs seasonal influences on the isotope observations is the major conclusion of this paper (it literally is in the title of the manuscript), it would help to talk and explain these influences in the introduction. Mention all the factors that you are examining for the two categories. I see that you have classified them in the abstract but it would serve the readers well to know about them from early on. Are there earlier studies that you can compare your results with?

We introduced the seasonal and synoptic-scale variations and the factors that we are examining for the two categories. (Same lines as for the last response. Because the responses to the first and second advice are interrelated, they are not listed separately).

Also, whenever you use terms like "altitude effect" and "continental-recycling", use a line or two to give the readers a gist. This would make it convenient for the readers lest they need a jolt to their memory or are not acquainted with the terms well.

We added the gist for "altitude effect", "continental-recycling" and "temperature effect":

Lines 687-690: "The  $\delta^{18}O$  is significantly anti-correlated with Alt in the SR\_4 region (r = -0.85, p < 0.05, Table S1), consistent with the "altitude effect" (the heavy isotope concentrations in fresh water decreasing with increasing altitude) in precipitation and water vapor (Dansgaard, 1964; Galewsky et al., 2016).".

Lines 667-668: "During the pre-monsoon period, all observations taken together exhibit a "temperature effect" (the  $\delta$ 's decreasing with temperature, Dansgaard 1964)".

Lines 73-75: "Continental recycling, i.e. the moistening of the near-surface air by the evapo-transpiration sfrom the land surface (transpiration by plants, evaporation of bare soil or standing water bodies, Brubaker et al 2013)".

3) Another point regarding the synoptic vs seasonal comparison is that although this is the focal point of the paper, and it is only talked about formally in section 4.7. This leaves the readers hanging for the major part of results and discussions. I understand that you talk about the seasonal effects followed by the synoptic effects individually, and then you use Iso-GSM to compare the two effects. However, for more clarity, once you end discussing the seasonal/synoptic effects, write a

summary sentence or two to tie up the results to the final seasonal vs synoptic comparison. This will keep the audience' attention till they reach section 4.7. No major editing is necessary. Just adding/shuffling some summary sentences at the end of subsections of sections 3 and 4 would do.

We agree that the analysis of the seasonal vs synoptic is important to interpret the observed variations. We now moved this section earlier in the manuscript. We put it as section 3.3, just after the description of the raw data in section 3.1 and 3.2, on this basis, and before the seasonal variations in section 3.4 and analyzing the factors controlling the spatial and seasonal distributions in section 4.

For our main results, we also focus on a better understanding the factors controlling the spatial and seasonal distributions, not only the relative contribution of seasonal-mean and synoptic variations. Synoptic vs seasonal influences is part of result and a basis for the analysis of controlling factors.

To clarify this we have reworded the last paragraph of introduction (lines 112-119):

"After describing our observed time series along the route (section 3.1 and 3.2), we quantify the relative contributions of seasonal-mean spatial variations and synopticscale variations that locally disturb the seasonal-mean to our observed time series (section 3.3). We show that our observed variations in both seasons are dominated by spatial variations, but are influenced by significant synoptic-scale variations during the monsoon period. On the basis of this, we then focus on analyzing the main mechanisms underlying these distributions (section 4)."

4) Paragraph 3 of Introduction is important since it provides a transition from previous work to your work. However, the phrasing seems weak. If it is a first of its kind study conducted using the isotope analyzers over the continental China, then you should emphasize it, by all means. Readers would appreciate the new science and techniques that this work is doing.

Thank you very much for your suggestion. Now we place more emphasis on the novelty of our measurements (lines 107-111):

"However, in situ observations documenting continuous spatial variations at the continental scale do not exist. This paper presents the first isotope dataset documenting the spatial variations of vapor isotopes over a large continental region (over 10000 km) both during the pre-monsoon and monsoon periods, based on vehicle-based in-situ observations."

(5) It is scientifically sound to not use the city datasets that may have been impacted by the traffic pollution. However, I am curious, if you can totally eliminate the role of water vapor emitted from country sources like irrigation, farms, power plants etc. in affecting the water vapor isotope or the humidity measurements in any way? If this is an assumption made, then please state it somewhere.

Thank you for your suggestion, we added (lines 234-236):

"Outside towns, country sources, such as irrigation, farms, and power plants, cannot be completely ruled out. However, we expect their influence to be much

**smaller than large-scale spatial variations."**

(6) I have some concerns regarding the use of Iso-GSM for the disentanglement of seasonal and synoptic influences. From figures 11 and 12, the model does not seem to simulate the observations well, especially in the monsoon period. Moreover, the grid spacing for the models is way coarser compared to the observations. I would suggest using the observations to perform the same analysis discussed in 4.7, and only use model inputs where observations are not available. Then in table 2, compare the Iso-GSM vs observation-based fractions. This would be far more accurate. Since the main result of this paper is disentangling the seasonal vs synoptic effects, I think it is crucial to be thorough with your methods here.

We assessed the relative contribution using multi-year averages from 2015 to 2020 and a new estimation method that takes into account bias and gives upper and lower bounds on the value of the contribution. We also added IASI satellite measurements to estimate the contribution:

We modified (lines 331-370):

"2.6 Method to decompose the observed daily variations

The temporal variations observed along the route for a given period represent a mixture of synoptic-scale perturbations, and of seasonal-mean spatial distribution:

 $\delta^{2}H_{\text{_daily}} = \delta^{2}H_{\text{_seaso}} + \delta^{2}H_{\text{_synoptic}}$ (4)

The first term represents the contribution of seasonal-mean spatial variations, whereas the second term represents the contribution of synoptic-scale variations. Since these relative contributions are unknown, we use outputs from Iso-GSM and IASI. The daily variations of  $\delta^2$ H simulated by Iso-GSM also represent a mixture of synoptic-scale perturbations and seasonal-mean spatial distribution, but with some errors relative to reality:

 $\delta^{2} H_{\text{daily}_{\text{Iso-GSM}}} = \delta^{2} H_{\text{seaso}_{\text{Iso-GSM}}} + \delta^{2} H_{\text{synoptic}_{\text{Iso-GSM}}}$ (5)

where  $\delta^2 H_{daily\_Iso-GSM}$  is the daily outputs of  $\delta^2 H$  for each location,  $\delta^2 H_{seaso\_Iso-GSM}$  is the multi-year monthly outputs of  $\delta^2 H$  for each location, and  $\delta^2 H_{synoptic\_Iso-GSM} = \delta^2 H_{daily\_Iso-GSM} - \delta^2 H_{seaso\_Iso-GSM}$ , each of these terms are affected by errors relative to observations:

$$\delta^{2}H_{\text{_daily_Iso-GSM}} = \delta^{2}H_{\text{_daily}} + \epsilon = (\delta^{2}H_{\text{_seaso}} + \epsilon_{\text{_seaso}}) + (\delta^{2}H_{\text{_synoptic}} + \epsilon_{\text{_synoptic}})$$
(6)

where  $\in_{seaso}$  and  $\in_{synoptic}$  are the errors on  $\delta^2 H_{seao\_Iso-GSM}$  and  $\delta^2 H_{synoptic\_Iso-GSM}$  relative to reality, respectively,  $\in$  is the sum of  $\in_{seaso}$  and  $\in_{synoptic}$ .

Correspondingly,  $\delta^{2}H_{\text{daily}} = \delta^{2}H_{\text{daily}_{\text{ISO-GSM}}} - \epsilon = (\delta^{2}H_{\text{seaso}_{\text{ISO-GSM}}} - \epsilon_{\text{seaso}}) + (\delta^{2}H_{\text{synoptic}_{\text{ISO-GSM}}} - \epsilon_{\text{synoptic}})$  (7)

These individual error components  $\in_{seaso}$  and  $\in_{synoptic}$  are unknown, but we know the sum of them ( $\in$ ), i.e. the difference between daily outputs and observations. For the decomposition, we made two extreme assumptions to estimate upper and lower bounds on the contribution values:

(2) If we assume that the error is purely synoptic, i.e. ∈ = ∈\_synoptic, and ∈\_seaso =0, then:

 $\delta^{2} \mathbf{H}_{\text{daily}} = \delta^{2} \mathbf{H}_{\text{seaso}_{\text{ISO-GSM}}} + (\delta^{2} \mathbf{H}_{\text{synoptic}_{\text{ISO-GSM}}} - \epsilon)$ (8)

To evaluate the contribution of these two terms, we calculate the slopes of  $\delta^2 H_{daily}$ as a function of  $\delta^2 H_{seaso_Iso-GSM}$  (a\_seaso ), and of  $\delta^2 H_{daily}$  -  $\delta^2 H_{seaso_Iso-GSM}$  (a\_synoptic ). The relative contributions of spatial and synoptic variations correspond to a\_seaso and a\_synoptic respectively. This will be the upper bound for the contribution of synoptic-scale variations, since some of the systematic errors of Iso-GSM will be included in the synoptic component. This is equivalent to using the seasonal-mean of Iso-GSM and the raw time series of observations.

(2) If we assume that the error is purely seasonal-mean, i.e.  $\in = \in \_$  seaso, and  $\in \_$  synoptic =0, then:

 $\delta^{2}H_{\text{daily}} = (\delta^{2}H_{\text{seaso}_{\text{Iso-GSM}}} - \epsilon) + \delta^{2}H_{\text{synoptic}_{\text{Iso-GSM}}}.$ (9)

To evaluate the contribution of these two terms, we calculate the slopes of  $\delta^2 H_{\_daily\_Iso-GSM}$  as a function of  $\delta^2 H_{\_seaso\_Iso-GSM} - \in (a_{\_seaso})$ , and of  $\delta^2 H_{\_daily} - (\delta^2 H_{\_seaso\_Iso-GSM} - \in)$  (a\_synoptic ). This will be the lower bound for the contribution of synoptic-scale variations, since we expect Iso-GSM to underestimate the synoptic variations.

The same analysis is also performed for  $\delta^2$ H retrieved from IASI, and the Iso-GSM simulation q (Table 2) and reanalysis q (Table 3)."

We updated the discussion on disentangling seasonal-mean and synoptic variations in section 3.3 (lines 462-486):

"**Table 2** The relative contribution ( in fraction) of spatial variations for a given season (a\_seaso) and of synoptic-scale variations (a\_synopic) to the daily variations of q and  $\delta^2$ H simulated by Iso-GSM. We checked that the sum of a\_seaso and a\_synoptic is always 1. The two values indicate the lower and upper bounds as calculated from equations 8 and 9.

| Perio                                              | d       | Data                     | Variable     | es (            | Controbutions        |       |  |  |  |  |
|----------------------------------------------------|---------|--------------------------|--------------|-----------------|----------------------|-------|--|--|--|--|
|                                                    |         |                          |              | a_seaso         | a_sync               | optic |  |  |  |  |
| Pre-
monsoon
(2019)                          |         | Inc. CSM                 | r q          | 0.73~1.0        | 2 0.27~-             | 0.02  |  |  |  |  |
|                                                    | on      | 180_051                  | $\delta^2 H$ | 0.60 ~0.9       | 0.40~0               | 0.02  |  |  |  |  |
|                                                    | )       | IASI                     | $\delta^2 H$ | 1.06~0.9        | 1.06~0.94 -0.06~0.06 |       |  |  |  |  |
| Monsoon
(2018)                                  |         |                          | r q          | 0.71~0.8        | 2 0.29~(             | 0.18  |  |  |  |  |
|                                                    | 180_GSN | $\delta^2 H$             | 0.09~0.8     | 0.09~0.87 0.91~ |                      |       |  |  |  |  |
|                                                    | 0)      | IASI                     | $\delta^2 H$ | 0.53~0.8        | 4 0.47~(             | 0.16  |  |  |  |  |
| Table 3 The same as Table 2, but for reanalysis q. |         |                          |              |                 |                      |       |  |  |  |  |
|                                                    | Period  |                          | Variables    | Contro          | Controbutions        |       |  |  |  |  |
|                                                    |         |                          |              | a_seaso         | a_synoptic           |       |  |  |  |  |
|                                                    | m
(  | Pre-
onsoon
(2019) | q            | 0.77~0.92       | 0.23~0.08            |       |  |  |  |  |
|                                                    |         | Ionsoon
(2018)        | q            | 0.69~0.95       | 0.31~0.05            |       |  |  |  |  |

During the pre-monsoon period, based on both the Iso-GSM simulation and NCEP/NCAR reanalysis, we can find that the seasonal-mean contribution to the measured q is higher than the synoptic-scale contribution:  $a_{seaso}$  is 73%~102% from

Iso-GSM and 77%~92% from reanalysis, whereas a synoptic is 27% ~ -2% from Iso-GSM and 23% ~ 8% from reanalysis (Table 2 and Table 3). The relative contribution of seasonal-mean spatial variations to the total measured variations in  $\delta^2 H$  (60% ~ 98%) is also higher than that of synoptic-scale variations (40%  $\sim$ 2%). This suggests that the observed variability in q and  $\delta^2$ H is mainly due to spatial variability, and marginally due to synoptic-scale variability. During the monsoon, seasonal-mean spatial variations are also the main contributions to the observed variations of q ( $a_{seaso}$  is 71% ~ 82% from Iso-GSM and 69% ~ 95% from reanalysis, whereas  $a_{synoptic}$  is 18% ~ 29% from Iso-GSM and 5% ~ 31% from reanalysis). Since Iso-GSM doesn't capture daily variations of  $\delta^2$ H very well during the monsoon period, the relative contribution has a large threshold range (a seaso is 9%~87%, a synoptic is 91% ~ 13%) after accounting for the errors. Therefore, we can not conclude the dominate contribution on  $\delta^2$ H from Iso-GSM outputs. IASI, which has a higher correlation with observations, provides an more credible range of  $a_{seaso}$  about 53% ~ 84%, and  $a_{synoptic}$  16% ~ 47%. These suggests that during the monsoon period, the synoptic contribution can be significant, but not dominate. Having understood the factors influencing the spatial and seasonal variation of vapor isotopes in section 4, we will be able to better understand the reasons for the inconsistent performance of Iso-GSM during the pre-monsoon and monsoon periods (in section 4.6)."

**Minor comments:**

1) Line 76: Isn't Bailey et al. (2013) a Hawaii-based study? Are there other studies based on larger continental settings?

We think we are the first to acquire large-scale in-situ observations. We modified as (lines 104-105):

"One study made vehicle-based in-situ observations to document spatial variations, but this was restricted to the Hawaii island (Bailey et al., 2013)."

**2) Section 2.1, 1st paragraph: Include continental recycling as well.**

We added the role of continental recycling (lines 73-76):

"Continental recycling, i.e. the moistening of the near-surface air by the evapotranspiration from the land surface (transpiration by plants, evaporation of bare soil or standing water bodies, Brubaker et al 2013), is also an important source of water vapor in both seasons."

**4) Line 124: Maybe write a few lines about the advantage of using d-excess, how does d-excess vary in comparison with d18O.**

We added how people usually interpret d-excess (lines 142-145):

"The second-order d-excess parameter is computed based on the commonly used definition (Dansgaard, 1964). The d-excess is usually interpreted as reflecting the moisture source and evaporation conditions (Jouzel et al., 1997), since the d-excess is more sensitive to non-equilibrium fractionation occurs than  $\delta^{18}O$ ".

5) Line 146: What is the value of calibration humidity correction term f? Is equation 3

**a standard equation for these calculations? Any references?**

We added the expiation and reference for f (lines 178-181) :

"f is the equation of  $\delta$  as a function of humidity, and humidity is in ppm. E.g., if we measured that f is  $\delta = a*\ln(\text{humidity})+b$ ) by measuring standard water with different humidity, then the full equation for humidity-dependent isotope bias correction would be  $\delta_{\text{measured}} - \delta_{\text{humidity calibration}} = a*\ln(\text{humidity}_{\text{measured}})+b - (a*\ln(20000)+b)$ ."

We added following reference for f:

(Schmidt et al., 2010;JingfengLiu et al., 2014)

JingfengLiu, CundeXiao, MinghuDing, and JiawenRen: Variationsinstablehydrogenandoxygenisotopesinatmosphericwatervaporinthemarineb oundarylayeracrossawidelatituderange, Journal of Environmental Sciences, 26, 2266-2276, 2014.

Schmidt, M., Maseyk, K., Lett, C., Biron, P., Richard, P., Bariac, T., and Seibt, U.: Concentration effects on laser-based  $\delta^{18}$ O and  $\delta^{2}$ H measurements and implications for the calibration of vapour measurements with liquid standards, Rapid Communications in Mass Spectrometry, 24, 3553-3561, 2010.

6) Figure 2,3: Have you tried using the same y-scale for pre-monsoon and monsoon period?

We tried, but it was too hard to see the spatial variations for the monsoon period.

**7) Figure 2: Are the meteorological observations also 10-minute average?**

We clarify the resolution of meteorological observations (Lines 242-243): *"And all of them also had been averaged to a 10-min temporal resolution."*

What is the resolution of P-daily and P-mean respectively? It is not clear to me how P-mean is calculated. If P-mean is the temporal-mean of the precipitation amount over the sampling days, then P-mean should have a single value for each season. Please clarify. Also, mention the grid size of the GPCP precipitation. How is the grid average precipitation calculated?

We clarified the grid size (1-deg) of the GPCP in line 253.

P-mean is the average value over the entire observation period of about one month for each observation location, so it is not a single value for each season but for each position. We clarified the method to calculate the P-mean in lines 256-259:

"When compare the time series of GPCP data with our observed isotopes, we linearly interpolate the daily GPCP data to the location of each observation location (P-daily).We also used the average of the GPCP precipitation over the entire observation period of about one month for each observation location (P-mean)."

8) Is the humidity (figure 2e,f) and q (figure 2i,j) obtained from the Picarro sampler and the rooftop weather station, respectively? Do they correlate well? Can the humidity (ppm) also be expressed in (g/kg) for ease of comparison?

Yes, the humidity (figure 2e,f) and q (figure 2i,j) is obtained from the Picarro

sampler and the rooftop weather station, respectively. We expressed the humidity (ppm) in (g/kg) and compare the q obtained from the Picarro, the rooftop weather station and NCAR in Figure 1. We added (lines 166-170):

"The specific humidity measured by Picarro is close to that measured by an independent sensor installed in the vehicle (Fig.4). The correlation between the humidity measured by the Picarro and the independent sensor are over 0.99, the slopes are approximately 1 and the average deviation are less than 1 g/kg both during premonsoon and monsoon periods."

9) Line 249: "During the pre-monsoon.....than in any other regions". Here you say that in the pre-monsoon period, the humidity, P-mean and d18O are higher in southwestern China than in other regions. However, since the data is not available for the entire pre-monsoon period in the northeast and northwest regions, are you assuming that the humidity, p-mean and d18O remain consistent for the entire pre-monsoon period? Have you estimated a sampling bias? It would be good to write this assumption too.

The time solution is 10min for humidity for each observation location. The P-mean is the average of the GPCP precipitation over the entire observation period of about one month for each observation location. The temporal 10-min average humidity when we measured have the same variation with the P-mean, so that we would like to show that the humidity we observed could reflect the information of seasonal mean precipitation. Since we have clarified the method to calculate the P-mean, I believe that this part should have been accurately expressed.

10) Line 277: Alternatively, the high d-excess in south China could also be coming from the moisture flow from Indian/Pacific Ocean as is talked about later. Or, it could be resulting from the deeper convective mixed layer in south China compared to north China where vapor with high d-excess is transported towards the surface as shown in figure 9.

We added it as the potential reason for the high d-excess in south China (lines 417-419):

"Alternatively, the high d-excess in south China could also result from the moisture flow from Indian/Pacific Ocean, or from the deeper convective mixed layer in south China compared to north China."

11) Line 314: I find it exciting to see that the controls on d18O (temperature correlation) and d-excess (precipitation-line correlation) can be differentiated so well just based on the observations alone.

Thank you. We highlighted this in the abstract by modifying lines 33-37 as:

"The spatial variations of vapor  $\delta^{18}O$  are mainly controlled by Rayleigh distillation along air mass trajectories during the pre-monsoon period, but are significantly influenced by different moisture sources, continental recycling processes and convection during moisture transport during the monsoon period. Thus, the North-South gradient observed during the pre-monsoon period is counteracted during the

**monsoon period."**

- 12) Change reference Noone and David, 2012 to Noone, 2012. We have modified this reference.
- 13) Figure 6: Can you explain the reason for a significant number of WR\_1 dried and depleted dots below Rayleigh curve? Are they within the uncertainty range of the Rayleigh curve?

Thank you for your suggestion. In current Figure 9, we added the uncertainty range of the Rayleigh curve calculated for different initial conditions of key moisture source regions (lines 523-536): during March 2019, light red and light blue Rayleigh curve are calculated for key moisture source regions of westerlies ( $\delta^2 H_0 = -168.04\%$ , T=5°C) and BoB ( $\delta^2 H_0 = -77.37\%$ , T=26.46°C) separately in (a); during July-August 2018, ight red and light blue Rayleigh curve are calculated for key moisture source regions of westerlies ( $\delta^2 H_0 = -149.64\%$ , T=6.16°C) and BoB ( $\delta^2 H_0 = -82.75\%$ , T=27.69°C) separately in (b). These initial  $\delta^2 H$  are derived from iso-GSM, the initial temperature and RH are derived from NCAR/NCEP 2.5-deg global reanalysis data.

We added following discussion in lines 558-560:

"The observations in the WR\_1 region (Fig.3c) are closer to the  $q-\delta^2 H$  Rayleigh distillation curve calculated for the key moisture source regions of westerlies, providing further evidence of the influence of water vapor source on vapor isotopes."

14) Since figure 10 is referred at multiple places in the paragraph starting at line 413, maybe change index of figure 10 to 7 to maintain the flow of the paper.

We moved Figure 10 to an earlier position in the manuscript (Section 2.4) due to the multiple references to the figure in the analysis and the fact that it is one of the results of the backward trajectory tracking, i.e., the meteorological data on the backward trajectory.

- 15) Line 439: replace 'bring' with 'brought'. We have modified 'bring' to 'brought'.
- 16) Line 442: "As continental....". Wrong grammar. We modified the grammar (lines 616-618):

"As continental recycling is known to enrich the water vapor (Salati et al., 1979) and is associated with high d-excess (Gat and Matsui, 1991; Winnick et al., 2014)."

17) Continental recycling from WR2, SR1 and SR3 is a strong factor influencing the isotope ratios. Perhaps, defining it in one line or so will help the readers.

We summarized the influence of continental recycling onWR2, SR1 and SR3 (lines 761-763):

"Except SR\_3 region, continental recycling also has a strong influence on isotopes in the WR2 and SR1 regions, which suggested by the high values of  $\delta^{18}$ O and d-excess, back-trajectories, the location on the q- $\delta$  diagram, and the higher slopes and intercepts of  $\delta^{18}$ O- $\delta^{2}$ H relationship."

- 18) Line 458: "...with a steepest slope....". Replace 'a' with 'the'.We have modified "with a steepest slope" to "with the steepest slope".
- 19) Line 493: Not all observations exhibit temperature effect. WR3 does not. We modified "all observations" to "all observations taken together".
- 20) Line 509: Another factor for the dominating effect of q at least for SR2,3,4 can be rain evaporation.

We incorporated this idea (lines 685-687):

"The absence of correlation with T suggests that the variations in q mainly reflect variations in relative humidity that are associated with different air mass origins or rain evaporation."

21) Line 512: Explain the altitude effect in a separate line.

We added the explanation of altitude effect when it first occur in lines 687-690: "The  $\delta^{18}O$  is significantly anti-correlated with Alt in the SR\_4 region (r = -0.85,

p<0.05, Table S1), consistent with the "altitude effect" (the heavy isotope concentrations in fresh water decreasing with increasing altitude) in precipitation and water vapor (Dansgaard, 1964;Galewsky et al., 2016)."

22) What is the difference between daily precipitation amount (P-daily) and temporalmean precip amount for sampled dates (P-mean)?

We clarified the definition and method to calculate the P-daily and P-mean in lines 256-259:

"When comparing the time series of GPCP data with our observations, we linearly interpolate the daily GPCP data to the location of each observation location (Pdaily).We also used the average of the GPCP precipitation over the entire observation period of about one month for each observation location (P-mean)."

**23) Line 540: Define 'dtra'.**

We changed "dtra" to "N" and defined it in the figure description of Figure 9 (lines 708-711):

"The x-axis "N" represents the number of days prior to the observations (from 1 to 10 days). For example, when the number of days is 2, the correlations is calculated with the temporal mean of meteorological data along the air mass trajectories during the 2 days before the observations."

24) Section 4.7: Going back to my major comment, why start the spatial vs synoptic comparison so late in 4.6 when you introduced the concept at the beginning of section 3? Tie this section with the previous sections.

We re-organized the manuscript and put the spatial vs synoptic comparison as section 3.3.

25) Table 2 description is incomplete.

We added "period" as the title of the  $1^{st}$  column, and "variables" as the title for the  $2^{nd}$  column (line 465).

26) I really like the summary plot 13. It provides a summary of all the meteorological processes described in the paper. Two questions: What is the significance of upward and downward pointing triangles for d-excess?

We would like to use the upward and downward pointing triangles to represent high and low values of d-excess, different colours correspond to high and low relative humidity. The idea is that the high relative humidity source of water vapor leads to low values of d-excess and vice versa. It is indeed sufficient to use only one feature, which makes it easier to read. We therefore removed the pointing of the triangles and used only the colour variations.

Please explain all the colored arrows in the figure description.

We explained all the colored arrows in the figure description (lines 881-888):

"Color gradient arrows from red to blue represent the initial to subsequent extension of the Rayleigh distillation process along the water vapor trajectory, corresponding to high to low values of  $\delta^{18}O$ ; green arrows represent high relative humidity and yellow arrows represent low relative humidity; orange twisted arrows represent continental recycling; blue-sized clouds represent strong and weak convective processes; green inverted triangles series representing low values of dexcess; yellow triangles series representing high values of d-excess."

27) 'Bay of Bengal' spelt wrong in the figures 5 and 13 Thanks you, we corrected the spelling.